# RL for Latent MDPs: Regret Guarantees and a Lower Bound

**Jeongyeol Kwon**
The University of Texas at Austin
kwonchungli@utexas.edu

**Yonathan Efroni**
Microsoft Research, NYC
jonathan.efroni@gmail.com

**Constantine Caramanis**
The University of Texas at Austin
constantine@utexas.edu

**Shie Mannor**
Technion, NVIDIA
shie@ee.technion.ac.il,
smannor@nvidia.com

## Abstract

In this work, we consider the regret minimization problem for reinforcement learning in latent Markov Decision Processes (LMDP). In an LMDP, an MDP is randomly drawn from a set of $M$ possible MDPs at the beginning of the interaction, but the identity of the chosen MDP is not revealed to the agent. We first show that a general instance of LMDPs requires at least $\Omega((SA)^M)$ episodes to even approximate the optimal policy. Then, we consider sufficient assumptions under which learning good policies requires polynomial number of episodes. We show that the key link is a notion of separation between the MDP system dynamics. With sufficient separation, we provide an efficient algorithm with local guarantee, *i.e.,* providing a sublinear regret guarantee when we are given a good initialization. Finally, if we are given standard statistical sufficiency assumptions common in the Predictive State Representation (PSR) literature (e.g., [6]) and a reachability assumption, we show that the need for initialization can be removed.

## 1 Introduction

Partially observable Markov decision processes (POMDPs) [42] give a general framework to describe partially observable sequential decision problems. In POMDPs, the underlying dynamics satisfy the Markovian property, but the observations give only partial information on the identity of the underlying states. With the generality of this framework comes a high computational and statistical price to pay: POMDPs are hard, primarily because optimal policies depend on the entire history of the process. But for many important problems, this full generality can be overkill, and in particular, does not have a way to leverage special structure. We are interested in settings where the hidden or latent (unobserved) variables have slow dynamics or are even static in each episode. This model is important for diverse applications, from serving a user in a dynamic web application [18], to medical decision making [45], to transfer learning in different RL tasks [8]. Yet, as we explain below, even this area remains little understood, and challenges abound.

Thus, in this work, we consider reinforcement learning (RL) for a special type of POMDP which we call a latent Markov decision process (LMDP). LMDPs consist of some (perhaps large) number $M$ of MDPs with joint state space $\mathcal{S}$ and actions $\mathcal{A}$. In episodic LMDPs with finite time-horizon $H$, the static latent (hidden) variable that selects one of $M$ MDPs is randomly chosen at the beginning of each episode, yet is not revealed to the agent. The agent then interacts with the chosen MDP throughout the episode (see Definition 1 for the formal description).

35th Conference on Neural Information Processing Systems (NeurIPS 2021).

**Related Work**. The LMDP framework has previously been introduced under many different names, *e.g.,* hidden-model MDP [11], Multitask RL [8], Contextual MDP [18], Multi-modal Markov decision process [45] and Concurrent MDP [9].

Learning in LMDPs is a challenging problem due to the unobservability of latent contexts. For instance, the exact planning problem is P-SPACE hard [45], inheriting the hardness of planning from the general POMDP framework. Nevertheless, the lack of dynamics of the latent variables, offers some hope. As an example, if the number of contexts $M$ is bounded, then the planning problem can be at least approximately solved (*e.g.,* by point-based value iteration (PBVI) [39], or mixed integer programming (MIP) [45]).

The most closely related work studying LMDPs is in the context of multitask RL [47, 8, 34, 18]. In this line of work, a common approach is to cluster trajectories according to different contexts, an approach that guided us in designing the algorithms in Section 3.4. However, previous work requires very long time-horizon $H \gg SA$ in order to guarantee that every state-action pair can be visited multiple times in a single episode. In contrast, we consider a significantly shorter time-horizon that scales poly-logarithmic with the number of states, *i.e.,* $H = poly \log(MSA)$. This short time-horizon results in a significant difference in learning strategy even when we get a feedback on the true context at the end of episode. We refer the readers to Appendix A for additional discussion on related work.

**Main Results.** To the best of our knowledge, none of the previous literature has obtained sample complexity guarantees or studied regret bounds in the LMDP setting. This paper addresses precisely this problem. We ask the following:

*Is there a sample efficient RL algorithm for LMDPs, with sublinear regret?*

The answer turns out to be not so simple. Our results comprise a first impossibility result, followed by positive algorithmic results under additional assumptions. Specifically:

- First, we find that for a general LMDP, polynomial sample complexity cannot be attained without further assumptions. That is, to find an approximately optimal policy we need at least $\Omega\left((SA)^M\right)$ samples, *i.e.,* at least exponential in the number of contexts $M$ (Section 3.1). This lower bound even applies to instances with deterministic MDPs.

- We find that there are several natural assumptions under which optimal policies can be learned with polynomial sample complexity. Similarly to mixture problems without dynamics, the key link is a notion of separation between the MDPs. With sufficient separation, we show that there is a planning-oracle efficient RL algorithm with polynomial sample complexity. A critical development is adapting the principle of optimism as in UCB, but to the partially observed setting where value-iteration cannot be directly applied, and thus neither can the UCRL algorithm for MDPs.

- Finally, under additional statistical sufficiency assumptions that are common in the Predictive State Representation (PSR) literature (e.g., [6]) and a reachability assumption, we show that the need for initialization can be entirely removed.

- Finally, we perform an empirical evaluation of the suggested algorithms on toy problems (Section 4), while focusing on the importance of the made assumptions.

## 2 Preliminaries

### 2.1 Problem Setup: Latent MDPs

We start with the definition of episodic reinforcement learning in latent Markov decision process:

**Definition 1 (Latent Markov Decision Process (LMDP))** *Consider a set of MDPs $\mathcal{M}$ with joint state space $\mathcal{S}$ and joint action space $\mathcal{A}$ in a finite time horizon $H$. Let $M = |\mathcal{M}|$, $S = |\mathcal{S}|$ and $A = |\mathcal{A}|$. Each MDP $\mathcal{M}_m \in \mathcal{M}$ is a tuple $(\mathcal{S}, \mathcal{A}, T_m, R_m, \nu_m)$ where $T_m : \mathcal{S} \times \mathcal{A} \times \mathcal{S} \to [0, 1]$ a transition probability maps a state-action pair and a next state to a probability, $R_m : \mathcal{S} \times \mathcal{A} \times \{0, 1\} \to [0, 1]$ a probability measure for rewards that maps a state-action pair and a binary reward to a probability, and $\nu_m$ is an initial state distribution. Let $w_1, ..., w_M$ be the mixing weights of LMDPs such that at the start of every episode, one MDP $\mathcal{M}_m \in \mathcal{M}$ is randomly chosen with probability $w_m$.*

We assume the mixing weights are uniform and known a priori, *i.e.*, $w_1 = ... = w_M = 1/M$. This is only for the ease of presentation, and does not affect any algorithmic idea or main results in this paper. The goal of the problem is to find a (possibly non-Markovian) policy $\pi$ in a policy class $\Pi$ that maximizes the expected return: $V_{\mathcal{M}}^* := \max_{\pi \in \Pi} \sum_{m=1}^{M} w_m \mathbb{E}_m^\pi \left[ \sum_{t=1}^{H} r_t \right]$ ,where $\mathbb{E}_m^\pi[\cdot]$ is expectation taken over the $m^{th}$ MDP with a policy $\pi$. If not specified otherwise, we find the best policy in a set of *history-dependent* policies $\pi : (\mathcal{S}, \mathcal{A}, \{0,1\})^* \times \mathcal{S} \to \Delta(\mathcal{A})$ that map an entire history to a probability distribution over actions.

We define the notion of regret relative to a (possibly approximate) planning oracle. Thus, suppose we have a planning-oracle with the following approximation guarantee: $V_{\mathcal{M}}^\pi \geq \rho_1 V_{\mathcal{M}}^* - \rho_2$, where $\pi$ is a returned policy when $\mathcal{M}$ is given to the planning-oracle, and $\rho_1, \rho_2$ are multiplicative and additive approximation constants respectively. We then define the regret as the comparison to the best approximation guarantee that the planning-oracle can achieve:

$$Regret(K) = \sum_{k=1}^{K} (\rho_1 V_{\mathcal{M}}^* - \rho_2) - V_{\mathcal{M}}^{\pi_k}, \tag{1}$$

where $\pi_k$ is a policy executed in the $k^{th}$ episode. For example, we can use point-based value-iteration (PBVI) [39] as a planning-oracle:

**Example 1** *PBVI [39] with discretization level $\epsilon_d > 0$ in the belief space (over $\mathbb{R}^M$) returns an $\epsilon_d H^2$ additive approximate policy. That is, equation (1) holds with $\rho_1 = 1$ and $\rho_2 = \epsilon_d H^2$.*

## 2.2 Predictive State Representation (PSR)

A partially observable dynamical system can be viewed as a model that generates a sequence of observations from observation space $\mathcal{O}$ with (controlled) actions from action space $\mathcal{A}$. A predictive state representation (PSR) is a compact description of a dynamical system with a set of observable experiments, or *tests* [41]. Specifically, a test of length $t$ is a sequence of action-observation pairs given as $\tau = a_1^\tau o_1^\tau o_2^\tau ... a_t^\tau o_t^\tau$. A *history* $h = a_1^h o_1^h a_2^h o_2^h ... a_t^h o_t^h$ is a sequence of action-observation pairs that has been generated prior to a given time. A *prediction* $\mathbb{P}(\tau|h) = \mathbb{P}(o_{1:t}^\tau | h || \boldsymbol{do}\ a_{1:t}^\tau)$ denotes the probability of seeing the test sequence from a given history, given that we intervene to take actions $a_1^\tau a_2^\tau ... a_t^\tau$. In latent MDPs, the observation space can be considered as a pair of next-states and rewards, i.e., $\mathcal{O} = \mathcal{S} \times \{0,1\}$ and $o_t = (s_{t+1}, r_t)$.

As we work with a special class of POMDPs, we customize the formulation for LMDPs. The set of histories consists of a subset of histories that end with different states, i.e., $\mathcal{H} = \bigcup_s \mathcal{H}_s$, where each element $h \in \mathcal{H}_s$ is a short sequence of state-action-rewards of length $l$ ending with state $s$:

$$h = s_1^h a_1^h r_1^h s_2^h ... s_{l-1}^h a_{l-1}^h r_{l-1}^h s = (s,a,r)_{1:l-1}^h s.$$

We define $\mathbb{P}_m^\pi(\mathcal{H}_s)$ a vector of probabilities where each coordinate is a probability of sampling each history in $\mathcal{H}_s$ in the $m^{th}$ MDP with a policy $\pi$. Likewise, each element in tests $\tau \in \mathcal{T}$ is a short sequence of action-reward-next states of length at most $l$:

$$\tau = a_1^\tau r_1^\tau s_2^\tau ... a_l^\tau r_l^\tau s_{l+1}^\tau = (a,r,s')_{1:l}^\tau.$$

We denote $\mathbb{P}_m(\mathcal{T}|s)$ as a vector of probability where each coordinate is a success probability of each test in the $m^{th}$ MDP starting from a state $s$. That is,

$$\mathbb{P}_m(\mathcal{T}|s)_i = \mathbb{P}_m(r_1^{\tau_i} s_2^{\tau_i} ... r_l^{\tau_i} s_{l+1}^{\tau_i} | s || \boldsymbol{do}\ a_1^{\tau_i} ... a_l^{\tau_i}).$$

### 2.2.1 Spectral Learning of PSRs in LMDPs

In spectral learning, we build a set of observable matrices that contains the (joint) probabilities of histories and tests, and then we can extract parameters from these matrices by performing singular value decomposition (SVD) and regressions [7]. In order to apply spectral learning techniques, we require the following technical conditions on *statistical sufficiency* of tests:

**Condition 1 (Sufficient Tests)** *For all $s \in \mathcal{S}$, for the test set $\mathcal{T}$, let $L_s = [\mathbb{P}_1(\mathcal{T}|s)|\mathbb{P}_2(\mathcal{T}|s)|...|\mathbb{P}_M(\mathcal{T}|s)]$. Then $\sigma_M(L_s) \geq \sigma_\tau$ for all $s \in \mathcal{S}$ with some $\sigma_\tau > 0$.*

Here, $\sigma_M(\cdot)$ is the minimum ($M^{th}$) singular value of a matrix. Another technical condition for spectral learning method is a rank non-degeneracy condition for sufficient histories:

**Condition 2 (Sufficient Histories)** *For all $s \in \mathcal{S}$, for the history set $\mathcal{H}_s$ ending with a state $s$, let $H_s = [\mathbb{P}_1^\pi(\mathcal{H}_s)|\mathbb{P}_2^\pi(\mathcal{H}_s)|...|\mathbb{P}_M^\pi(\mathcal{H}_s)]^\top$ with a sampling policy $\pi$. Then $\sigma_M(L_s H_s) \geq \mathbb{P}^\pi(end\ state = s) \cdot \sigma_h$ with some $\sigma_h > 0$.*

Here $\mathbb{P}^\pi(end\ state = s)$ is a probability of sampling a history ending with $s$. Along with the rank condition for tests, pairs of histories and tests can be thought as many short snap-shots of long trajectories obtained by external experts or some exploration policy (e.g., random policy in uniformly ergodic MDPs). Following the notations in [7], let $P_{\mathcal{T},\mathcal{H}_s} = L_s H_s$. Conditions 1 and 2 ensure $\sigma_M(P_{\mathcal{T},\mathcal{H}_s}) > 0$. Under these conditions, the goal of spectral learning algorithm is to output PSR parameters which are used to compute $\hat{\mathbb{P}}(\tau|h)$, the estimated probability of any future observations (or tests $\tau$) given any sampled histories $h$. We refer to Appendix E.1 for a detailed procedure.

## 2.3 Notations

We denote the underlying LMDP with true parameters as $\mathcal{M}^*$. With slight abuse of notation, we denote the $l_1$ distance between two probability distributions $\mathcal{D}_1$ and $\mathcal{D}_2$ on a random variable $X$ conditioned on event $E$ as $\|(\mathbb{P}_{X\sim\mathcal{D}_1} - \mathbb{P}_{X\sim\mathcal{D}_2})(X|E)\|_1 = \sum_{X\in\mathcal{X}} |\mathbb{P}_{X\sim\mathcal{D}_1}(X|E) - \mathbb{P}_{X\sim\mathcal{D}_2}(X|E)|$, where $\mathcal{X}$ is a support of $X$. When we do not condition on any event, we omit the conditioning on $E$. When we measure a transition or reward probability at a state-action pair $(s,a)$, we use $T$ or $R$ instead of $\mathbb{P}$. We use $\mathbb{P}_m$ to refer to the probability of any event measured in the $m^{th}$ context (or in $m^{th}$ MDP). In particular, $\mathbb{P}_m(s',r|s,a) = T_m(s'|s,a)R_m(r|s,a)$. If we use $\mathbb{P}$ without any subscript, it is a probability of an event measured outside of the context, *i.e.*, $\mathbb{P}(\cdot) = \sum_{m=1}^M w_m \mathbb{P}_m(\cdot)$. If the probability of an event depends on a policy $\pi$, we add superscript $\pi$ to $\mathbb{P}$. Similarly, $\mathbb{E}_m[\cdot]$ is expectation taken over the $m^{th}$ context and $\pi$ is added as superscript if the expectation depends on $\pi$. We use $\hat{\cdot}$ to denote any estimated quantities. $a \lesssim b$ implies $a$ is less than $b$ up to some constant and logarithmic factors. $poly(\cdot)$ means the order of polynomial complexity (up to logarithmic factors) in referenced parameters. We interchangeably use $o$, an observation, to replace a pair of next-state and immediate reward $(s',r)$ to simplify the notation. We occasionally express a length $t > 0$ history $(s_1, a_1, r_1, ..., s_t, a_t, r_t)$ compactly as $(s,a,r)_{1:t}$.

# 3 Main Results

In this section, we first obtain a hardness result for the general case. We then consider sample- and computationally efficient algorithms under additional assumptions.

## 3.1 Fundamental Limits of Learning General LMDPs

We first study the fundamental limits of the problem. In particular, we are interested in whether we can learn the optimal policy after interacting with the LMDP for a number of episodes polynomial in the problem parameters. We prove a worst-case lower bound, exhibiting an instance of LMDP that requires at least $\Omega\left((SA)^M\right)$ episodes:

**Theorem 3.1 (Lower Bound)** *There exists an LMDP such that for finding an $\epsilon$-optimal policy $\pi_\epsilon$ for which $V_{\mathcal{M}}^{\pi_\epsilon} \geq V_{\mathcal{M}}^* - \epsilon$, we need at least $\Omega\left((SA/M)^M/\epsilon^2\right)$ episodes.*

The hard instance consists of fully deterministic MDPs with possibly stochastic rewards, indicating an exponential lower bound in the number of contexts even for the easiest types of LMDPs. The example is constructed such that, in the absence of knowing true contexts, all wrong action sequences of length $M$ cannot provide any information with zero reward, whereas the only correct action sequence gets a total reward of 1 under one specific context. The construction is given in Appendix B.

Theorem 3.1 prevents a design of efficient algorithms with growing number of contexts. We note here that Theorem 3.1 holds even for restricted classes of policies, *e.g.*, memoryless policies. Furthermore, our construction of hard instances does not allow to find any approximate policy with $\rho_1 = \omega((SA)^{-M})$ within a polynomial number of episodes either. To the best of our knowledge, this

---

**Algorithm 1** Latent Upper Confidence Reinforcement Learning (L-UCRL)

Initialize visit counts $N_m(s, a)$, $N(m)$ and parameters $(\hat{T}_m, \hat{R}_m, \hat{\nu}_m)$ properly

1: **for** each $k^{th}$ episode **do**
2:  Get a policy $\pi_k$ for $\widetilde{\mathcal{M}}_k$ in Lemma 3.2
3:  Play policy $\pi_k$ and get the trajectory $\tau = (s, a, r)_{1:H}$
4:  Get an estimated belief over contexts $\hat{b}$ with either Algorithm 2 (when contexts are given), or Algorithm 3 (when we infer contexts)
5:  **for** $m = 1, ..., M$ and $t = 1, ..., H$ **do**
6:    $N_m(s_{t+1}|a_t, s_t) \leftarrow N_m(s_{t+1}|a_t, s_t) + \hat{b}(m)$
7:    $N_m(r_t|s_t, a_t) \leftarrow N_m(r_t|s_t, a_t) + \hat{b}(m)$
8:    $N_m(s_1) \leftarrow N_m(s_1) + \hat{b}(m)$
9:    Update empirical parameters $\hat{T}_m, \hat{R}_m, \hat{\nu}_m$
10:  **end for**
11: **end for**

---

is the first lower bound of its kind for LMDPs. Next, we investigate natural assumptions which help us to develop an efficient algorithm when only polynomial number of episodes are available.

### 3.2 The Critical First Step: Contexts in Hindsight

Suppose the true context of the underlying MDP is revealed to the agent at the end of each episode. We do not require any assumptions on the environments in this scenario. Note that this scenario is different from fully observable settings (*i.e.,* knowing the true context at the beginning of an episode). In the latter scenario, we would simply have $M$-decoupled RL problems in standard MDPs. While this can be considered as a "warm-up" for the sequel, it is motivated by real-world examples. Moreover, the key technical insight here will prove important for the sequel as well.

Knowing contexts in hindsight allows us to construct a confidence set for parameters:

$$\mathcal{C} = \{\mathcal{M} \mid \|(\nu_m - \hat{\nu}_m)(s)\|_1 \leq \sqrt{c_\nu/N(m)}, \ \|(T_m - \hat{T}_m)(s'|s, a)\|_1 \leq \sqrt{c_T/N_m(s, a)},$$
$$\|(R_m - \hat{R}_m)(r|s, a)\|_1 \leq \sqrt{c_R/N_m(s, a)}, \quad \forall m, s, a\}, \tag{2}$$

where $N_m(s, a)$ is the number of times each state-action pair $(s, a)$ in $m^{th}$ MDP is visited, and $N(m)$ is the number of episodes we interact with the $m^{th}$ MDP. With properly set parameters $c_T = O(S \log(K/\eta)), c_R = O(\log(K/\eta))$ and $c_\nu = O(S \log(K/\eta))$ for the confidence intervals, $\mathcal{M}^* \in \mathcal{C}$ with high probability for all $K$ episodes.

With the construction of confidence sets, it is then natural to try to design an optimistic RL algorithm, as in UCRL [22]. An obvious optimistic value in light of (2) is $\max_{\pi, \mathcal{M} \in \mathcal{C}} V_\mathcal{M}^\pi$. However, solving this optimization problem is more general than solving an LMDP. In fully observable settings, we could replace the complex optimization problem by adding a proper exploration bonus to obtain an optimistic value function [4].

In partially observable environments, value iteration is only defined in terms of belief-states and not the observed states. For this reason, existing techniques solely based on the value-iteration cannot be directly applied for LMDPs. Yet, we find that proper analysis of the Bellman update rule over the belief state reveals that an empirical LMDP with properly adjusted *hidden* rewards is optimistic:

**Proposition 3.2** *We construct an optimistic LMDP $\widetilde{\mathcal{M}}$ whose parameters are given such that:*

$$\widetilde{T}_m(s'|s, a) = \hat{T}_m(s'|s, a), \ \widetilde{R}_m^{obs}(r|s, a) = \hat{R}_m(r|s, a), \ \widetilde{\nu}_m(s) = \hat{\nu}_m(s),$$
$$\widetilde{R}_{init}^{hid}(m) = \min\left(1, \sqrt{c_\nu/N(m)}\right) \ \widetilde{R}_m^{hid}(s, a) = H \min\left(1, \sqrt{5(c_R + c_T)/N_m(s, a)}\right),$$

*where $\widetilde{R}_{init}^{hid}(m)$ is an initial hidden reward given when starting an episode with a context $m$, and $\widetilde{R}_m^{obs}(\cdot|s, a)$ is a probability measure of an observable immediate reward $r$ whereas $\widetilde{R}_m^{hid}(s, a)$ is a hidden immediate reward (that is not visible to the agent) for a state-action pair $(s, a)$ in a context $m$. Then for any policy $\pi$, the expected long-term reward is optimistic, i.e., $V_{\widetilde{\mathcal{M}}}^\pi \geq V_{\mathcal{M}^*}^\pi$.*

| **Algorithm 2** Access to True Contexts | **Algorithm 3** Inference of Contexts |
|---|---|
| **Input:** Receive true context $m^*$ in hindsight | **Input:** Trajectory $\tau = (s_1, a_1, r_1, ..., s_H, a_H, r_H)$ |
| **Output :** Belief over hidden contexts: | **Output:** Return an estimate of belief over contexts $\hat{b}$: |
| $$\hat{b}(m) = \begin{cases} 1, & \text{for } m = m^* \\ 0, & \text{for } m \neq m^* \end{cases}$$ | $$\hat{p}_m(\tau) = \Pi_{t=1}^H (\alpha + (1 - 2\alpha S)\hat{\mathbb{P}}_m(s_{t+1}, r_t\|s_t, a_t)),$$ $$\hat{b}(m) = \frac{\hat{p}_m(\tau)}{\sum_{m=1}^M \hat{p}_m(\tau)}.$$ |

Here, *hidden* reward is a deterministic reward that happens for every state-action pair, but does not appear in any observation history during the episode. We note that most existing planning algorithms can incorporate the hidden-reward structure without changes to maximize the long-term observed + hidden rewards. For instance, the PBVI algorithm [39] can be executed as it is in the planning step. Hence in each episode, we can build one optimistic model from Proposition 3.2, and call the planning-oracle to get a policy to execute for the episode. Then we simply run the policy and update model parameters in a straight-forward manner. The algorithm can be efficiently implemented as long as some efficient (approximate) planning algorithms are available.

To establish Proposition 3.2, we make use of the 'alpha vector' representation [42] of the value function of general POMDPs. Detailed analysis is deferred to Appendix C.1. With the optimistic model constructed in Proposition 3.2, planning-oracle efficient implementation is straightforward. The resulting latent upper confidence reinforcement learning (L-UCRL) algorithm is summarized in Algorithm 1. Based on the established optimism in Proposition 3.2 and by carefully bounding the on-policy errors we arrive to the following regret guarantee of L-UCRL.

**Theorem 3.3** *Let $N = HK$. The regret of the Algorithm 1 is bounded by:*
$$Regret(K) \leq \sum_{k=1}^K (V_{\widetilde{\mathcal{M}}_k}^{\pi_k} - V_{\mathcal{M}^*}^{\pi_k}) \lesssim HS\sqrt{MAN}.$$

Proof of Theorem 3.3 is given in Appendix C.4. The central result of this section, Theorem 3.3 leads to the following observation: a polynomial sample complexity is possible for the LMDP model assuming the context of the underlying MDP is supplied at the end of each episode. Next, we explore ways to relax this assumption, while keep supplying with a polynomial sample complexity guarantee.

### 3.3 When we can Infer Contexts?

Without explicit access to the true context at the end of an episode, it is natural to estimate the context from the sampled trajectory. One *sufficient* condition to infer the context is the following:

**Assumption 1 ($\delta$-Strongly Separated MDPs)** *For all $m, m_1, m_2 \in [M]$ such that $m_1 \neq m_2$, for all $(s, a) \in \mathcal{S} \times \mathcal{A}$, $l_1$ distance between probability of observations $o = (s', r)$ of two different MDPs in LMDP is at least $\delta > 0$, i.e., $\|(\mathbb{P}_{m_1} - \mathbb{P}_{m_2})(o|s, a)\|_1 \geq \delta$ for some constant $\delta > 0$.*

In order to reliably infer the true contexts the seperatedness between MDPs alone is not sufficient, since we need to estimate the contexts from the current empirical estimates of LMDPs. In order to reliably estimate the context from empirical estimate of LMDPs, we need a well-initialized empirical transition model of the LMDP:

$$\|(\hat{T}_m - T_m)(s'|s, a)\|_1, \ \|(\hat{\nu}_m - \nu_m)(s)\|_1, \ \|(\hat{R}_m - R_m)(r|s, a)\|_1 \leq \epsilon_{init}, \qquad \forall(s, a), \quad (3)$$

for some initialization error $\epsilon_{init} > 0$. Note that while the initialization error is relatively small, it can be still not good enough to obtain a near-optimal policy (i.e., it will result in a linear regret). We can consider as if the state-action pairs are already visit at least $N_0 = c_T / \epsilon_{init}^2$ times in each context.

Once the initialization is given along with separation between MDPs, we can modify Algorithm 1 to update the empirical estimate of LMDP using the estimated belief over contexts computed in Algorithm 3. Note that when we update the model parameters, we increase the visit count of state-action pair $(s, a)$ at $m^{th}$ MDP by $\hat{b}(m)$. With Assumption 1, it approximately adds a count for the

---

**Algorithm 4** (Informal) Recovery of LMDP parameters

---

Learn PSR parameters up to precision $o(\delta)$

Get clusters $\{\hat{T}_m(\cdot|s,a), \hat{R}_m(\cdot|s,a)\}_{(s,a)\in\mathcal{S}\times\mathcal{A}, m\in[M]}$ with learned PSR parameters

Build each MDP model by correctly assigning contexts to estimated model parameters

**Return** Well-initialized model $\{\hat{T}_m, \hat{R}_m\}_{m\in[M]}$

---

correctly estimated context, but even without Assumption 1, the update steps can still be applied. In fact, this is equivalent to an implementation of the so-called (online) expectation-maximization (EM) algorithm [10] for latent MDPs. Thus Algorithm 1 with Algorithm 3 essentially results in combining L-UCRL and the EM algorithm.

In terms of performance guarantees, using Algorithm 3 as a sub-routine for L-UCRL gives the same order of regret as in Theorem 3.3 as long as the true context can be almost reliably inferred:

**Theorem 3.4** *Suppose Assumption 1 holds with $H > C \cdot \delta^{-4}\log^2(1/\alpha)\log(N/\eta)$ for some absolute constants $C, \delta > 0$, and a parameter $\alpha > 0$ such that $\alpha\ln(1/\alpha) \le \delta^2/(200S)$. If the initialization parameters satisfy equation (3) with some initialization error $\epsilon_{init} \le \delta^2/(200\ln(1/\alpha))$, then with probability at least $1 - \eta$, the regret of Algorithm 1 is bounded by:*

$$Regret(K) \lesssim HS\sqrt{MAN}.$$

The proof of Theorem 3.4 is given in Appendix D.3. The provable guarantees are only given for well-separated LMDPs. Nevertheless, we empirically evaluate Algorithm 1 as a function of separations and initialization (see Figure 1).

An interesting consequence of Assumption 1 is that the length of episode can be logarithmic in the number of state and actions. With much longer time-horizons $H \ge \Omega(S^2A/\delta^2)$, [8, 18] assumed similar $\delta$-separation only for some $(s,a)$ pairs. While Assumption 1 requires a stronger assumption of $\delta$-separation for all state-actions, the requirement on the time-horizon can be significantly weaker with large state and action spaces. For a more discussion on the separation condition, we refer the readers to Appendix D.1.

### 3.4 Learning LMDPs without Initialization

Finally, we discuss efficient initialization with some additional assumptions. Clustering trajectories is the cornerstone of all our technical results, as this allows us to estimate the parameters of each hidden MDP and then apply the techniques of Section 3.2. The challenge is how to cluster when we have short trajectories, and no good initialization.

The key is again in Assumption 1. In Section 3.3, we use a good initialization to obtain accurate estimates of the belief states. These can then be clustered, thanks to Assumption 1, allowing us to obtain the true label in hindsight. Without initialization, we cannot accurately compute the belief state, so this avenue is blocked. Instead, our key idea is to leverage a predictive state representation (PSR) of the POMDP dynamics, and then show that Assumption 1 allows us to cluster in this space.

Algorithm 4 gives our approach. We first explain the high-level idea, and subsequently detail some of the more subtle points. Suppose we have PSR parameters allowing us to estimate $\mathbb{P}(o|h\|\mathbf{do}\ a)$, (the probabilities of any future observations $o = (s', r)$ given a history $h$ and intervening action $a$) to within accuracy $o(\delta)$. We then show that we can again apply Assumption 1, to (almost) perfectly cluster the MDPs by true context at the end of the episode. After we collect transition probabilities at all states near the end of episode, we can construct a full transition model for each MDP.

Learning the PSR to sufficient accuracy requires an additional assumption. We show that the following standard assumption on statistical sufficiency of histories and tests, is sufficient for our purposes (see also Section 2.2.1 and Appendix E.1):

**Assumption 2 (Sufficient Tests/Histories)** *Let $\mathcal{T}$ and $\mathcal{H}$ be the set of all possible tests and histories of length $l = O(1)$ respectively, with a given sampling policy $\pi$ (e.g., uniformly random policy) for histories $\mathcal{H}$. $\mathcal{T}$ and $\mathcal{H}$ satisfy Condition 1 and 2 respectively.*

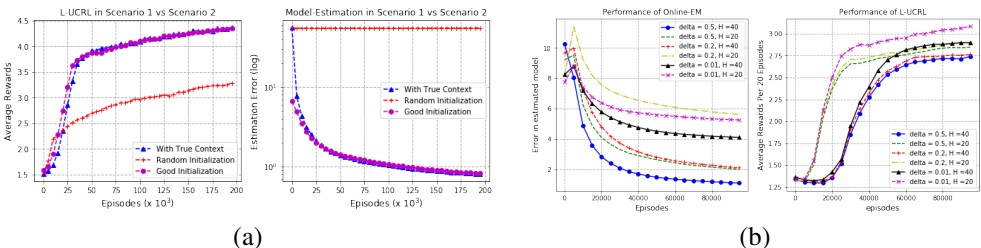

(a)                   (b)

**Figure 1:** (a) L-UCRL when true contexts are revealed in hindsight and when we run with the EM algorithm. (b) EM + L-UCRL (Algorithm 1) under different levels of separation and horizon length.

While the worst-case instance may require $l \geq M$ to satisfy the full-rank conditions, we assume that the length of sufficient tests/histories is $l = O(1)$. In fact, $l = 1$ has been (implicitly) the common assumption in the literature on learning POMDPs [20, 5, 17, 25]. Empirically, we observe that the more MDPs differ, the more easily they satisfy Assumption 2. See Figure 2. At this point, we are not aware whether sample-efficient learning is possible with only Assumption 1.

Though the main idea and key assumption are above, a few important details and technical assumptions remain to complete this story. The primary guarantee still required is that we have access to an exploration policy with sufficient mixing, to guarantee we can collect all required information to perform the PSR-based clustering. The following assumption ensures that additional $\tilde{O}(M/\alpha_2)$ sample trajectories obtained with the exploration policy $\pi$ can provide $M$ clusters of estimated one-step predictions $\mathbb{P}_m(o|s, a)$ for every state $s$ and intervening action $a$.

**Assumption 3 (Reachability of States)** *There exists a priori known exploration policy $\pi$ such that, for all $m \in [M]$ and $s \in \mathcal{S}$, we have $\mathbb{P}_m^\pi(s_{H-1} = s) \geq \alpha_2$ for some $\alpha_2 > 0$.*

A subtle point here is that we still have an ambiguity issue in the ordering of contexts (or labels) assigned in different states, which prevents us from recovering the full model for each context. We resolve this issues ambiguity assuming the MDP is connected, and give the full description of Algorithm 4 in Appendix E.2. We conclude this section with an (informal) end-to-end guarantee:

**Theorem 3.5 (Informal)** *Let Assumption 2 hold for an LMDP instance with a sampling policy $\pi$. Furthermore, assume the LMDP satisfies Assumptions 1 and 3. Then there exists an algorithm such that with probability at least $2/3$, it returns a good initialization of LMDP parameters that satisfies (3) in time $poly(A^l, S, H, M, \sigma_h^{-1}, \sigma_\tau^{-1}, \alpha_2^{-1}, \delta, \epsilon_{init})$.*

Theorem 3.5 completes the pipeline for learning in latent MDPs: we initialize the parameters by the estimated PSR and clustering (see Appendix E) up to *some* accuracy, and then we run L-UCRL to refine the model and policy up to *arbitrary* accuracy (Algorithm 1). Full version of Theorem 3.5 can be found in Theorem E.3.

## 4 Experiments

In this section, we evaluate the proposed algorithm on synthetic data. Our first two experiments illustrate the performance of L-UCRL (Algorithm 1) for various levels of separation and quality of initialization. Then, we empirically study the performance of the PSR-Clustering algorithm for randomly generated LMDPs for different levels of separation and time-horizon.

### 4.1 The Value of True Contexts in Hindsight

We first study the importance of receiving the true contexts in hindsight for the approach analyzed in this work, by comparing the performance of Algorithm 1 when instantiating it with Algorithm 2 or 3 as a sub-routine. We generate random instances of LMDPs of size $M = 7, S = 15, A = 3$ and set the time-horizon $H = 30$. The reward distribution is set to be 0 for most state-action pairs. We compare when we give a true context to the algorithm (Algorithm 2) and when we infer a context with random initialization or good initialization (Algorithm 3). In the latter, it is equivalent to

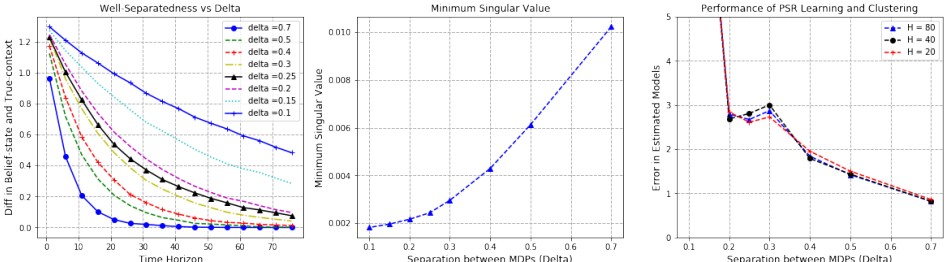

**Figure 2:** PSR learning and Clustering (Algorithm 4). **Left:** Convergence of belief state. **Middle:** $M^{th}$ singular value of sufficient histories/tests matrix $P_{\mathcal{T},\mathcal{H}}$. **Right:** Accuracy of the estimated model.

running the EM algorithm for the model estimation. For the planning algorithm, we use Q-MDP heuristic [32] which shows good empirical performance. We measure the model estimation error as $error := \min_{\sigma \in \text{Perm}_M} \sum_{(m,s,a)} \|(\mathbb{P}_m - \hat{\mathbb{P}}_{\sigma(m)})(s',r|s,a)\|_1$, where $\text{Perm}_M$ denotes all length $M$ permutation sequences. The measured errors are averaged over 10 independent experiments.

The experimental results are given in Figure 1(a). When the true context is given at the end of episode (with Algorithm 2), L-UCRL converges to the optimal policy as our theory suggests. On the other hand, if the true context is not given (with Algorithm 3), the quality of initialization becomes crucial; when the model is poorly initialized, the estimated model converges to a local optimum which leads to a sub-optimal policy. When the model is well-initialized, L-UCRL performs as well as when true contexts are given in hindsight.

## 4.2 Performance of L-UCRL with Good Initialization

In our second experiment, we focus on the performance of L-UCRL (Algorithm 1) along with Algorithm 3 under different levels of separation ($\delta$ in Assumption 1) when approximately good model parameters are given. For various levels of $\delta$, we generate the parameters for transition probabilities randomly while keeping the distance between different MDPs to satisfy $\delta \leq \|(T_{m_1} - T_{m_2})(s'|s,a)\|_1 \leq 2\delta$ for $m_1 \neq m_2$.

We show the error in the estimated model and average long-term rewards in Figure 1(b). When the separation is sufficient (larger $\delta$ or $H$), the estimated model converges fast to the true parameters. When the separation gets small (smaller $\delta$ or $H$), the convergence speed gets slower. This type of transition in the convergence speed of EM (the update of model parameters with Algorithm 3) is observed both in theory and practice when the overlap between mixture components gets larger (*e.g.,* [29]). On the other hand, the policy steadily improves regardless of the level of separation. We conjecture that this is because the optimal policy would only need the model to be accurate in the total-variation distance, not in the actual estimated parameters.

## 4.3 Initialization with PSR and Clustering

In the third experiment, we evaluate the initialization algorithm (Algorithm 4) for randomly generated LMDP instances. Since PSR learning requires a (relatively) large number of short sample trajectories, we evaluate this step on smaller instances with $S = 7, A = 2, M = 3$. The LMDP instances are generated similarly as in the second experiment with different levels of $\delta$ and $H$. The reward and initial distributions are set the same across all MDPs. To learn the parameters of PSR, we run $10^6$ episodes with $H = 4$. We assume histories and tests of length 1 are statistically sufficient with the uniformly random policy. In the clustering step, we run an additional $5 \cdot 10^3$ episodes to obtain longer trajectories of length $H = 20, 40$ and $80$. We report the experimental results in Figure 2.

We first observe how the level of separation $\delta$ between MDPs impacts trajectory separation, i.e., belief state vs true label (left). Recall that this separation property is the key for clustering trajectories. We then examine the performance of Algorithm 4 (see full Algorithm 5) for various levels of separation. Empirically, it succeeds to get a good initialization of an LMDP model when we have sufficient separation. As the separation level decreases, the algorithm starts to fail (Right). There are two possible sources of the failure: (1) the belief state is far from the true context, and (2) the similarity between MDPs drops the $M^{th}$ singular value of $P_{\mathcal{T},\mathcal{H}}$ (Middle). We can compensate for (1) if we

have a longer time-horizon to infer true contexts, as in the leftmost graph. For (2), if the $M^{th}$ singular value drops, we require more samples for the estimation of PSR parameters. In our experiments, as we decreased $\delta$ we found that failure in the spectral learning step was the more significant of the two.

## 5 Future Work

There are several interesting research avenues in continuation of this work. An interesting direction is to study RL algorithms for LMDPs with no underlying assumptions. Although our lower bound suggests such an algorithm necessarily suffers an exponential dependence in the number of contexts, if this number is small, such dependence might be acceptable. Specifically, we conjecture that general LMDPs can be learned with sample complexity of $poly\left((HSA)^M, \epsilon^{-1}\right)$. For a special case when MDPs are deterministic, we show that the exponential dependence in $M$ is sufficient In Appendix G. The case for general LMDPs is an interesting open question. Furthermore, a needed empirical advancement is to design efficient ways to learn the set of sufficient histories/tests for learning predictive state representation of LMDPs. This can dramatically improve the performance of our algorithms when a sufficiently good initial model needs to be learned.

### Acknowledgement

The research was funded by NSF grant 2019844, and by the Army Research Office and was accomplished under Cooperative Agreement Number W911NF-19-2-0333. The views and conclusions contained in this document are those of the authors and should not be interpreted as representing the official policies, either expressed or implied, of the Army Research Office or the U.S. Government. The U.S. Government is authorized to reproduce and distribute reprints for Government purposes notwithstanding any copyright notation herein.

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
