# Appendix A    Related Work

Due to the vast volume of literature on the RL, we only review the most closely related research to our problem.

**Previous Study on LMDPs**    As mentioned earlier, LMDPs have been previously introduced with different names. In the work of [11, 45, 9], the authors study the planning problem in LMDPs, when the true parameters of the model is given. The authors in [45] have shown that, as for POMDPs [38], it is P-SPACE hard to find an exact optimal policy, and NP-hard to find an optimal memoryless policy of an LMDP. On the positive side, several heuristics are proposed for practical uses of finding the optimal memoryless policy [45, 9].

LMDP has been studied in the context of multitask RL [47, 8, 34, 18]. In this line of work, a common approach is to cluster trajectories according to different contexts under some separation assumption, an approach that guided us in designing the algorithms in Section 3.4. However, in this line of work, the authors assume very long time-horizon such that they can visit every state-action pair multiple times in a single episode. In order to satisfy such assumption, the time-horizon must be at least $H \geq \Omega(SA)$. In contrast, we consider a significantly shorter time-horizon that scales poly-logarithmic with the number of states, *i.e.,* $H = poly \log(MSA)$. This short time-horizon results in a significant difference in learning strategy even when we get a feedback on the true context at the end of episode.

**Approximate Planning in POMDPs**    The study of learning in partially observable domains has a long history. Unlike in MDPs, finding the optimal policy for a POMDP is P-SPACE hard even with known parameters [42]. Even finding a memoryless policy is known to be NP-hard [31]. Due to the computational intractability of exact planning, various approximate algorithms and heuristics within a policy class of interest [21, 37, 43, 44, 39, 30]. Since LMDP is a special case of POMDP, any of these methods can be applied to solve LMDP. We will assume that the planning-oracle achieves some approximation guarantees with respect to maximum long-term rewards obtained by the optimal policy. We show that when the context is identifiable in hindsight, then we can quickly perform as good as the policy obtained by the planning-oracle with true parameters.

**Spectral Methods for POMDPs**    Previous studies of partially-observed decision problems assumed the number of observations is larger than the number of hidden states, as well as, that a set of single observations forms *sufficient statistics* to learn the hidden structure [5, 17]. With such assumptions, one can apply tensor-decomposition methods by constructing multi-view models [2, 1] and recovering POMDP parameters under uniformly-ergodic (or stationary) assumption on the environment [5, 17]. Our work is differentiated from the mentioned works in two aspects. First, for LMDPs, the observation space is *smaller* than the hidden space. Therefore constructing a multi-view model with a set of single observations is not enough to learn hidden structures of the system. Second, we are not aware of any natural conditions for tensor-decomposition methods to be applicable for learning LMDPs. Therefore, we do not pursue the application of tensor-methods in this work.

**Predictive State Representation**    Since the introduction of PSR [33, 41], it has become one major alternative to POMDPs for modeling partially-observed environments. The philosophy of PSR is to express the internal state only with a set of observable experiments, or *tests*. Various techniques have been developed for learning PSR with statistical consistency and global optimality guarantees [20, 7, 19]. We use the PSR framework to get an initial estimate of the system. However, the sample complexity of learning PSR is quite high [24] (see also our finite-sample analysis of spectral learning technique in Section E.1). Therefore, we only learn PSR up to some desired accuracy and convert it to an LMDP parameter by clustering of trajectories (Section E.2) to warm-start the optimal policy learning.

**Other Related Work**    In a relatively well-studied setting of contextual decision processes (CDPs), the context is always given as a side information *at the beginning* of the episode [23, 36]. This makes the decision problem a fully observed decision problem. LMDP is different since the context is hidden. The main challenge comes from the partial observability which results in significant differences in terms of analysis from CDPs. Another line of work on decision making with latent contexts considers the problem of latent bandits [35, 16, 15]. It would be interesting to understand whether any previous results on latent bandits can be extended to latent MDPs. Another line of research on theoretical

studies with partial observability considers the environment with rich observations [26, 12, 13]. Rich observation setting assumes that any observation happens from only one internal state which removes the necessity to consider histories, and thus the nature of the problem is different from our setting. Recent work considers a sample-efficient algorithm for undercomplete POMDPs [25], *i.e.,* when the observation space is larger than the hidden state space and a set of single observations is statistically sufficient to learn hidden structures. In contrast, our problem is a special case of POMDPs where the observation space is smaller than the hidden state space.

## Appendix B    Fundamental Limits of Learning LMDPs (Theorem 3.1)

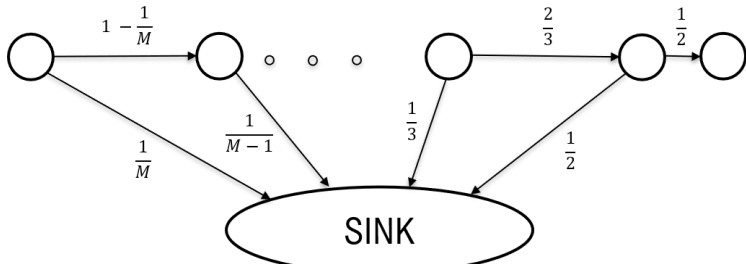

**Figure 3:** External view of the system dynamics with wrong action sequences without context information. Arrows indicate transition probabilities of a surrogate Markov chain that represents the external view.

We consider the following constructions with $M$ deterministic MDPs and $H = M$, $S = M + 1$ with $A$ actions:

1. At the start of episode, one of $M$-MDPs in $\{\mathcal{M}_1, \mathcal{M}_2, ..., \mathcal{M}_M\}$ are chosen with probability $1/M$.

2. At each time step, each MDP either goes to the next state or go to the SINK-state depending on the action chosen at the time step. Once we fall into the SINK state, we keep staying in the SINK state throughout the episode without any rewards.

3. Rewards of all state-action pairs are all 0 except at time step $t = M$ with the right action choice $a_M$ only in the first MDP $\mathcal{M}_1$.

4. At time step $t = 1$, there are three state-transition possibilities:
   - $\mathcal{M}_1$: For all actions $a \in \mathcal{A}$ except $a_1$, we go to the SINK state. For the action $a_1$, we go to the next state.
   - $\mathcal{M}_M$: For all actions $a \in \mathcal{A}$ except $a_1$, we go to the *next* state. For the action $a_1$, we go to the SINK state.
   - $\mathcal{M}_2, ..., \mathcal{M}_{M-1}$: For all actions $a \in \mathcal{A}$, we go to the next state.

5. At time step $t = 2$, we again have three cases but now $\mathcal{M}_1$ and $\mathcal{M}_K$ would look the same:
   - $\mathcal{M}_1, \mathcal{M}_M$: For all actions $a \in \mathcal{A}$ except $a_2$, we go to the SINK state. For the action $a_2$, we go to the next state.
   - $\mathcal{M}_{M-1}$: For all actions $a \in \mathcal{A}$ except $a_2$, we go to the next state. For the action $a_2$, we go to the SINK state.
   - $\mathcal{M}_2, ..., \mathcal{M}_{M-2}$: For all actions $a \in \mathcal{A}$, we go to the next state.

   ...

6. At time step $t = M - 1$,
   - $\mathcal{M}_1, \mathcal{M}_3, ..., \mathcal{M}_M$: For all actions $a \in \mathcal{A}$ except $a_{M-1}$, we go to the SINK state. For the action $a_{M-1}$, we go to the next state.
   - $\mathcal{M}_2$: For all actions $a \in \mathcal{A}$ except $a_{M-1}$, we go to the next state. For the action $a_{M-1}$, we go to the SINK state.

7. At time step $t = M$, there are two possibilities of getting rewards:
   - $\mathcal{M}_1$: For the action $a_M \in \mathcal{A}$, we get reward 1. For all other actions, we get no reward.

- $\mathcal{M}_2, ..., \mathcal{M}_M$: For all actions $a \in \mathcal{A}$, we get no reward.

Note that the right action sequence is $a^* = (a_1, a_2, ..., a_M)$. However, without the information on true contexts, the system dynamics with any wrong action sequence among the $A^M - 1$ wrong sequences, is exactly viewed as Figure 3 with zero rewards, i.e.,

$$\mathbb{P}(s_{1:H}, r_{1:H} \| \mathbf{do}\ a_{1:H}^{(1)}) = \mathbb{P}(s_{1:H}, r_{1:H} \| \mathbf{do}\ a_{1:H}^{(2)}),$$

for any two wrong action sequences $a^{(1)} = a_{1:H}^{(1)}$, $a^{(2)} = a_{1:H}^{(2)}$ such that $a^{(1)}, a^{(2)} \neq a^*$. The probability distribution of observation sequences with any wrong action sequence is the same as the distribution of sequences generated by the surrogate Markov chain in Figure 3. Therefore, we cannot gain any information from executing wrong action sequences besides of eliminating this wrong action sequence. Note that there are $A^M$ possible choice of action sequences. Hence the problem is reduced to find one specific sequence among $A^M$ possibilities without any other information on the correct action sequence. It leads to the conclusion that before we play most of $A^M$ action sequences, we cannot find the correct one.

We formalize the lower bound argument with an $\epsilon$-additive approximation factor. First, we get the lower bound $\Omega(A^M/\epsilon^2)$ by properly adjusting the reward distribution at the last time step $t = H$ as the following:

- $\mathcal{M}_1$: For the action $a_M \in \mathcal{A}$ at time step $H = M$, we get a reward from Bernoulli distribution $Ber(1/2 + \epsilon)$. For all other actions, we get a reward from Bernoulli distribution $Ber(1/2)$.

- $\mathcal{M}_2, ..., \mathcal{M}_M$: For all actions $a \in \mathcal{A}$, we get a reward from Bernoulli distribution $Ber(1/2)$.

Note that the distribution of the final reward with $a^*$ is $1/M \cdot Ber(1/2+\epsilon) + (M-1)/M \cdot Ber(1/2)$, whereas the distribution of all other action sequences is $Ber(1/2)$. Similarly to the above, for any wrong action sequence, the probability of observations is identical. Hence, identifying the optimal action sequence $a^*$ among all $A^M$ action sequences requires $\Omega(A^M/\epsilon^2)$ trials, i.e., to identify an $\epsilon$ optimal arm among $A^M$ actions.

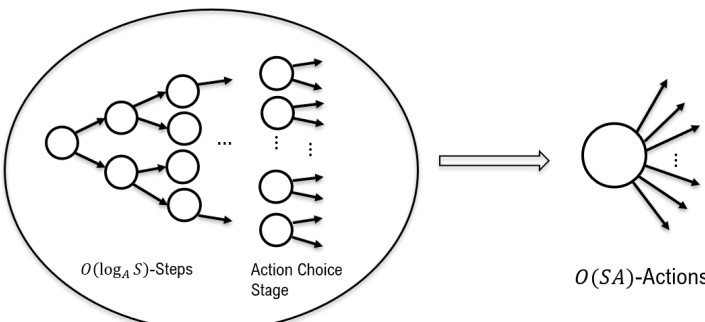

**Figure 4:** Effectively amplifying the number of actions

Now the above argument can be easily extended to get a lower bound of $\Omega\left(\left(\frac{SA}{M}\right)^M \cdot \frac{1}{\epsilon^2}\right)$, by effectively amplifying the number of actions up to $O\left(\frac{SA}{M}\right)$. That is, we can amplify the effective number of actions by considering a big state consisting of a tree of states with $O(\log_A S)$-depth (see Figure 4). Since we have such $M$ big states, total number of states is $O(MS)$ in our lower bound example with amplified number of actions, or conversely if the number of total states is $S$, then effective number of actions is $O(SA/M)$ per each big state. This gives the $\Omega\left((SA/M)^M/\epsilon^2\right)$ lower bounds. For the completely formal argument, we refer readers to the lower bound argument for multi-armed bandits developed in [14]. Since all action-sequences yield the same trajectory distributions (except the last reward with the correct action-sequence), we can apply their proof to our problem with a minor adjustment as if we have a bandit problem with $(SA/M)^M$ number of arms (e.g., we consider a class of action sequences as an entire action set in bandit settings).

# Appendix C  Analysis of L-UCRL when True Contexts are Revealed

## C.1  Analysis of Optimism in Alpha-Vectors

We start with an important observation that the upper confidence bound (UCB) type algorithm can be implemented in the belief-state space. Even though the exact planning in a belief-state space is not implementable, we can still discuss how the value iteration is performed in partially observable domains. Let $h$ be an entire history at time $t$, and denote $b(h)$ be a belief state over $M$ MDPs corresponding to a history $h$. The value iteration with a (history-dependent) policy $\pi$ is given as

$$Q_t^\pi(h,s,a) = b(h)^\top \bar{R}(s,a) + \mathbb{E}_{s',r|h,s,a}[V_{t+1}^\pi(h',s')],$$

for $t = 1, ..., H$, where $h' = ha(rs')$ is a concatenated history. Here $Q_t^\pi(h,s,a)$ and $V_t^\pi(h,s)$ are state-action-value and state-value function at time step $t$ respectively given a history $h$ and a policy $\pi$. $\bar{R}(s,a) \in \mathbb{R}^M$ is a vector where value of $m^{th}$ coordinate $\bar{R}_m(s,a)$ is an expected immediate reward at $(s,a)$ in $m^{th}$ MDP, *i.e.*, $\bar{R}_m(s,a) = \mathbb{E}_{r \sim R_m(r|s,a)}[r]$. In case there exists a hidden reward $R_m^{hid}(s,a)$, we define $\bar{R}_m(s,a) = \mathbb{E}_{r \sim R_m(r|s,a)}[r] + R_m^{hid}(s,a)$. At the end of episode, we set $V_{H+1}^\pi = 0$. We first need the following lemma on the policy evaluation procedure of a POMDP.

**Lemma C.1** *For any history $h$ at time $t$, the value function for a policy $\pi$ can be written as*

$$V_t^\pi(h,s) = b(h)^\top \alpha_{t,s}^{h,\pi}, \tag{4}$$

*for some $\alpha_{t,s}^{h,\pi} \in \mathbb{R}^M$ uniquely decided by $t, s, h$ and $\pi$.*

*Proof.*  We will show that the value of $\alpha_{t,s}^{h,\pi}$ is decided only by a history and policy, and is not affected by the history to belief-state mapping. On the other hand, the Bayesian update for $h'$ is given by

$$b_m(ha(rs')) = \frac{b_m(h)T_m(s'|s,a)R_m(r|s,a)}{\sum_m b_m(h)T_m(s'|s,a)R_m(r|s,a)} = \frac{b_m(h)\mathbb{P}_m(s',r|s,a)}{\mathbb{P}(s',r|h,s,a)}.$$

Thus, the value iteration for policy evaluation in LMDPs can be written as:

$$Q_t^\pi(h,s,a) = b(h)^\top \bar{R}(s,a) + \sum_{(s',r)} \sum_m b_m(h)\alpha_{t+1,s'}^{ha(rs'),\pi}(m)\mathbb{P}_m(s',r|s,a),$$

$$V_t^\pi(h,s) = \sum_a \pi(a|h)Q_t^\pi(h,s,a). \tag{5}$$

Let us explain how the alpha vectors [42] can be constructed recursively from the time step $H+1$. Note that $V_{H+1}(h,s) = 0$ for any $h$ and $s$, therefore $\alpha_{H+1,s}^{h,\pi} = 0$. Then we can define the set of alpha vectors recursively such that

$$\alpha_{t,s}^{h,a,*,\pi}(m) = \bar{R}_m(s,a),$$
$$\alpha_{t,s}^{h,a,(s',r),\pi}(m) = \mathbb{P}_m(s',r|s,a)\alpha_{t+1,s'}^{ha(s'r),\pi}(m) \qquad \forall(s,a,r,s'), \tag{6}$$

Finally, the alpha vector for the value with respect to $h$ is constructed as

$$\alpha_{t,s}^{h,\pi}(m) = \sum_a \pi(a|h)\left(\alpha_{t,s}^{h,a,*,\pi}(m) + \sum_{s',r} \alpha_{t,s}^{h,a,(s',r),\pi}(m)\right).$$

Note that in the construction of alpha vectors, the mapping from history to belief-state is not involved, and the value function can be represented as $V_t^\pi(h,s) = b(h)^\top \alpha_{t,s}^{h,\pi}$.  □

Now consider the optimistic model defined in Lemma 3.2. For the optimistic model, the intermediate alpha vectors are constructed with the following recursive equation:

$$\tilde{\alpha}_{t,s}^{h,a,*,\pi}(m) = \mathbb{E}_{r \sim \tilde{R}_m^{obs}(r|s,a)}[r] + \tilde{R}_m^{hid}(s,a),$$
$$\tilde{\alpha}_{t,s}^{h,a,(s',r),\pi}(m) = \tilde{T}_m(s'|s,a)\tilde{R}_m^{obs}(r|s,a)\tilde{\alpha}_{t+1,s'}^{ha(s'r),\pi}(m) \qquad \forall(s,a,r,s'), \tag{7}$$

From the constructions of alpha vectors above, we can show the optimism in alpha vectors:

**Lemma C.2** *Let $\alpha_{t,s}^{h,\pi}$ and $\tilde{\alpha}_{t,s}^{h,\pi}$ be alpha vectors constructed with $\mathcal{M}^*$ and $\widetilde{\mathcal{M}}$ respectively. Then for all $t, s, h, \pi$, we have*

$$\tilde{\alpha}_{t,s}^{h,\pi} \succeq \alpha_{t,s}^{h,\pi}.$$

The lemma implies that if the history is mapped to the same belief states in both models, then we also have the optimism in value functions. Note that in general, different models will lead each history to different belief states. At the initial time-step, however, we start from similar belief states, and we can claim Lemma 3.2. The remaining proof of Lemma 3.2 is given in Appendix C.3.

## C.2   Proof of Lemma C.2

We show this by mathematical induction moving reverse in time from $t = H$. The inequality is trivial when $t = H + 1$ since all $\alpha_{H+1,s}^{h,\pi} = \tilde{\alpha}_{H+1,s}^{h,\pi} = 0$ for any $h, \pi, s$. Now we investigate $\alpha_{t,s}^{h,\pi}(m)$. It is sufficient to show that for all $a \in \mathcal{A}$,

$$\alpha_{t,s}^{h,a,*,\pi}(m) + \sum_{s',r} \alpha_{t,s}^{h,a,(s',r),\pi}(m) \leq \tilde{\alpha}_{t,s}^{h,a,*,\pi}(m) + \sum_{s',r} \tilde{\alpha}_{t,s}^{h,a,(s',r),\pi}(m).$$

Recall equations for alpha vectors (6), (7).

$$\tilde{\alpha}_{t,s}^{h,a,*,\pi}(m) + \sum_{s',r} \tilde{\alpha}_{t,s}^{h,a,(s',r),\pi}(m) - \alpha_{t,s}^{h,a,*,\pi}(m) - \sum_{s',r} \alpha_{t,s}^{h,a,(s',r),\pi}(m)$$

$$\geq \left( \mathbb{E}_{r \sim \tilde{R}_m^{obs}(r|s,a)}[r] - \mathbb{E}_{r \sim R_m^{obs}(r|s,a)}[r] \right) + \tilde{R}_m^{hid}(s,a)$$

$$+ \sum_{s',r} \left( \tilde{T}_m(s'|s,a)\tilde{R}_m^{obs}(r|s,a)\tilde{\alpha}_{t+1,s'}^{ha(s',r),\pi}(m) - T_m(s'|s,a)R_m(r|s,a)\alpha_{t+1,s'}^{ha(s',r),\pi}(m) \right)$$

$$\geq \left( \mathbb{E}_{r \sim \tilde{R}_m^{obs}(r|s,a)}[r] - \mathbb{E}_{r \sim R_m(r|s,a)}[r] \right) + H \min\left( 1, \sqrt{5(c_R + c_T)/N_m(s,a)} \right)$$

$$+ \sum_{s',r} \left( \tilde{T}_m(s'|s,a)\tilde{R}_m^{obs}(r|s,a)\alpha_{t+1,s'}^{ha(s',r),\pi}(m) - T_m(s'|s,a)R_m(r|s,a)\alpha_{t+1,s'}^{ha(s',r),\pi}(m) \right),$$

where the last inequality comes from the induction hypothesis. On the other hand, note that $\tilde{R}_m^{obs}$ and $\tilde{T}_m$ are simply empirical estimates after visiting the state-action pair $N_m(s,a)$ times. Thus, it is easy to see that with high probability,

$$\left| \mathbb{E}_{r \sim \tilde{R}_m^{obs}(r|s,a)}[r] - \mathbb{E}_{r \sim R_m^{obs}(r|s,a)}[r] \right| \leq \|\hat{R}_m - R_m(r|s,a)\|_1 \leq \sqrt{c_R/N_m(s,a)},$$

$$\sum_{s',r} \left| \hat{T}_m(s'|s,a)\hat{R}_m(r|s,a)\alpha_{t+1,s'}^{h,a,(s',r),\pi}(m) - T_m(s'|s,a)R_m(r|s,a)\alpha_{t+1,s'}^{h,a,(s',r),\pi}(m) \right|$$

$$\leq H \sum_{s',r} \left| \hat{T}_m(s'|s,a)\hat{R}_m(r|s,a) - T_m(s'|s,a)R_m(r|s,a) \right|$$

$$\leq H \left( \|(\hat{T}_m - T_m)(s'|s,a)\|_1 + \|(\hat{R}_m - R_m)(r|s,a)\|_1 \right) \leq H \left( \sqrt{c_R/N_m(s,a)} + \sqrt{c_T/N_m(s,a)} \right),$$

where we used that all alpha vectors in the original system satisfies $\|\alpha_{t,s}^{h,\pi}\|_\infty \leq H$ for all $t, s, h, \pi$. This completes the proof of Lemma C.2.

## C.3   Proof of Lemma 3.2

The remaining step is to show the optimism at the initial time. When $t = 1$, history $h$ is simply the initial state $s$. The belief state after observing the initial state is given by

$$b_m(s) = \frac{w_m \nu_m(s)}{\sum_{s'} w_m \nu_m(s')}, \quad \tilde{b}_m(s) = \frac{w_m \tilde{\nu}_m(s)}{\sum_{s'} w_m \tilde{\nu}_m(s')}.$$

The expected long-term reward with $\pi$ for each model is therefore

$$V_{\mathcal{M}^*}^\pi = \sum_s \mathbb{P}(s_1 = s)V(s) = \sum_s \mathbb{P}(s_1 = s)b(s)^\top \alpha_{1,s}^{s,\pi} = \sum_s \sum_m w_m \nu_m(s)\alpha_{1,s}^{s,\pi}(m),$$

$$V_{\widetilde{\mathcal{M}}}^{\pi} = \sum_s \sum_m w_m \tilde{\nu}_m(s) \tilde{\alpha}_{1,s}^{s,\pi}(m).$$

Following the similar arguments, we have

$$V_{\widetilde{\mathcal{M}}}^{\pi} - V_{\mathcal{M}^*}^{\pi} \geq H \sum_m w_m \sqrt{c_\nu/N(m)} - H \sum_m w_m \sum_s |\nu_m(s) - \tilde{\nu}_m(s)| \geq 0,$$

which proves the claim of Lemma 3.2.

## C.4 Proof of Theorem 3.3

Let us define a few notations. Suppose $\mathcal{M} = (\mathcal{S}, \mathcal{A}, T_m, R_m, \nu_m)$ a LMDP and a context $m$ is randomly chosen at the start of an episode following a probability distribution $(w_1, w_2, ..., w_M)$. Let $\bar{R}_m(s,a) = \mathbb{E}_{r \sim R_m(r|s,a)}[r]$ be an expected (observable) reward of taking action $a$ at $s$ in $m^{th}$ MDP. With a slight abuse in notation, we use $\mathbb{E}_{\pi,\mathcal{M}}[\cdot]$ to simplify $\mathbb{E}_{m \sim (w_1,...,w_M)} \left[ \mathbb{E}_{\pi,T_m,R_m,\nu_m}[\cdot] \middle| m \right] = \sum_{m=1}^{M} w_m \mathbb{E}_m^{\pi}[\cdot]$.

We start with the following lemma on the difference in values in terms of difference in parameters.

**Lemma C.3** *Let $\mathcal{M}_1 = (\mathcal{S}, \mathcal{A}, T_m^1, R_m^1, \nu_m^1)$ and $\mathcal{M}_2 = (\mathcal{S}, \mathcal{A}, T_m^2, R_m^2, \nu_m^2)$ be two latent MDPs with different transition, reward and initial distributions. Then for any history-dependent policy $\pi$,*

$$|V_{\mathcal{M}_1}^{\pi} - V_{\mathcal{M}_2}^{\pi}| \leq H \cdot \mathbb{E}_{\pi,\mathcal{M}_2} \left[ \|(\mu_m^1 - \mu_m^2)(s)\|_1 \right] + \sum_{t=1}^{H} \mathbb{E}_{\pi,\mathcal{M}_2} \left[ |\bar{R}_m^1(s_t,a_t) - \bar{R}_m^2(s_t,a_t)| \right]$$

$$+ H \cdot \sum_{t=1}^{H} \mathbb{E}_{\pi,\mathcal{M}_2} \left[ \|(\mathbb{P}_m^1 - \mathbb{P}_m^2)(s',r|s_t,a_t)\|_1 \right]. \tag{8}$$

The proof of Lemma C.3 is proven in C.4.1.

Equipped with Lemma 3.2 and C.3, we now can prove the main theorem. We first define a few new notations. Let $\#_k(m,s,a)$ be a count of visiting $(s,a)$ in the $m^{th}$ MDP by running a policy $\pi_k$ chosen at the $k^{th}$ episode. Let $N_m^k(s,a)$ be the total number of visit at $(s,a)$ in the $m^{th}$ MDP before the beginning of $k^{th}$ episode, *i.e.*, $N_m^k(s,a) = \sum_{k'=1}^{k-1} \#_{k'}(m,s,a)$. Let $\mathcal{F}_k$ be the filteration of events after running $k$ episodes. Let $\tilde{V}_k^{\pi}$ the value of the optimistic model chosen at the $k^{th}$ episode with a policy $\pi$. Let $\pi^*$ be the optimal policy for the true LMDP $\mathcal{M}^*$. Finally, let us denote $(\cdot)^k$ for the model parameter in the optimistic model at $k^{th}$ episode.

The expected reward $\widetilde{R}_m(s,a)$ in optimistic model is equivalent to $\tilde{R}_m^{hid}(s,a) + \mathbb{E}_{r \sim \tilde{R}_m^{obs}(\cdot|s,a)}[r]$. Using the Lemma C.3, the total regret can be rephrased as the following:

$$\sum_{k=1}^{K} V_{\mathcal{M}^*}^{\pi^*} - V_{\mathcal{M}^*}^{\pi_k} \leq \sum_{k=1}^{K} V_{\widetilde{\mathcal{M}}_k}^{\pi^*} - V_{\mathcal{M}^*}^{\pi_k} \leq \sum_{k=1}^{K} V_{\widetilde{\mathcal{M}}_k}^{\pi_k} - V_{\mathcal{M}^*}^{\pi_k}$$

$$\leq \sum_{k=1}^{K} \sum_{(m,s,a)} \mathbb{E}_{\pi_k,\mathcal{M}^*}[\#_k(m,s,a)] \cdot \left( H \cdot \|(\tilde{\mathbb{P}}_m^k - \mathbb{P}_m)(s',r|s,a)\|_1 \right.$$

$$+ \left| \tilde{R}_m^{hid,k}(s,a) \right| + \left| \mathbb{E}_{r \sim \tilde{R}_m^{obs,k}(r|s,a)}[r] - \mathbb{E}_{r \sim R_m(r|s,a)}[r] \right| \Bigg)$$

$$+ H \cdot \sum_{k=1}^{K} \sum_m \left( \|(\tilde{\mu}_m^k - \mu_m^*)(s)\|_1 \mathbb{E}_{\mathcal{M}^*}[\#_k(m)] + \sqrt{c_\nu/N^k(m)} \mathbb{E}_{\mathcal{M}^*}[\#_k(m)] \right).$$

Note that $\tilde{R}_m^{hid,k}(s,a) = H \min \left( 1, 5\sqrt{(c_T + c_R)/N_m^k(s,a)} \right) \geq H \|(\tilde{\mathbb{P}}_m^k - \mathbb{P}_m)(s',r|s,a)\|_1$, and this is the dominating term. Therefore, the upper bounding equation can be reduced to

$$\sum_{k=1}^{K} V_{\widetilde{\mathcal{M}}_k}^{\pi_k} - V_{\mathcal{M}^*}^{\pi_k} \leq 3H \sum_{k=1}^{K} \sum_{(m,s,a)} \left( 5\sqrt{(c_T + c_R)/N_m^k(s,a)} \mathbb{E}_{\pi_k,\mathcal{M}^*}[\#_k(m,s,a)] \right)$$

$$+ 2H \sum_{k=1}^{K} \sum_{m} \left( \sqrt{c_\nu / N^k(m)} \mathbb{E}_{\mathcal{M}^*}[\#_k(m)] \right).$$

Observe that the expected value of $N_m^k(s,a)$ is $\sum_{k'=1}^{k-1} \mathbb{E}_{\pi_{k'}, \mathcal{M}^*}[\#_{k'}(m,s,a)]$. Let this quantity $\mathbb{E}[N_m^k]$. We can check that

$$Var\left(\#_k(m,s,a) | \mathcal{F}_{k-1}\right) \le H \mathbb{E}_{\pi_k, \mathcal{M}^*}[\#_k(m,s,a)].$$

From the Bernstein's inequality for martingales, for any $(s,a)$ (ignoring constants),

$$N_m^k(s,a) \ge \mathbb{E}[N_m^k(s,a)] - c_1 \sqrt{H \mathbb{E}[N_m^k(s,a)] \log(MSAK/\eta)} - c_2 H \log(MSAK/\eta),$$

for some absolute constants $c_1, c_2 > 0$ and for all $k$ and $(m,s,a)$, with probability at least $1 - \eta$. From this, we can show that

$$H \sum_{k=1}^{K} \sum_{(k,s,a)} \sqrt{(c_T + c_R)/N_m^k(s,a)} \mathbb{E}[\#_k(m,s,a)]$$

$$\le H \sum_{(k,s,a)} \left( \sum_{k=1}^{k_0} \mathbb{E}[\#_k(m,s,a)] + \sum_{k=k_0+1}^{K} \sqrt{(c_T + c_R)/N_m^k(s,a)} \mathbb{E}[\#_k(m,s,a)] \right)$$

$$\lesssim H \sum_{(m,s,a)} \left( H \log(MSAK/\eta) + 2 \sum_{k=k_0+1}^{K} \sqrt{(c_T + c_R)/\mathbb{E}[N_m^k(s,a)]} \mathbb{E}[\#_k(m,s,a)] \right),$$

where $k_0$ is a threshold point where the expected number of visit at $(m,s,a)$ exceeds $4H \log(MSAK/\eta)$. Note that after this point we can assume, with high probability, that $N_m^k(s,a) \ge \mathbb{E}[N_m^k(s,a)]/4$. To bound the summation of the remaining term, for a fixed $(m,s,a)$, we denote $X_k = \mathbb{E}[N_m^k(s,a)]/H$ and $x_k = \mathbb{E}[\#_k(m,s,a)]/H$. Note that $X_{k+1} = X_k + x_k$ and $x_k \le 1$. Then,

$$\sum_{k=k_0+1}^{K} \sqrt{1/X_k} x_k \le \int_{X_{k_0}}^{X_K} \sqrt{\frac{1}{x-1}} dx \le 2\sqrt{X_K}.$$

Plugging this equation, we bound the remaining terms:

$$H \sum_{(m,s,a)} \left( H \log(MSAK/\eta) + 2 \sum_{k=k_0+1}^{K} \sqrt{(c_T + c_R)/\mathbb{E}[N_m^k(s,a)]} \mathbb{E}[\#_k(m,s,a)] \right),$$

$$\le H^2 MSA \log(MSAK/\eta) + 4H \sum_{(m,s,a)} \sqrt{(c_T + c_R) N_m^K(s,a)}$$

$$\le H^2 MSA \log(MSAK/\eta) + 4H \sqrt{(c_T + c_R) HMSAK},$$

where in the last step, we used Cauchy-Schwartz inequality with $\sum_{(m,s,a)} N_m^K(s,a) = HK$. Similarly, we can show that

$$H \sum_{k=1}^{K} \sum_{m} \sqrt{c_\nu/N^k(m)} \mathbb{E}[\#_k(m)] \lesssim HM \log(MSAK/\eta) + 4H\sqrt{c_\nu MK}.$$

Our choice of confidence parameters $c_T$ for a transition probability is $c_T = O(S \log(MSAK/\eta))$, and this is the dominating factor. Thus, the total regret is dominated by

$$H\sqrt{c_T HMSAK} \lesssim HS\sqrt{MAN \log(MSAK/\eta)},$$

which in turn gives a total regret bound of $O\left( HS\sqrt{MAN \log(MSAN/\eta)} \right)$ where $N = HK$.

### C.4.1 Proof of Lemma C.3

*Proof.* We first observe that

$$V_{\mathcal{M}_1}^{\pi} - V_{\mathcal{M}^2}^{\pi} = \sum_{m=1}^{M} w_m \left( \mathbb{E}_m^{1,\pi} \left[ \sum_{t=1}^{H} r_t \right] - \mathbb{E}_m^{2,\pi} \left[ \sum_{t=1}^{H} r_t \right] \right)$$

$$= \sum_{m=1}^{M} w_m \sum_{t=1}^{H} \left( \sum_{(s_1,a_1,r_1,...,s_t,a_t,r_t)} r_t \mathbb{P}_m^{1,\pi}(s_1,...,r_t) - r_t \mathbb{P}_m^{2,\pi}(s_1,...,r_t) \right),$$

where $\mathbb{P}_m^{1,\pi}(s_1,a_1,r_1,...,r_{t-1},s_t) := \nu_m^p(s_1)\Pi_{i=1}^{t-1}\pi(a_i|s_1,...,r_{i-1},s_i)T_m(s_{i+1}|s_i,a_i)R_m(r_i|s_i,a_i)$. We decompose the main difference as

$$\sum_{(s,a,r)_{1:t}} r_t(\mathbb{P}_m^{1,\pi}((s,a,r)_{1:t}) - \mathbb{P}_m^{2,\pi}((s,a,r)_{1:t}))$$

$$= \sum_{((s,a,r)_{1:t})} r_t(R_m^1(r_t|s_t,a_t) - R_m^2(r_t|s_t,a_t))\mathbb{P}_m^{1,\pi}((s,a,r)_{1:t-1},s_t,a_t)$$

$$+ \sum_{(s,a,r)_{1:t}} r_t R_m(r_t|s_t,a_t)(\mathbb{P}_m^{1,\pi} - \mathbb{P}_m^{2,\pi})((s,a,r)_{1:t-1},s_t,a_t)$$

$$\leq \sum_{s_t,a_t} \left| \mathbb{E}_{r_t \sim R_m^1(\cdot|s_t,a_t)}[r_t] - \mathbb{E}_{r_t \sim R_m^2(\cdot|s_t,a_t)}[r_t] \right| \mathbb{P}_m^{2,\pi}(s_t,a_t)$$

$$+ \|(\mathbb{P}_m^{1,\pi} - \mathbb{P}_m^{2,\pi})((s,a,r)_{1:t-1},s_t,a_t)\|_1.$$

Now we bound the total variation distance of the length $t$ histories. For notational convenience, let us denote $|\mathbb{P}_1 - \mathbb{P}_2|(\cdot) = |\mathbb{P}_1(\cdot) - \mathbb{P}_2(\cdot)|$ for any probability measures $\mathbb{P}_1, \mathbb{P}_2$. Then,

$$\sum_{(s,a,r)_{1:t-1},s_t,a_t} |\mathbb{P}_m^{1,\pi} - \mathbb{P}_m^{2,\pi}|((s,a,r)_{1:t-1},s_t,a_t)$$

$$= \sum_{(s,a,r)_{1:t-1},s_t} |\mathbb{P}_m^{1,\pi} - \mathbb{P}_m^{2,\pi}|((s,a,r)_{1:t-1},s_t) \sum_{a_t} \pi(a_t|(s,a,r)_{1:t-1},s_t)$$

$$= \sum_{(s,a,r)_{1:t-1},s_t} |\mathbb{P}_m^{1,\pi} - \mathbb{P}_m^{2,\pi}|((s,a,r)_{1,t-1},s_t)$$

$$\leq \sum_{(s,a,r)_{1:t-1},s_t} |\mathbb{P}_m^{1,\pi} - \mathbb{P}_m^{2,\pi}|((s,a,r)_{1:t-2},s_{t-1},a_{t-1})\mathbb{P}_m^1(s_t,r_{t-1}|s_{t-1},a_{t-1})$$

$$+ \sum_{(s,a,r)_{1:t-1},s_t} \mathbb{P}_m^{2,\pi}((s,a,r)_{1:t-2},s_{t-1},a_{t-1})|\mathbb{P}_m^1 - \mathbb{P}_m^2|(s_t,r_{t-1}|s_{t-1},a_{t-1})$$

$$\leq \sum_{(s,a,r)_{1:t-2},s_{t-1},a_{t-1}} |\mathbb{P}_m^{1,\pi} - \mathbb{P}_m^{2,\pi}|((s,a,r)_{1:t-2},s_{t-1},a_{t-1})$$

$$+ \sum_{(s,a,r)_{1:t-2},s_{t-1},a_{t-1}} \|(\mathbb{P}_m^{1,\pi} - \mathbb{P}_m^{2,\pi})(s_t,r_{t-1}|s_{t-1},a_{t-1})\|_1 \mathbb{P}_m^{2,\pi}((s,a,r)_{1:t-2},s_{t-1},a_{t-1})$$

$$= \|(\mathbb{P}_m^{1,\pi} - \mathbb{P}_m^{2,\pi})((s,a,r)_{1:t-2},s_{t-1},a_{t-1})\|_1$$

$$+ \sum_{s_{t-1},a_{t-1}} \|(\mathbb{P}_m^{1,\pi} - \mathbb{P}_m^{2,\pi})(s_t,r_{t-1}|s_{t-1},a_{t-1})\|_1 \mathbb{P}_m^{2,\pi}(s_{t-1},a_{t-1}).$$

We can apply the same expansion recursively to bound total variation for length $t-1$ histories. Now plug this relation to the regret bound, we have

$$|V_{\mathcal{M}_1}^{\pi} - V_{\mathcal{M}_2}^{\pi}| \leq \sum_{m=1}^{M} w_m \sum_{(s,a)} \sum_{t=1}^{H} \left| \mathbb{E}_{r \sim R_m^1(\cdot|s,a)}[r] - \mathbb{E}_{r \sim R_m^2(\cdot|s,a)}[r] \right| \mathbb{P}_m^{2,\pi}(s_t = s, a_t = a)$$

$$+ \sum_{m=1}^{M} w_m \sum_{t=1}^{H} \left( \sum_s |\nu_m^1(s) - \nu_m^2(s)|\mathbb{P}_m^2(s_1 = s) \right)$$

$$+ \sum_{(s,a)} \sum_{t'=1}^{t} \|(\mathbb{P}_m^{1,\pi} - \mathbb{P}_m^{2,\pi})(s',r|s,a)\|_1 \mathbb{P}_m^{2,\pi}(s_{t'} = s, a_{t'} = a) \Bigg)$$

$$\leq \sum_{m=1}^{M} w_m \sum_{t=1}^{H} \left( \mathbb{E}_m^{2,\pi} \left[ |\bar{R}_m^1(s_t, a_t) - \bar{R}_m^2(s_t, a_t)| \right] \right)$$

$$+ H \cdot \sum_{m=1}^{M} w_m \left( \|(\mu_m^1 - \mu_m^2)(s)\|_1 + \sum_{t=1}^{H} \mathbb{E}_m^{2,\pi} \left[ \|(\mathbb{P}_m^1 - \mathbb{P}_m^2)(s',r|s_t,a_t)\|_1 \right] \right),$$

giving the equation (8) as claimed. $\qquad\square$

## Appendix D  Learning with Separation and Good Initialization

### D.1  Well-Separated Condition for MDPs

In this subsection, we formalize a condition for *clusterable* mixtures of MDPs: the overlap of trajectories from different MDPs should be small in order to correctly infer the true contexts from sampled trajectories. We call the underlying MDPs *well-separated* if they satisfy the following separation condition:

**Condition 3 (Well-Separated MDPs)** *If a trajectory $\tau$ of length $H$ is sampled from MDP $M_{m^*}$ by running any policy $\pi \in \Pi$, we have*

$$\mathbb{P}_{\tau \sim \mathcal{M}_{m^*}, \pi} \left( \frac{\mathbb{P}_{\tau \sim \mathcal{M}_m, \pi}(\tau)}{\mathbb{P}_{\tau \sim \mathcal{M}_{m^*}, \pi}(\tau)} > (\epsilon_p/M)^{c_1} \right) < (\epsilon_p/M)^{c_2} \qquad \forall m \neq m^*. \tag{9}$$

*for a target failure probability $\epsilon_p > 0$ where $c_1, c_2 \geq 4$ are some universal constants.*

Here, $\mathbb{P}_{\tau \sim \mathcal{M}_m, \pi}$ is a probability of getting a trajectory from the context $m$ with policy $\pi$. One *sufficient* condition that ensures the well-separated condition (9) is Assumption 1 as guaranteed by the following lemma:

**Lemma D.1** *Under the Assumption 1 with a constant $\delta = \Theta(1)$, if the time horizon is sufficiently long such that $H > C \cdot \delta^{-4} \log^2(1/\alpha) \log(M/\epsilon_p)$ for some absolute constant $C > 0$ and $\alpha = \delta^2/(200S)$, then the well-separated condition (9) holds true with $c_1, c_2 \geq 4$.*

Proof of Lemma D.1 is given in Appendix D.2. We remark here that we have not optimized the requirement on the time horizon $H$ to satisfy Condition 3, and we conjecture it can be improved. We also mention here that the required time-horizon can be much shorter if the KL-divergence between distributions is larger, even though the $l_1$ distance remains the same. Finally, we remark that Assumption 1 is only a sufficient condition, and can be relaxed as long as Condition 3 is satisfied.

**Remark 1 (Logarithmic dependency of $H$ on $1/\epsilon_p$)** *We note here that the convergence guarantee for the online EM might be extended to allow some small probability of wrong inference of contexts. In that case, we do not require $H$ to scale logarithmically with the inverse of a failure probability. It would be an analogous to the local convergence guarantee in a mixture of well-separated Gaussian distributions [28, 27]. The situation is even more complicated since we may run a possibly different policy in each episode. It would be an interesting question whether the online EM implementation would eventually get some good converged policy and model parameters in more general settings.*

### D.2  Proof of Lemma D.1

In this proof, we assume all probabilistic event is taken with true context $m^*$: unless specified, we assume $\mathbb{P}(\cdot)$ and $\mathbb{E}[\cdot]$ are measured with context $m^*$.

Suppose a trajectory $\tau$ is obtained from MDP $M_{k^*}$. Let us denote the probability of getting $\tau$ from $m^{th}$ MDP by running policy $\pi$ as $\mathbb{P}_{\tau \sim M_m, \pi}(\tau) = \mathbb{P}_m(\tau)$. It is enough to show that

$$\ln \left( \frac{\mathbb{P}_{m^*}(\tau)}{\mathbb{P}_m(\tau)} \right) > c \ln(M/\epsilon_p), \quad \forall m \neq m^*,$$

with probability $1 - (\epsilon_p/M)^4$. Note that for any history-dependent policy $\pi$,

$$\ln\left(\frac{\mathbb{P}_{m^*}(\tau)}{\mathbb{P}_m(\tau)}\right) = \sum_{t=1}^{H}\ln\left(\frac{\mathbb{P}_{m^*}(s_{t+1}, r_t|s_t, a_t)}{\mathbb{P}_m(s_{t+1}, r_t|s_t, a_t)}\right).$$

For simplicity, let us compactly denote $(s', r)$ as $o$, and $(s_{t+1}, r_t)$ as $o_t$. Note that in general, $\ln\left(\frac{\mathbb{P}_{m^*}(\tau)}{\mathbb{P}_m(\tau)}\right)$ can be unbounded due to zero probability assignments. Thus we consider a relaxed MDP that assigns non-zero probability to all observations. Let $\alpha > 0$ be sufficiently small such that $\alpha \ln(1/\alpha) < \delta^2/(200S)$. We define similar probability distributions such that $\hat{\mathbb{P}}_m$

$$\hat{\mathbb{P}}_m(o|s, a) = \alpha + (1 - 2\alpha S)\mathbb{P}_m(o|s, a).$$

We split the original target into three terms and bound each of them:

$$\ln\left(\frac{\mathbb{P}_{m^*}(\tau)}{\mathbb{P}_m(\tau)}\right) = \ln\left(\frac{\mathbb{P}_{m^*}(\tau)}{\hat{\mathbb{P}}_m(\tau)}\right) + \ln\left(\frac{\hat{\mathbb{P}}_m(\tau)}{\mathbb{P}_m(\tau)}\right).$$

Note that $\|\mathbb{P}_m - \hat{\mathbb{P}}_m(o|s, a)\|_1 \leq 4S\alpha$. For the first term, we investigate the expectation of this quantity first:

$$\mathbb{E}\left[\sum_{t=1}^{H}\ln\left(\frac{\mathbb{P}_{m^*}(o_t|s_t, a_t)}{\hat{\mathbb{P}}_m(o_t|s_t, a_t)}\right)\right] = \mathbb{E}\left[\sum_{t=1}^{H}\mathbb{E}\left[\ln\left(\frac{\mathbb{P}_{m^*}(o_t|s_t, a_t)}{\hat{\mathbb{P}}_m(o_t|s_t, a_t)}\right)\Bigg|s_1, a_1, r_1, s_2, ..., r_{t-1}, s_t, a_t\right]\right]$$

$$= \mathbb{E}\left[\sum_{t=1}^{H}\sum_{o_t}\mathbb{P}_{m^*}(o_t|s_t, a_t)\ln\left(\frac{\mathbb{P}_{m^*}(o_t|s_t, a_t)}{\hat{\mathbb{P}}_m(o_t|s_t, a_t)}\right)\right]$$

$$= \mathbb{E}\left[\sum_{t=1}^{H}D_{KL}(\mathbb{P}_{m^*}(o_t|s_t, a_t), \hat{\mathbb{P}}_m(o_t|s_t, a_t))\right] \geq H\delta^2,$$

where in the last step we applied Pinsker's inequality.

Now we want to apply Chernoff-type concentration inequalities for martingales. We need the following lemma on a sub-exponential property of $\mathbb{P}(X)$ on a general random variable $X$:

**Lemma D.2** *Suppose $X$ is arbitrary discrete random variable on a finite support $\mathcal{X}$. Then, $\ln(1/\mathbb{P}(X))$ is a sub-exponential random variable [48] with Orcliz norm $\|\ln(1/\mathbb{P}(X))\|_{\psi_1} = 1/e$.*

*Proof.* Following the definition of sub-exponential norm [48], we find $\|\ln(1/\mathbb{P}(X))\|_{\psi_1} = O(1)$:

$$\|\ln(1/\mathbb{P}(X))\|_{\psi_1} = \sup_{q\geq 1} q^{-1}\mathbb{E}_X[\ln^q(1/\mathbb{P}(X))]^{1/q}$$

$$= \sup_{q\geq 1} q^{-1}\left(\sum_{X\in\mathcal{X}}\mathbb{P}(X)\ln^q(1/\mathbb{P}(X))\right)^{1/q}.$$

For any $q \geq 1$, let us first find maximum value of $p\ln^q(1/p)$ for $0 \leq p \leq 1$. Taking a log and finding a derivative with respect to $p$ yields

$$\frac{1}{p} + q\frac{(-1/p)}{\ln(1/p)} = \frac{1}{p}(1 - q/\ln(1/p)).$$

Hence $p\ln^q(1/p)$ takes a maximum at $p = e^{-q}$ with value $(q/e)^q$. This gives a bound for sub-exponential norm:

$$\|\ln(1/\mathbb{P}(X))\|_{\psi_1} = \sup_{q\geq 1} q^{-1}\left(\sum_{X\in\mathcal{X}}\mathbb{P}(X)\ln^q(1/\mathbb{P}(X))\right)^{1/q}$$

$$\leq \sup_{q\geq 1} q^{-1}(q/e) = 1/e.$$

$\square$

With the above Lemma and the sum of sub-exponential martingales, it is easy to verify (see Proposition 5.16 in [48]) that

$$\mathbb{P}\left(\ln\left(\mathbb{P}_{m^*}(\tau)\right) \leq \mathbb{E}\left[\ln\left(\mathbb{P}_{m^*}(\tau)\right)\right] - H\epsilon_1\right) \leq \exp\left(-c \cdot \min(\epsilon_1, \epsilon_1^2)H\right),$$

where $c > 0$ is some absolute constant, since $\ln(\mathbb{P}_{m^*}(\tau)) = \sum_{t=1}^{H} \ln(\mathbb{P}_{m^*}(o_t|s_t, a_t))$ is a sum of $H$ sub-exponential martingales. We can also apply Azuma-Hoeffeding's inequality to control the statistical deviation in $\ln(\hat{\mathbb{P}}_m(\tau))$:

$$\mathbb{P}\left(\ln\left(\hat{\mathbb{P}}_m(\tau)\right) \geq \mathbb{E}\left[\ln\left(\hat{\mathbb{P}}_m(\tau)\right)\right] + H\epsilon_2\right) \leq \exp\left(-\frac{H\epsilon_2^2}{2\log^2(1/\alpha)}\right),$$

since $\hat{\mathbb{P}}_m(\tau)$ is bounded by $\ln(1/\alpha)$.

Now let $\epsilon = \epsilon_1 + \epsilon_2 = c_2 \cdot \log(1/\alpha)\sqrt{2\log(M/\epsilon_p)/H}$ for some absolute constant $c_2 > 0$. If the time horizon $H \geq C_0\delta^{-4}\log^2(1/\alpha)\log(M/\epsilon_p)$ for some sufficiently large constant $C_0 > 0$, then a simple algebra shows that

$$\ln\left(\frac{\mathbb{P}_{m^*}(\tau)}{\hat{\mathbb{P}}_m(\tau)}\right) \geq H\delta^2 - H\epsilon \geq H\delta^2/2,$$

with probability at least $1 - (\epsilon_p/M)^5$.

Finally, we bound extra terms caused by using approximated probabilities. We note that

$$\ln\left(\frac{\hat{\mathbb{P}}_m(o|s, a)}{\mathbb{P}_m(o|s, a)}\right) \geq -4\alpha S, \qquad \forall(o, s, a),$$

given $2\alpha S$ is sufficiently small. Therefore for any trajectory, we have $\ln\left(\hat{\mathbb{P}}_m(\tau)/\mathbb{P}_m(\tau)\right) \geq -4\alpha SH \geq -H\delta^2/4$. Thus we have $\ln\left(\mathbb{P}_{m^*}(\tau)/\mathbb{P}_m(\tau)\right) \geq H\delta^2/4 \geq 4\log(M/\epsilon_p)$ with probability at least $1 - (\epsilon_p/M)^5$, which satisfies Condition 3.

### D.3   Proof of Theorem 3.4

The key component is the following lemma on the correct estimation of belief in contexts.

**Lemma D.3** *Let a trajectory is sampled from $m^{*th}$ MDP. Under the Assumption 1 with good initialization $\epsilon_{init} < \delta^2/(200\ln(1/\alpha))$ in (3) and $H > C \cdot \delta^{-4}\log^2(1/\alpha)\log(N/\eta)$ for some universal constant $C > 0$, we have*

$$\hat{b}(m^*) \geq 1 - (N/\eta)^{-4},$$

*with probability at least $1 - (N/\eta)^{-4}$.*

Since we have estimated belief is almost approximately correct for $O(N)$ episodes with $\epsilon_p = O(1/N)$, we now have the confidence intervals for transition matrices and rewards:

**Corollary 1** *With probability at least $1 - 1/N$, for all round of episodes, we have*

$$\|(\hat{T}_m - T_m^*)(s'|s, a)\|_1 \leq \sqrt{c_T/N_m(s, a)} + 1/N^3,$$
$$\|(\hat{R}_m - R_m^*)(r|s, a)\|_1 \leq \sqrt{c_R/N_m(s, a)} + 1/N^3,$$
$$\|(\hat{\nu}_m - \nu_m^*)(s)\|_1 \leq \sqrt{c_\nu/N_m(s)} + 1/N^3.$$

*for all $s, a, r, s'$.*

The corollary is straight-forward since the estimation error accumulated from errors in beliefs throughout $K$ episodes is at most $1/N^3$. If we build an optimistic model with the estimated parameters as in Lemma 3.2, the optimistic value with any policy for the model satisfies

$$V_{\widetilde{\mathcal{M}}}^{\pi} \geq V_{\mathcal{M}^*}^{\pi} - H^2/N^2. \tag{10}$$

Equation (10) is a consequence of Lemma 3.2 and LMDP version of sensitivity analysis in partially observable environments [40], which can also be inferred from C.3. Following the same argument in the proof of Theorem 3.3, we can also show that the estimated visit counts at $(s, a)$ is at least

$$N_m(s, a) \geq \mathbb{E}[N_m(s, a)] - c_1 \sqrt{H\mathbb{E}[N_m(s, a)] \log(MSAK/\eta)} - c_2 H \log(MSAK/\eta) - 1/N^2,$$

for some absolute constants $c_1, c_2 > 0$ for all $(s, a)$, with probability at least $1 - \eta$. The additional regret caused by small errors in belief estimates is therefore bounded by

$$SH^2/N^2 * N + H^2 MSA/N^2 \leq 1/N,$$

assuming $N = HK \gg H^2 S^2 MA$. The remaining steps are equivalent to the proof of Theorem 3.3.

## D.4  Proof of Lemma D.3

*Proof.*  The proof for Lemma D.3 is an easy replication of the proof for Lemma D.1. We show that

$$\sum_{t=1}^{H} \ln\left(\frac{\alpha + (1 - 2\alpha S)\hat{P}_{m^*}(o_t|s_t, a_t)}{\alpha + (1 - 2\alpha S)\hat{P}_m(o_t|s_t, a_t)}\right) \geq 8 \log(N/\eta), \tag{11}$$

with probability at least $1 - (N/\eta)^{-4}$ for all $m^* \neq m$.

Let $Q_m = \alpha + (1 - 2\alpha S)\hat{P}_m$ for all $m$. Note that $\|Q_m - Q_{m^*}\|_1 \geq \delta/2$ due to the initialization condition. Furthermore, $|\ln(Q_m(o|s, a)/Q_{m^*}(o|s, a))| \leq \ln(1/\alpha)$. Hence we can apply Azuma-Hoeffeding's inequality to get

$$\sum_{t=1}^{H} \ln\left(\frac{Q_{m^*}(o_t|s_t, a_t)}{Q_m(o_t|s_t, a_t)}\right) \geq \mathbb{E}\left[\sum_{t=1}^{H} \ln\left(\frac{Q_{m^*}(o_t|s_t, a_t)}{Q_m(o_t|s_t, a_t)}\right)\right] - \ln(1/\alpha)\sqrt{H \log(N/\eta)}$$

with probability at least $1 - (MN)^{-4}$. To lower bound the expectation, we can proceed as before:

$$\mathbb{E}\left[\sum_{t=1}^{H} \ln\left(\frac{Q_{m^*}(o_t|s_t, a_t)}{Q_m(o_t|s_t, a_t)}\right)\right] = \mathbb{E}\left[\sum_{t=1}^{H} \sum_{o_t} P_{m^*}(o_t|s_t, a_t) \ln\left(\frac{Q_{m^*}(o_t|s_t, a_t)}{Q_m(o_t|s_t, a_t)}\right)\right]$$

$$= \mathbb{E}\left[\sum_{t=1}^{H} \sum_{o_t} Q_{m^*}(o_t|s_t, a_t) \ln\left(\frac{Q_{m^*}(o_t|s_t, a_t)}{Q_m(o_t|s_t, a_t)}\right)\right]$$

$$+ \mathbb{E}\left[\sum_{t=1}^{H} \sum_{o_t} (P_{m^*} - Q_{m^*})(o_t|s_t, a_t) \ln\left(\frac{Q_{m^*}(o_t|s_t, a_t)}{Q_m(o_t|s_t, a_t)}\right)\right]$$

$$\geq \mathbb{E}\left[\sum_{t=1}^{H} D_{KL}(Q_{m^*}(o_t|s_t, a_t), Q_m(o_t|s_t, a_t))\right] - \mathbb{E}\left[\sum_{t=1}^{H} \|P_{m^*} - Q_{m^*}\|_1\right] \ln(1/\alpha)$$

$$\geq H\delta^2/4 - H(2\alpha S + \epsilon_{init})\ln(1/\alpha).$$

As long as $2\alpha S \ln(1/\alpha) \leq \delta^2/200$ and $\epsilon_{init} \ln(1/\alpha) \leq \delta^2/200$, we have

$$\mathbb{E}\left[\sum_{t=1}^{H} \ln\left(\frac{Q_{m^*}(o_t|s_t, a_t)}{Q_m(o_t|s_t, a_t)}\right)\right] \geq H\delta^2/8.$$

If $H \geq C\delta^{-4}\ln(1/\alpha)^2 \log(N/\eta)$ for sufficiently large constant $C > 0$, (11) holds with probability at least $1 - (N/\eta)^{-4}$. The implication of lemma is:

$$\hat{b}(k^*) \geq 1 - (N/\eta)^{-8} \cdot M \geq 1 - (N/\eta)^{-4},$$

which proves the claimed lemma.  □

# Appendix E  Algorithm Details for Initialization

## E.1  Spectral Learning of PSRs

In this subsection, we implement a spectral algorithm to learn PSR in detail. Recall that we define $P_{\mathcal{T},\mathcal{H}_s} = L_s H_s$ in Condition 1, 2 such that

$$(P_{\mathcal{T},\mathcal{H}_s})_{i,j} = \mathbb{P}^\pi(\tau_i, h_{s,j}) = (L_s)_{i,:}(H_s)_{(:,j)}.$$

where $P_{\mathcal{T},\mathcal{H}_s} \in \mathbb{R}^{|\mathcal{T}| \times |\mathcal{H}_s|}$ is a matrix of joint probabilities of tests and histories ending with $s$. Let the top-$k$ left and right singular vectors of $P_{\mathcal{T},\mathcal{H}_s}$ be $U_s$ and $V_s$ respectively. Note that with the rank conditions, $U_s^\top P_{\mathcal{T}_s,\mathcal{H}_s} V_s$ is invertible. We also consider a matrix of joint probabilities of histories, intermediate action-reward-next-state pairs, and tests $P_{\mathcal{T},(s',r)a,\mathcal{H}_s} = L_{s'} D_{(s',r),a,s} H_s$, where $D_{(s',r),a,s} = diag(\mathbb{P}_1(s',r|a,s),...,\mathbb{P}_M(s',r|a,s))$. For the simplicity in notations, we occasionally replace $(s',r)$ by a single letter $o$. The transformed PSR parameters of the LMDP can be computed by

$$B_{o,a,s} = U_{s'}^\top P_{\mathcal{T},oa,\mathcal{H}_s} V_s (U_s P_{\mathcal{T},\mathcal{H}_s} V_s)^{-1} = (U_{s'}^\top L_{s'}) D_{o,a,s} (U_s^\top L_s)^{-1}.$$

The initial and normalization parameters can be computed as

$$b_{1,s} = U_s^\top \mathbb{P}(\mathcal{T}, s_1 = s) = U_s^\top \mathbb{P}(\mathcal{T}|s)(w \cdot \nu)(s) = (U_s^\top L_s)(w \cdot \nu)(s),$$
$$b_{\infty,s}^\top = P_{\mathcal{H}_s}^\top V_s (U_s^\top P_{\mathcal{T},\mathcal{H}_s} V_s)^{-1},$$

where $P_{\mathcal{H}_s} \in \mathbb{R}^{|\mathcal{H}_s|}$ is a vector of probability of sampling a history in $\mathcal{H}_s$, and $(w \cdot \nu)(s)$ is $M$ dimensional vector with each $m^{th}$ entry $w_m \nu_m(s)$. For the normalization factor, note that $P_{\mathcal{H}_s}^\top = 1^\top H_s$, therefore

$$b_{\infty,s}^\top = 1^\top H_s V_s (U_s^\top P_{\mathcal{T},\mathcal{H}_s} V_s)^{-1} = 1^\top (U_s^\top L_s)^{-1}(U_s^\top L_s H_s V_s)(U_s^\top P_{\mathcal{T},\mathcal{H}_s} V_s)^{-1} = 1^\top (U_s^\top L_s)^{-1}.$$

It is easy to verify that

$$\mathbb{P}((s,a,r)_{1:t}, s_t) = b_{s_t,\infty}^\top B_{o_{t-1},a_{t-1},s_{t-1}}...B_{o_1,a_1,s_1} b_{s_1,1} = 1^\top D_{o_{t-1},a_{t-1},s_{t-1}}...D_{o_1,a_1,s_1}(w \cdot \nu)(s_1).$$

With Assumption 2, we assume that a set of histories and tests $\mathcal{H}, \mathcal{T}$ contain all possible observations of a fixed length $l$. Furthermore, we assume that the short trajectories are collected such that each history is sampled from the sampling policy $\pi$ and then the intervening action sequence for test is uniformly randomly selected. We estimate the joint probability matrices with $N$ short trajectories such that

$$(\hat{P}_{\mathcal{H}_s})_i = \frac{1}{N}\#(h_{s,i}), \quad (\hat{P}_{\mathcal{T},\mathcal{H}_s})_{i,j} = \frac{A^l}{N}\#(\tau_i, h_{s,j}), \quad (\hat{P}_{\mathcal{T},oa,\mathcal{H}_s})_{i,j} = \frac{A^{l+1}}{N}\#(\tau_i, oa, h_{s,j}),$$

where $\#$ means the number of occurrence of the event when we sample histories from the sampling policy $\pi$. For instance, $\#(\tau_i, h_{s,j})$ means the number of occurrence of $j^{th}$ history in $\mathcal{H}_s$ and test resulting in $i^{th}$ test in $\mathcal{T}$. Factors $A^l$ and $A^{l+1}$ are importance sampling weights for intervening actions. The initial PSR states are estimated separately: $(\hat{\mathbb{P}}_{\mathcal{T},s_1=s})_i = \frac{A^l}{N}\#(\tau_i, s_1 = s)$, assuming we get $N$ sample trajectories from the beginning of each episode.

Now let $\hat{U}_s, \hat{V}_s$ be left and right singular vectors of $\hat{P}_{\mathcal{T},\mathcal{H}_s}$. Then the spectral learning algorithm outputs parameters for PSR:

$$\hat{B}_{o,a,s} = \hat{U}_{s'}^\top \hat{P}_{\mathcal{T},oa,\mathcal{H}_s} \hat{V}_s (\hat{U}_s^\top \hat{P}_{\mathcal{T},\mathcal{H}_s} \hat{V}_s)^{-1},$$
$$\hat{b}_{\infty,s}^\top = \hat{P}_{\mathcal{H}_s}^\top \hat{V}_s (\hat{U}_s^\top \hat{P}_{\mathcal{T},\mathcal{H}_s} \hat{V}_s)^{-1},$$
$$\hat{b}_{1,s} = \hat{U}_s^\top \hat{\mathbb{P}}(\mathcal{T}, s_1 = s). \tag{12}$$

Then, the estimated probability of a sequence with any history-dependent policy $\pi$ is given by

$$\hat{\mathbb{P}}^\pi((s,a,r)_{1:t-1}, s_t) = \Pi_{i=1}^{t-1}\pi(a_i|(s,a,r)_{1:i-1}, s_i) \cdot \hat{b}_{\infty,s_t}^\top \hat{B}_{o_{t-1},a_{t-1},s_{t-1}}...\hat{B}_{o_1,a_1,s_1}\hat{b}_{1,s_1}. \tag{13}$$

The update of PSR states and the prediction of next observation is given as the following:

$$\hat{b}_1 = \hat{b}_{1,s_1}, \quad \hat{b}_t = \frac{\hat{B}_{o_{t-1},a_{t-1},s_{t-1}}\hat{b}_{t-1}}{\hat{b}_{\infty,s_t}^\top \hat{B}_{o_{t-1},a_{t-1},s_{t-1}}\hat{b}_{t-1}}, \tag{14}$$

$$\hat{\mathbb{P}}(s', r|(s, a, r)_{1:t-1}, s_t|| \boldsymbol{do}\ a) = \hat{b}_{s',\infty}^\top \hat{B}_{(s',r),a,s_t} \hat{b}_t. \tag{15}$$

From the above procedure, we can establish a formal guarantee on the estimation of probabilities of length $t > 0$ trajectories obtained with *any* history-dependent policies:

**Theorem E.1** *Suppose the LMDP and a set of histories $\mathcal{H}$ and tests $\mathcal{T}$ satisfies Assumption 2. If the number of short trajectories $N = n_0$ satisfies*

$$n_0 \geq C \cdot \frac{MA^{2l+1}}{p_\pi \sigma_\tau^2 \sigma_h^2} \frac{t^2}{\epsilon_t^2} \left(S + \frac{A^l}{\sigma_h^2}\right) \log(SA/\eta),$$

*where $C > 0$ is an universal constant, and $p_\pi = \min_s \mathbb{P}^\pi(\text{end state} = s)$, then for any (history dependent) policy $\pi$, with probability at least $1 - \eta$,*

$$\|(\mathbb{P}^\pi - \hat{\mathbb{P}}^\pi)((s, a, r)_{1:t-1}, s_t)\|_1 \leq \epsilon_t.$$

We mention that the formal finite-sample guarantee of PSR learning only exists for hidden Markov models [20], an extension to LMDPs requires re-derivation of the proof to include the effect of arbitrary decision making policies. For completeness, we provide the proof of Theorem E.1 in Appendix F.1.

As a result of spectral learning of PSR (see a detailed procedure in Appendix E.1), we can provide a key ingredient to cluster longer trajectories to recover the original LMDP model.

**Theorem E.2** *Suppose we have successfully estimated PSR parameters from the spectral learning procedure in Appendix E.1, such that we have the following guarantee on estimated probabilities of trajectories with any history-dependent policy $\pi$:*

$$\|(\mathbb{P}^\pi - \hat{\mathbb{P}}^\pi)((s, a, r)_{1:t-1}, s_t)\|_1 \leq \epsilon_t,$$

*for sufficiently small $\epsilon_t > 0$. Suppose we will execute a policy $\pi$ for $t$ time steps, observe a history $((s, a, r)_{1:t-1}, s_t)$, and then estimate probabilities of all possible future observations (or tests $o_{t:t+l-1}$) with intervening action sequence $a_{t:t+l-1}^\tau$. Then we have the following guarantee on conditional probabilities with target accuracy $\epsilon_c > 0$:*

$$\|(\mathbb{P}^\pi - \hat{\mathbb{P}}^\pi)(o_{t:t+l-1}|(s, a, r)_{1:t-1}, s_t|| \boldsymbol{do}\ a_{t:t+l-1}^\tau)\|_1 \leq 4\epsilon_c,$$

*with probability at least $1 - \epsilon_t/\epsilon_c$.*

## E.2 Clustering with PSR Parameters and Separation

We begin with the high-level idea of the algorithm that works as the following: suppose we have a new trajectory of length $H$ and the last two states are $s_{H-1}, s_H$ from unknown context $m^*$. We first consider true conditional probability given a history of $h = (s, a, o)_{1:H-2}$. Here $H > C_0 \cdot \delta^{-4} \log^2(1/\alpha) \log(N/\eta)$ is the length of episodes which satisfies the required condition for $H$ in Lemma D.1, and $N$ is total number of episodes to be run with L-UCRL (Algorithm 1). Under Condition 3 with a failure probability $\epsilon_p = O(1/N)$, the true belief state over contexts $b$ at time step $O(H)$ satisfies

$$b(m^*) \geq 1 - (\eta/N)^4.$$

With PSR parameters, we can estimate prediction probabilities at time step $H - 1$ for any given histories. This in turn implies that for any intervening actions $a_1^\tau, ..., a_{l'}^\tau$ of length $l'$, the prediction probability given the history of length $H - 1$ is nearly close to the prediction in the $m^{*th}$ MDP:

$$\|(\mathbb{P} - \mathbb{P}_{m^*})(o_1^\tau...o_{l'}^\tau|h|| \boldsymbol{do}\ a_1^\tau...a_{l'}^\tau)\|_1 \leq (\eta/N)^4,$$

with probability at least $1 - (\eta/N)^4$. On the other hand, note that in the $m^{*th}$ MDP, $\mathbb{P}_{m^*}(o_1^\tau...o_{l'}^\tau|h|| \boldsymbol{do}\ a_1^\tau...a_{l'}^\tau) = \mathbb{P}_{m^*}(o_1^\tau...o_{l'}^\tau|s_{H-1}|| \boldsymbol{do}\ a_1^\tau...a_{l'}^\tau)$. Therefore, combining with Theorem E.2, we have that

$$\|(\mathbb{P} - \hat{\mathbb{P}})(o_1^\tau...o_{l'}^\tau|h|| \boldsymbol{do}\ a_1^\tau...a_{l'}^\tau)\|_1 \leq 4\epsilon_c, \qquad \forall a_1^\tau...a_{l'}^\tau \in \mathcal{A}^{l'},$$

**Algorithm 5** Recovery of LMDP parameters
___
**Input:** A set of short histories $\mathcal{H}$ and tests $\mathcal{T}$ for learning PSR, and tests $\mathcal{T}'$ for clustering
1: // Learn PSR parameters up to precision $o(\delta)$
2: Estimate PSR parameters $\{\hat{b}_{1,s}, \hat{b}_{\infty,s}, \hat{B}_{o,a,s}, \forall o, a, s\}$ following (12) in Appendix E.1 up to precision $o(\delta)$
3: // Get clusters $\{\hat{T}_m(\cdot|s,a), \hat{R}_m(\cdot|s,a)\}_{(s,a)\in\mathcal{S}\times\mathcal{A}, m\in[M]}$ with learned PSR parameters
4: Initialize $V_s = \{\}$ for all $s \in \mathcal{S}$
5: **for** $n_1/3$ episodes **do**
6:     Play exploration policy $\pi$ and get a trajectory $h = (s_1, a_1, r_1, ..., s_H, a_H, r_H)$
7:     Get PSR state $\hat{b}_{H-1}$ at time step $H-1$ using equation (14)
8:     Compute $p_{H-1}(\mathcal{T}') = \hat{\mathbb{P}}(\mathcal{T}'|(s,a,r)_{1:H-2}, s_{H-1})$ using equation (15)
9:     Add $p_{H-1}$ in $V_{s_{H-1}}$
10: **end for**
11: **for** all $s \in \mathcal{S}$ **do**
12:     Find $M$-cluster centers $C_s$ that cover all points in $V_s$ (*e.g.*, with $k$-means++ [3])
13: **end for**
14: // Build each MDP model by correctly assigning contexts to estimated transition and reward probabilities
15: **for** $n_1/3$ episodes **do**
16:     Play exploration policy $\pi$ until time-step $H-1$ and get a PSR state $\hat{b}_{H-1}$ at time step $H-1$
17:     Play an uniformly sampled action $a$ and get a PSR state $\hat{b}_H$ at time step $H$
18:     Compute $p_{H-1}(\mathcal{T}'), p_H(\mathcal{T}')$
19:     Find centers (labels) $c_{H-1} \in C_{s_{H-1}}$ and $c_H \in C_{s_H}$ such that $c_{H-1}$ and $c_H$ are the closest to $p_{H-1}$ and $p_H$ respectively.
20:     If $s_{H-1}$ and $s_H$ are different, let two centers $c_{H-1}, c_H$ be in the same context
21: **end for**
22: If reordering of contexts are inconsistent, return FAIL
23: Otherwise, construct $\hat{T}_m$ and $\hat{R}_m$ from cluster centers $\{C_s\}_{s\in\mathcal{S}}$
24: **for** $n_1/3$ episodes **do**
25:     Play exploration policy $\pi$ and get a PSR state $\hat{b}_H$ at time step $H$
26:     Compute $p_H(\mathcal{T}')$ and find centers $c_H \in C_{s_H}$ that is closest to $p_H(\mathcal{T}')$
27:     Get the context $m$ where $c_H$ belongs to, and update initial state distribution $\hat{\nu}_m$ of $m^{th}$ MDP
28: **end for**
___

with probability at least $1 - A^{l'}\epsilon_t/\epsilon_c$. In other words, the prediction probability estimated with PSR parameters are almost correct within error $(\eta/N)^4 + 4\epsilon_c$ with probability at least $1 - A^{l'}\epsilon_t/\epsilon_c$.

In a slightly more general context, let $\mathcal{T}'$ be a set of all tests of length $l'$ with all possible intervening $A^{l'}$ action sequences where $1 \le l' \le l$. The core idea of clustering is to have the error in prediction probability $\epsilon_c$ smaller than the separation of prediction probabilities between different MDPs. Let $\delta_{psr}$ be the average $l_1$ distance between predictions of all length $l'$ tests such that:

$$\sum_{a_1^\tau...a_{l'}^\tau \in \mathcal{A}^{l'}} \|(\mathbb{P}_{m_1} - \mathbb{P}_{m_2})(o_1^\tau...o_{l'}^\tau|s||\textbf{\textit{do}}\ a_1^\tau...a_{l'}^\tau)\|_1 \ge A^{l'}\cdot\delta_{psr}, \qquad \forall s \in \mathcal{S}, \qquad (16)$$

for all $m_1 \ne m_2 \in [M]$. For instance, Assumption 2 alone gives (16) with $l' = l$ and $A^{l'}\cdot\delta_{psr} \ge \sigma_\tau$.

$$\sum_{a_1^\tau...a_{l'}^\tau \in \mathcal{A}^{l'}} \|(\mathbb{P}_{m_1} - \mathbb{P}_{m_2})(o_1^\tau...o_{l'}^\tau|s||\textbf{\textit{do}}\ a_1^\tau...a_{l'}^\tau)\|_1 \ge \|L_s(e_{m_1} - e_{m_2})\|_1$$

$$\ge \|L_s(e_{m_1} - e_{m_2})\|_2 \ge \sqrt{2}\sigma_\tau,$$

where $e_m$ is a standard basis vector in $\mathbb{R}^M$ with 1 at the $m^{th}$ position. If MDPs satisfy the Assumption 1, then equation (16) holds with $l' = 1$ and $\delta_{psr} = \delta$. The discussion in the main text (Section 3.4) is referring to this case.

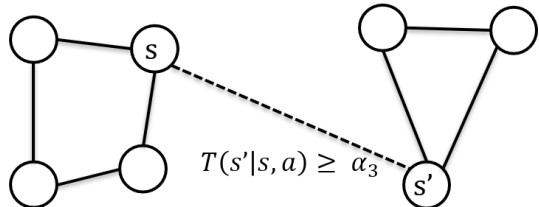

**Figure 5:** Connected graph constructed from an MDP with Assumption 4

Once the equation (16) is given true with some $\delta_{psr} = \Theta(1)$, with high probability, we can identify the context by grouping trajectories with same ending state and similar $l'$-step predictions at time-step $H - 1$. Hence a prediction at the $(H - 1)^{th}$ time step serves as a label for each trajectory.

We are then left with recovering the full LMDP models. Even though we can cluster trajectories according to predictions conditioning on length $H - 1$ histories, if we have two trajectories landed in two different states at $(H - 1)^{th}$ time-step, we have no means to combine them even if they are still from the same context. In order to resolve this, our approach requires the following assumption:

**Assumption 4** *For all $m \in [M]$, let $\mathcal{G}_m$ be an undirected graph where each node in $\mathcal{G}_m$ corresponds to each state $s \in \mathcal{S}$. Suppose we connect $(s, s')$ in $\mathcal{G}_m$ (assign an edge between $s, s'$) for $s \neq s'$ if there exists at least one action $a \in \mathcal{A}$ such that $T_m(s'|s, a) \geq \alpha_3$ for some $\alpha_3 > 0$. Then, $\mathcal{G}_m$ is connected, i.e., from any states there exists a path to any other states on $\mathcal{G}_m$.*

The high-level idea of Assumption 4 is to consider a graph between states as in Figure 5. We want to recover edges between different states $s, s'$ in $\mathcal{G}_m$ so that we can assign same labels resulted from the same context but ended at different states.

With Assumption 4, if we have a trajectory that ends with last two states $(s_{H-1}, s_H) = (s, s')$ where $s \neq s'$, then we can find labels of this trajectory according to two different labeling rules at state $s$ and $s'$. Hence, we can associate labels assigned by predictions at two different states $s, s'$. Afterwards, even if we have two trajectories ending at different states from the same context, we can assign the same label to two trajectories if we have seen a connection between $(s, s')$. In other words, this step connects labels according to the same context in different states $s, s'$. Note that even if there is no direct connection, we can infer the identical context if we have a path in a graph by crossing over states that have direct connections.

**Remark 2** *Assumption 4 is satisfied if, for instance, each MDP has a finite diameter $D > 0$ [22] where*

$$D = \min_{\pi} \max_{m, s \neq s'} \mathbb{E}_m^{\pi}[\# \text{ of steps}(s \to s')],$$

*$D$ is the minimum required number of expected steps in any MDP (with some deterministic memoryless policy $\pi$) to move from any state $s$ to any other states $s'$. In this case, each $\mathcal{G}_m$ is connected with $\alpha_3 \geq 1/D$, since if we have some disconnected groups of states in $\mathcal{G}_m$, then the diameter cannot be smaller than $1/\alpha_3$ (see also Figure 5). Note that in general, we only need $\alpha_3$ to be bounded below to make each graph $\mathcal{G}_m$ connected for all states. With the connectivity of $\mathcal{G}_m$, we can associate labels in all different states in a consistent way to resolve ambiguity in the ordering of contexts.*

As we get more trajectories that end with various $s_{H-1}$ and $s_H$, whenever $s_{H-1} \neq s_H$, we can associate labels across more different states, and recover more connections (edges in $\mathcal{G}_m$). Then, once every node in $\mathcal{G}_m$ is connected in each context $m$, we can recover full transition and reward models for the context $m$ since we resolved the ambiguity in the ordering of labels of all different states. After we recover transition and reward models, we recover initial distribution of each MDP with a few more length $H$ trajectories. The full clustering procedure is summarized in Algorithm 5.

To reliably estimate the parameters with Algorithm 5 to serve as a good initialization for Algorithm 1, we require

$$\epsilon_c \leq \frac{1}{4} \cdot \min(\epsilon_{init}, \delta_{psr}), \ (MN)^{-4} + A^{l'} \epsilon_t / \epsilon_c \leq 0.01/n_1,$$

which in turn implies the desired accuracy in total variation distance between full length $t$ trajectories: $\epsilon_t \ll A^{-l'} \epsilon_c / n_1$. From Theorem E.1, total sample complexity we need for the initialization to be

$$n_0 \geq C_0 \cdot \frac{H^2 M n_1^2}{\epsilon_c^2} \cdot \frac{A^{2l+2l'+1}}{p_s \sigma_\tau^2 \sigma_h^2} \left( S + \frac{A^l}{\sigma_h^2} \right) poly \log(N/\eta),$$

for sufficiently large absolute constant $C_0 > 0$. We conclude this section with the full end-to-end guarantees (full version of 3.5).

**Theorem E.3** *Let Assumption 2 hold for an LMDP instance with a sampling policy $\pi$. Furthermore, assume the LMDP satisfies Assumptions 1 and 3. We learn the PSR parameters with $n_0$ short trajectories of length $2l + 1$ where*

$$n_0 = poly(A^l, S, \epsilon_c^{-1}, \sigma_h^{-1}, \sigma_\tau^{-1}, \alpha_2^{-1}, \alpha_3^{-1}, H, M),$$

*where $\epsilon_c < \min(\delta, \epsilon_{init})$ is a desired accuracy for estimated predictions, and $\alpha_3 > 0$ is a parameter related to the connectivity of MDPs (see Assumption 4). Let the number of additional episodes with time-horizon $H \geq C \cdot \delta^{-4} \log^2(1/\alpha) \log(N/\eta)$ (as in Theorem 3.4)*

$$n_1 = C_1 \cdot MA \log(MS)/(\alpha_2 \alpha_3),$$

*with some absolute constant $C_1 > 0$. Then with probability at least $2/3$, Algorithm 5 returns a good initialization of LMDP parameters that satisfies (3).*

Note that the $2/3$ probability guarantee can be boosted to arbitrarily high precision $1 - \eta$ by repeating Algorithm 5 $O(\log(1/\eta))$ times, and selecting a model via majority vote.

### E.2.1 Proof of Theorem E.3

*Proof.* Let $n_1 \geq C_1 \cdot \log(n_1) MA/(\alpha_2 \alpha_3)$, $\epsilon_c = c \cdot \min(\delta_{psr}, \epsilon_{init})$ for some sufficiently large constant $C_1 > 0$ and sufficiently small constant $c > 0$. Let $\epsilon_t = \epsilon_c/(10 n_1 A^{l'})$. Plugging this to the Theorem E.1 and Theorem E.2, if we use $n_0$ short trajectories for learning PSR where

$$n_0 \geq C_0 \cdot \frac{H^2 M^3}{\epsilon_c^2 \alpha_2^2 \alpha_3^2} \cdot \frac{A^{2l+2l'+3}}{p_s \sigma_\tau^2 \sigma_h^2} \left( S + \frac{A^l}{\sigma_h^2} \right) poly \log(N/\eta),$$

then the error of the estimated conditional probability given a trajectory and a test is less than $\epsilon_c$ with probability at least $9/10$ for all $n_1$ trajectories (over the randomness of new trajectories).

With Assumption 3, with $n_1 \gg M/\alpha_2 \log(MS)$, we can visit all states in all MDPs at least once at time step $H - 1$ after $n_1/3$ episodes with probability larger than $9/10$. Furthermore, for all $n_1/3$ trajectories $h_1, ..., h_{n_1/3}$ up to $H - 1$ time step, we have

$$\|(\mathbb{P}^\pi - \hat{\mathbb{P}}^\pi)(\mathcal{T}'|h_i)\|_1 \leq A^{l'} \epsilon_c, \qquad \forall i \in [n_1],$$

with probability at least $9/10$ by union bound. Let $k_i$ and $s_i$ be the true context and ending state of $h_i$. With Assumption 1 and the separation Lemma D.1, we also have with probability at least $1 - \eta$ that

$$\|\mathbb{P}^\pi(\mathcal{T}'|h_i) - \mathbb{P}_{k_i}^\pi(\mathcal{T}'|s_i)\|_1 \leq A^{l'} \cdot (\eta/N)^4, \qquad \forall i \in [n_1],$$

where $N \gg n_1$ is the number of episodes to be run after initialization with Algorithm 1. Note that the prediction probabilities are $\delta_{psr}$-separated, Theorem E.2 ensures that all possible sets of $l'$-step predictions are within error $\epsilon_c \ll \delta_{psr}$. Thus, we are guaranteed that all $h_i$s whose estimated $\hat{\mathbb{P}}^\pi(\mathcal{T}'|h_i)$ are within $A^{l'} \epsilon_c$-error are generated from the same context. Note that with Assumption 1, we have $l' = 1$ and $\delta_{psr} = \delta$.

Suppose now that we have Assumption 1. In this case, we set $T'$ be a set of all possible observations of length 1. Now we are remained with the recovery of full transition and reward models for each context. Note that same guarantees in the previous paragraph hold for predictions at the time step $H$ with probability $9/10$. With Assumption 4 (see also Remark 2), we build a connection graph for each context. That is, with $n_1 = O(MA \log(MS)/(\alpha_2 \alpha_3))$ episodes (since we need to see at least one occurrence of all edges in all contexts, *i.e.,* all $(m, s)$ with edges to neighborhood states $s'$ via action $a$), we have pairs of $(s_{H-1}, s_H)$ in the same trajectory where $s_{H-1}$ and $s_H$ are sufficient to recover

all edges in all graphs $\mathcal{G}_m$. Note that each edge occurs with probability at least $O(\alpha_2\alpha_3/(MA))$ and there are at most $MS^2$ edges, which gives a desired number of trajectories for clustering.

More specifically, by associating 1-step predictions at time steps $H-1$ and $H$ in the same trajectory, we can connect labels found at $s_{H-1}$ with estimated quantity $\mathbb{P}_m(\cdot|s_{H-1},a)$ and the same one found at $s_H$, as we confirm these labels are in the same true context $m$ of the trajectory. We can aggregate more sample trajectories until we recover all edges in the connection graph $\mathcal{G}_m$. As long as this association results in a consistent reordering of contexts in all states, we can recover the full transition models (as well as rewards and initial distributions) for all contexts.

Now we visit every state $s$ with probability at least $\alpha_2$ at time step $H-1$ by Assumption 3. Then, by taking uniform action $a$ at time step $H-1$, with probability at least $\alpha_3/A$, we reveal the connection from $s$ to some other state $s'$ (which is essential for the consistent reordering of contexts) at time step $H$ by Assumption 4. If we repeat this process for $n_1 = C_1 \cdot MA \log(MS)/(\alpha_2\alpha_3)$ episodes, we can collect all necessary information for the reordering of contexts in all different states. In conclusion, Algorithm 5 recovers $T_m$ and $R_m$ up to $\epsilon_c$-accuracy for all $m, s, a$ (not necessarily in the same order in $m$). Initial state distributions for all contexts can be similarly recovered. The entire process succeeds with probability at least $2/3$. $\qquad\square$

## Appendix F   Proofs for Spectral Learning of PSR

In this section, we provide deferred proofs for the Lemmas used in Appendix E.1. If the norm $\|\cdot\|$ is used without subscript, we mean $l_2$-norm for vectors and operator norm for matrices.

### F.1   Proof of Theorem E.1

Let us define a few notations before we get into the detail. Let us denote $p_s = 1^\top P_{H_s} = \mathbb{P}(\text{end state} = s)$, and empirical counterpart $\hat{p}_s = 1^\top \hat{P}_{H_s}$, for the (empirical) probability of sampling a history ending with $s$. First, we normalize joint probability matrices:

$$P_{\mathcal{T},\mathcal{H}|s} = \frac{P_{\mathcal{T},\mathcal{H}_s}}{p_s}, \quad P_{\mathcal{T},oa,\mathcal{H}|s} = \frac{P_{\mathcal{T},oa,\mathcal{H}_s}}{p_s}, \quad \hat{P}_{\mathcal{T},\mathcal{H}|s} = \frac{\hat{P}_{\mathcal{T},\mathcal{H}_s}}{p_s}, \quad \hat{P}_{\mathcal{T},oa,\mathcal{H}|s} = \frac{\hat{P}_{\mathcal{T},oa,\mathcal{H}_s}}{p_s}.$$

We occasionally express unnormalized PSR states with PSR parameters $\{(b_{\infty,s}, B_{o,a,s}, b_{1,s})\}$ as given a history $(s,a,o)_{1:t-1}$ as

$$b_{t,s_1} = B_{(o,a,s)_{t-1}} B_{(o,a,s)_{t-2}} ... B_{(o,a,s)_1} b_{1,s_1} = B_{(o,a,s)_{t-1:1}} b_{1,s_1}.$$

The empirical counterpart will be defined similarly with $\hat{\cdot}$ on the top. We often concisely use $h_t$ instead of $(s,a,o)_{1:t-1} = (s,a,r)_{1:t-1}s_t$. We represent the probability of choosing actions $a_1, ..., a_{t-1}$ when the history is $h_{t-1}$ as

$$\pi(a_{1:t-1}|h_{t-1}) = \pi(a_1|h_1)\pi(a_2|h_2)...\pi(a_{t-1}|h_{t-1}).$$

Now suppose that empirical estimates of probability matrices satisfy the following:

$$\|P_{\mathcal{H}_s} - \hat{P}_{\mathcal{H}_s}\|_2 \le \epsilon_{0,s}$$
$$\|\mathbb{P}(\mathcal{T}, s_1 = s) - \hat{\mathbb{P}}(\mathcal{T}, s_1 = s)\|_2 \le \epsilon_{1,s}$$
$$\|P_{\mathcal{T},\mathcal{H}|s} - \hat{P}_{\mathcal{T},\mathcal{H}|s}\|_2 \le \epsilon_{2,s}$$
$$\|P_{\mathcal{T},oa,\mathcal{H}|s} - \hat{P}_{\mathcal{T},oa,\mathcal{H}|s}\|_2 \le \epsilon_{3,oas},$$

for all $s, a, o$. The following lemma shows how the error in estimated matrices affects the accuracy of PSR parameters.

**Lemma F.1** *Let the true transformed PSR parameters with $\hat{U}_s, \hat{V}_s$ be*

$$\tilde{B}_{o,a,s} = \hat{U}_{s'}^\top P_{\mathcal{T},oas,\mathcal{H}_s} \hat{V}_s (\hat{U}_s^\top P_{\mathcal{T},\mathcal{H}_s} \hat{V}_s)^{-1},$$
$$\tilde{b}_{\infty,s} = P_{\mathcal{H}_s}^\top \hat{V}_s (\hat{U}_s^\top P_{\mathcal{T},\mathcal{H}_s} \hat{V}_s)^{-1},$$
$$\tilde{b}_{1,s} = \hat{U}_s^\top \mathbb{P}(\mathcal{T}, s_1 = s),$$

for all $s, a, o$. Let $\sigma_{M,s}$ be the minimum $(M^{th})$ singular value of $\sigma_M(\hat{U}_s^\top P_{\mathcal{T}, \mathcal{H}|s} \hat{V}_s)$. Then, we have that

$$\|\tilde{B}_{o,a,s} - \hat{B}_{o,a,s}\|_2 \leq \frac{\epsilon_{3,oas}}{\sigma_{M,s}} + \sqrt{A^l}\mathbb{P}^\pi(o|s||\textbf{\textit{do}}\ a)\frac{2\epsilon_{2,s}}{\sigma_{M,s}^2},$$

$$\|\tilde{b}_{s,\infty} - \hat{b}_{s,\infty}\|_2 \leq \frac{\epsilon_{0,s}}{p_s \sigma_{M,s}} + \frac{2\epsilon_{2,s}}{\sigma_{M,s}^2},$$

$$\|\tilde{b}_{s,1} - \hat{b}_{s,1}\|_2 \leq \epsilon_{1,s},$$

where $\mathbb{P}^\pi(\cdot)$ is the probability of events when we sample histories with the exploration policy $\pi$.

The proofs of helping lemmas will be proved at the last of this subsection. We define the following quantities with error bounds similarly as in [20]:

$$\delta_{\infty,s} = \|L_s^\top \hat{U}_s(\tilde{b}_{\infty,s} - \hat{b}_{\infty,s})\|_\infty \leq \|L_s^\top\|_{\infty,2}\|\tilde{b}_{\infty,s} - \hat{b}_{\infty,s}\|_2 \leq \sqrt{A^l}\|\tilde{b}_{\infty,s} - \hat{b}_{\infty,s}\|_2,$$

$$\delta_{1,s} = \|(\hat{U}_s^\top L_s)^{-1}(\tilde{b}_{1,s} - \hat{b}_{1,s})\|_1 \leq \sqrt{M}\|\tilde{b}_{1,s} - \hat{b}_{1,s}\|_2/\sigma_M(\hat{U}_s^\top L_s),$$

$$\Delta_{o,a,s} = \|(\hat{U}_s^\top L_s)^{-1}(\tilde{B}_{o,a,s} - \hat{B}_{o,a,s})(\hat{U}_s^\top L_s)\|_1 \leq \sqrt{M}\|\tilde{B}_{o,a,s} - \hat{B}_{o,a,s}\|_2/\sigma_M(\hat{U}_s^\top L_s),$$

$$\Delta = \max_{a,s}\sum_o \Delta_{o,a,s},\ \delta_\infty = \max_s \delta_{\infty,s},\ \delta_1 = \max_s \delta_{1,s}. \tag{17}$$

We let $\epsilon_t = \delta_\infty + (1 + \delta_\infty)((1 + \Delta)^t \delta_1 + (1 + \Delta)^t - 1)$. We first note that for any fixed action sequence $a_{1:t-1}$, it holds that

$$\sum_{(s,o)_{1:t-1}} |\tilde{b}_{\infty,s}^\top \tilde{B}_{(o,a,s)_{t-1:1}} \tilde{b}_{1,s_1} - \hat{b}_{\infty,s}^\top \hat{B}_{(o,a,s)_{t-1:1}} \hat{b}_{1,s_1}|$$

$$\leq \delta_\infty + (1 + \delta_\infty)((1 + \Delta)^t \delta_1 + (1 + \Delta)^t - 1).$$

This equation is a direct consequence of the Lemma 12 in [20]. However, here we aim to get the bound for *all* history dependent policy, hence we need to establish the theorem by re-deriving the induction hypothesis with considering the policy. We now bound the original equation. Observe first that

$$\sum_{(s,a,r)_{1:t-1},s_t} |\mathbb{P}^\pi((s,a,r)_{1:t-1}, s_t) - \hat{\mathbb{P}}^\pi((s,a,r)_{1:t-1}, s_t)|$$

$$= \sum_{(s,a,r)_{1:t-1},s_t} \pi(a_{1:t-1}|h_{t-1})|\tilde{b}_{\infty,s}^\top \tilde{B}_{(o,a,s)_{t-1:1}} \tilde{b}_{1,s_1} - \hat{b}_{\infty,s}^\top \hat{B}_{(o,a,s)_{t-1:1}} \hat{b}_{1,s_1}|.$$

Following the steps in [20], for each $s_1$, we will prove the following Lemma:

**Lemma F.2** *For any t, we have*

$$\sum_{(s,a,o)_{1:t-1}} \pi(a_{1:t-1}|h_{t-1})\|(\hat{U}_{s_t}^\top L_{s_t})^{-1}(\tilde{B}_{(o,a,s)_{t-1:1}} \tilde{b}_{1,s_1} - \hat{B}_{(o,a,s)_{t-1:1}} \hat{b}_{1,s_1})\|_1$$

$$\leq (1 + \Delta)^t \delta_1 + (1 + \Delta)^t - 1. \tag{18}$$

We are now ready to prove the original claim. Let us denote $\tilde{b}_{t,s_1} = \tilde{B}_{(o,a,s)_{t-1:1}} \tilde{b}_{1,s_1}$ and $\hat{b}_{t,s_1} = \hat{B}_{(o,a,s)_{t-1:1}} \hat{b}_{1,s_1}$. The remaining step is to involve the effect of error in $\hat{b}_{\infty,s_t}$. Following the similar steps, we decompose the summation as:

$$\sum_{(s,a,o)_{1:t-1}} \pi(a_{1:t-1}|h_{t-1})|\tilde{b}_{\infty,s_t}^\top \tilde{B}_{(o,a,s)_{t-1:1}} \tilde{b}_{1,s_1} - \hat{b}_{\infty,s_t}^\top \hat{B}_{(o,a,s)_{t-1:1}} \hat{b}_{1,s_1}|$$

$$= \sum_{(s,a,o)_{1:t-1}} \pi(a_{1:t-1}|h_{t-1})|(\tilde{b}_{\infty,s_t} - \hat{b}_{\infty,s_t})^\top (\hat{U}_{s_t}^\top L_{s_t})(\hat{U}_{s_t}^\top L_{s_t})^{-1}\tilde{b}_{t,s_1}|$$

$$+ \sum_{(s,a,o)_{1:t-1}} \pi(a_{1:t-1}|h_{t-1})|(\tilde{b}_{\infty,s_t} - \hat{b}_{\infty,s_t})^\top (\hat{U}_{s_t}^\top L_{s_t})(\hat{U}_{s_t}^\top L_{s_t})^{-1}(\tilde{b}_{t,s_1} - \hat{b}_{t,s_1})|$$

$$+ \sum_{(s,a,o)_{1:t-1}} \pi(a_{1:t-1}|h_{t-1})|\tilde{b}_{\infty,s_t}^\top(\hat{U}_{s_t}^\top L_{s_t})(\hat{U}_{s_t}^\top L_{s_t})^{-1}(\tilde{b}_{t,s_1} - \hat{b}_{t,s_1})|.$$

For the first term,

$$\sum_{(s,a,o)_{1:t-1}} \pi(a_{1:t-1}|h_{t-1})|(\tilde{b}_{\infty,s_t} - \hat{b}_{\infty,s_t})^\top(\hat{U}_{s_t}^\top L_{s_t})(\hat{U}_{s_t}^\top L_{s_t})^{-1}\tilde{b}_{t,s_1}|$$

$$\leq \sum_{(s,a,o)_{1:t-1}} \pi(a_{1:t-1}|h_{t-1})\|(\tilde{b}_{\infty,s_t} - \hat{b}_{\infty,s_t})^\top(\hat{U}_{s_t}^\top L_{s_t})\|_\infty\|(\hat{U}_{s_t}^\top L_{s_t})^{-1}\tilde{b}_{t,s_1}\|_1$$

$$\leq \sum_{(s,a,o)_{1:t-1}} \pi(a_{1:t-1}|h_{t-1})\delta_{\infty,s_t}\|(\hat{U}_{s_t}^\top L_{s_t})^{-1}\tilde{b}_{t,s_1}\|_1$$

$$\leq \delta_\infty \sum_{(s,a,o)_{1:t-1}} \pi(a_{1:t-1}|h_{t-1})\mathbb{P}(h_t|a_{1:t-1}) \leq \delta_\infty.$$

Following the similar step, the second term is bounded by $\delta_\infty((1+\Delta)^t\delta_1 + (1+\Delta)^t - 1)$. For the last term, note that $\tilde{b}_{\infty,s_t}^\top(\hat{U}_{s_t}^\top L_{s_t}) = 1^\top$. Therefore,

$$\sum_{(s,a,o)_{1:t-1}} \pi(a_{1:t-1}|h_{t-1})|\tilde{b}_{\infty,s_t}^\top(\hat{U}_{s_t}^\top L_{s_t})(\hat{U}_{s_t}^\top L_{s_t})^{-1}(\tilde{b}_{t,s_1} - \hat{b}_{t,s_1})|$$

$$\leq \sum_{(s,a,o)_{1:t-1}} \pi(a_{1:t-1}|h_{t-1})\|(\hat{U}_{s_t}^\top L_{s_t})^{-1}(\tilde{b}_{t,s_1} - \hat{b}_{t,s_1})\|_1$$

$$\leq ((1+\Delta)^t\delta_1 + (1+\Delta)^t - 1).$$

Therefore, we conclude that

$$\sum_{(s,a,o)_{1:t-1}} |\mathbb{P}^\pi((s,a,o)_{1:t-1}) - \hat{\mathbb{P}}^\pi((s,a,o)_{1:t-1})| \leq \delta_\infty + (1+\delta_\infty)((1+\Delta)^t\delta_1 + (1+\Delta)^t - 1).$$

Finally, in other to make the error term smaller than $\epsilon_t$, we want the followings:

$$\delta_\infty \leq \epsilon_t/8, \Delta \leq \epsilon_t/4t, \delta_1 \leq \epsilon_t/4.$$

We need the following lemma on finite-sample error in estimated probability matrices:

**Lemma F.3** *For a sufficiently large constant $C > 0$, the errors in empirical estimates of the probability matrices are bounded by*

$$\|P_{\mathcal{H}_s} - \hat{P}_{\mathcal{H}_s}\|_2 \leq C\sqrt{\frac{p_s}{N}\log(SA/\eta)},$$

$$\|P_{\mathcal{T},\mathcal{H}_s} - \hat{P}_{\mathcal{T},\mathcal{H}_s}\|_2 \leq CA^l\sqrt{\frac{p_s}{N}\log(SA/\eta)},$$

$$\|\mathbb{P}(\mathcal{T}, s_1 = s) - \hat{\mathbb{P}}(\mathcal{T}, s_1 = s)\|_2 \leq CA^l\sqrt{\frac{\mathbb{P}(s_1 = s)}{N}\log(SA/\eta)},$$

$$\|P_{\mathcal{T},oa,\mathcal{H}_s} - \hat{P}_{\mathcal{T},oa,\mathcal{H}_s}\|_2 \leq CA^{l+1}\left(\sqrt{\frac{\mathbb{P}^\pi(o|s||\boldsymbol{do}\,a)p_s}{NA}\log(SA/\eta)} + \frac{\log(SA/\eta)}{N}\right),$$

*for all $s, a, o$ with probability at least $1 - \eta$.*

This lemma follows the same concentration argument to Proposition 19 in [20] using McDiarmid's inequality. The proofs of three lemmas are given at the end of this subsection. With Lemma F.1, F.3 and equation (17), we now decide the sample size. For $\Delta$,

$$\Delta \leq \sum_o \Delta_{o,a,s}$$

$$\leq \frac{\sqrt{M}}{\sigma_M(\hat{U}_s^\top L_s)}\left(\sum_o \frac{\epsilon_{3,oas}}{\sigma_{M,s}} + \sqrt{A^l}\mathbb{P}^\pi(o|s||\boldsymbol{do}\,a)\frac{2\epsilon_{2,s}}{\sigma_{M,s}^2}\right)$$

$$\leq \frac{\sqrt{M}}{\sigma_M(\hat{U}_s^\top L_s)} \left( \frac{\sum_o \epsilon_{3,oas}}{\sigma_{M,s}} + \sqrt{A^l} \frac{2\epsilon_{2,s}}{\sigma_{M,s}^2} \right).$$

The summation of $\epsilon_{3,oas}$ is bounded by

$$\sum_o \epsilon_{3,oas} \leq CA^{l+1} \sum_o \left( \sqrt{\frac{\mathbb{P}^\pi(o|s||\boldsymbol{do}\ a)}{NAp_s} \log(SA/\eta)} + \frac{\log(SA/\eta)}{Np_s} \right)$$

$$\leq CA^{l+1} \left( \sqrt{\frac{2S}{NAp_s} \log(SA/\eta)} + \frac{2S\log(SA/\eta)}{Np_s} \right).$$

Also note that

$$\epsilon_{2,s} \leq CA^l \sqrt{\frac{\log(SA/\eta)}{Np_s}}.$$

In order to have $\Delta < \epsilon_t/(4t)$, the sample size should be at least

$$N \geq C' \cdot K \frac{t^2}{\epsilon_t^2} \left( \frac{A^{2l+1}S}{p_s\sigma_M^2(\hat{U}_s^\top L_s)\sigma_{M,s}^2} + \frac{A^{3l+1}}{p_s\sigma_K^2(\hat{U}_s^\top L_s)\sigma_{M,s}^4} \right) \log(SA/\eta),$$

for some large constant $C' > 0$.

Finally, $\sigma_{M,s} = \sigma_M(\hat{U}_s^\top P_{\mathcal{T},\mathcal{H}|s}\hat{V}_s) \geq (1-\epsilon_0)\sigma_M(P_{\mathcal{T},\mathcal{H}|s})$ where $\epsilon_0 = \epsilon_{2,s}^2/((1-\epsilon_t)\sigma_M(P_{\mathcal{T},\mathcal{H}|s}))^2$ by applying Lemma F.9 twice. Hence, as long as $N \gg 1/\sigma_M(P_{\mathcal{T},\mathcal{H}|s})$, it holds that $\sigma_{M,s} \geq \sigma_M(P_{\mathcal{T},\mathcal{H}|s})/2$. Similarly, we have $\sigma_M(\hat{U}_s^\top L_s) \geq \sigma_M(L_s)/2$. Plugging this inequality in the sample complexity completes the Theorem E.1.

## F.2 Proof of Theorem E.2

*Proof.* We first define an extended policy $\pi'$ which runs the given policy $\pi$ for $t$ times and play intervening action sequences $a_t...a_{t+l-1}$. Let us denote $o = (r, s')$ to represent a pair of reward and next state compactly. A simple corollary of Theorem E.1 is the following lemma:

**Lemma F.4** *With the estimated PSR parameters in Theorem E.1, for any given trajectory $(s, a, o)_{1:t-1}$, the following holds:*

$$\sum_{a_t,r_t,...,s_{t+l}} \pi(a_{1:t-1}|h_{t-1})|b_{\infty,s_{t+l}}^\top B_{(o,a,s)_{t+l-1:1}} b_{1,s_1} - \hat{b}_{\infty,s_{t+l}}^\top \hat{B}_{(o,a,s)_{t+l-1:1}} \hat{b}_{1,s_1}|$$

$$\leq \epsilon_l \mathbb{P}^\pi((s,a,o)_{1:t-1}) + 2\pi(a_{1:t-1}|h_{t-1})\|(\hat{U}_{s_t}^\top L_{s_t})^{-1}(\tilde{b}_{t,s_1} - \hat{b}_{t,s_1})\|_1.$$

On top of this lemma, we also have the following lemma that bounds the probability of bad events in which the error in estimated probability can be arbitrarily large:

**Lemma F.5** *For any history-dependent policy $\pi$, with the PSR parameters guaranteed in Theorem E.1, we have*

$$|\hat{\mathbb{P}}^{\pi'}((s,a,o)_{1:t-1}) - \mathbb{P}^\pi((s,a,o)_{1:t-1})| \leq \epsilon_c \mathbb{P}^\pi((s,a,o)_{1:t-1}),$$

$$\pi(a_{1:t-1}|h_{t-1})\|(\hat{U}_{s_t}^\top L_{s_t})^{-1}(\tilde{b}_{t,s_1} - \hat{b}_{t,s_1})\|_1 \leq \epsilon_c \mathbb{P}^\pi((s,a,o)_{1:t-1}),$$

*with probability at least $1 - \epsilon_t/\epsilon_c$.*

By the definition of conditional test probability, note that

$$|\hat{\mathbb{P}}^{\pi'}(\tau|(s,a,o)_{1:t-1}) - \mathbb{P}^{\pi'}(\tau|(s,a,o)_{1:t-1})| = \left| \frac{\hat{\mathbb{P}}^{\pi'}(\tau,(s,a,o)_{1:t-1})}{\hat{\mathbb{P}}^\pi((s,a,o)_{1:t-1})} - \frac{\mathbb{P}^{\pi'}(\tau,(s,a,o)_{1:t-1})}{\mathbb{P}^\pi((s,a,o)_{1:t-1})} \right|,$$

which is less than

$$\frac{(1+\epsilon_c)}{\mathbb{P}^\pi((s,a,o)_{1:t-1})}\left|\hat{\mathbb{P}}^{\pi'}(\tau,(s,a,o)_{1:t-1}) - \mathbb{P}^{\pi'}(\tau,(s,a,o)_{1:t-1})\right| + \epsilon_c \frac{\mathbb{P}^{\pi'}(\tau,(s,a,o)_{1:t-1})}{\mathbb{P}^\pi((s,a,o)_{1:t-1})},$$

with probability at least $1 - \epsilon_t/\epsilon_c$. Now we sum over all possible trajectories $\tau$ in $\mathcal{O}$ with intervening actions $a_1...a_l$ after observing $(s,a,o)_{1:t-1}$. Under the good event guaranteed in Lemma F.5, the summation over all possible future trajectories is less than

$$\frac{(1+\epsilon_c)}{\mathbb{P}^\pi((s,a,o)_{1:t-1})}\left((\epsilon_t + 2\epsilon_c)\mathbb{P}^\pi((s,a,o)_{1:t-1}) + \epsilon_c \mathbb{P}^\pi((s,a,o)_{1:t-1})\right) \le 4\epsilon_c,$$

from Lemma F.4 and F.5. Therefore, for a fixed intervening action sequences $a_t,...,a_{t+l-1}$, we can conclude that

$$\|\mathbb{P}^\pi(\mathcal{O}|(s,a,o)_{1:t-1}||\textbf{\textit{do}}\ a_{t:t+l-1}) - \hat{\mathbb{P}}^\pi(\mathcal{O}|(s,a,o)_{1:t-1}||\textbf{\textit{do}}\ a_{t:t+l-1})\|_1 \le 4\epsilon_c,$$

with probability at least $1 - \epsilon_t/\epsilon_c$. $\qquad\square$

## F.3 Proof of Lemma F.1

*Proof.* The proof of the lemma can be done by unfolding expressions:

$$\begin{aligned}
\|\tilde{B}_{o,a,s} - \hat{B}_{o,a,s}\| &= \|\hat{U}_{s'}^\top P_{\mathcal{T},oa,\mathcal{H}|s}\hat{V}_s(\hat{U}_s^\top P_{\mathcal{T},\mathcal{H}|s}\hat{V}_s)^{-1} - \hat{U}_{s'}^\top P_{\mathcal{T},oa,\mathcal{H}|s}\hat{V}_s(\hat{U}_s^\top \hat{P}_{\mathcal{T},\mathcal{H}|s}\hat{V}_s)^{-1}\| \\
&\le \|(\hat{U}_{s'}^\top(P_{\mathcal{T},oa,\mathcal{H}|s} - \hat{P}_{\mathcal{T},oa,\mathcal{H}|s})\hat{V}_s)(\hat{U}_s^\top P_{\mathcal{T},\mathcal{H}|s}\hat{V}_s)^{-1}\| \\
&\quad + \|(\hat{U}_{s'}^\top P_{\mathcal{T},oa,\mathcal{H}|s}\hat{V}_s)((\hat{U}_s^\top P_{\mathcal{T},\mathcal{H}|s}\hat{V}_s)^{-1} - (\hat{U}_s^\top \hat{P}_{\mathcal{T},\mathcal{H}|s}\hat{V}_s)^{-1})\| \\
&\le \frac{\epsilon_{3,oas}}{\sigma_{M,s}} + \|P_{\mathcal{T},oa,\mathcal{H}|s}\|_2\frac{2\epsilon_{2,s}}{\sigma_{M,s}^2} \le \frac{\epsilon_{3,oas}}{\sigma_{M,s}} + \mathbb{P}^\pi(o|s||\textbf{\textit{do}}\ a)\sqrt{A^l}\frac{2\epsilon_{2,s}}{\sigma_{M,s}^2},
\end{aligned}$$

where we used Lemma F.10 from matrix perturbation theory for the second inequality, and

$$\begin{aligned}
\|P_{\mathcal{T},ao,\mathcal{H}_s}\|_2 &\le \sqrt{\sum_{\tau\in\mathcal{T},h\in\mathcal{H}_s}\mathbb{P}^\pi(oo_1^\tau o_2^\tau...o_l^\tau|h||\textbf{\textit{do}}\ aa_1^\tau...a_l^\tau)^2\mathbb{P}^\pi(h)^2} \\
&\le \sqrt{\sum_{a_1,a_2,...,a_l}\sum_{o_1,...,o_l}\sum_{h\in\mathcal{H}_s}\mathbb{P}^\pi(oo_1...a_l|h||\textbf{\textit{do}}\ aa_1...a_l)^2\mathbb{P}^\pi(h)^2} \\
&\le \sqrt{\sum_{a_1,a_2,...,a_l}\mathbb{P}^\pi(o|h||\textbf{\textit{do}}\ a)^2\sum_{o_1,...,o_l}\sum_{h\in\mathcal{H}_s}\mathbb{P}^\pi(o_1...o_l|hao||\textbf{\textit{do}}\ a_1...a_l)^2\mathbb{P}^\pi(h)^2} \\
&\le \mathbb{P}^\pi(o|h||\textbf{\textit{do}}\ a)\sqrt{\sum_{a_1,a_2,...,a_l}\sum_{h\in\mathcal{H}_s}\mathbb{P}^\pi(h)^2} = \mathbb{P}^\pi(o|h||\textbf{\textit{do}}\ a)\sqrt{A^l}\sqrt{\sum_{h\in\mathcal{H}_s}\mathbb{P}^\pi(h)^2} \\
&\le \mathbb{P}^\pi(o|h||\textbf{\textit{do}}\ a)\sqrt{A^l}\sum_{h\in\mathcal{H}_s}\mathbb{P}^\pi(h) = \mathbb{P}^\pi(o|h||\textbf{\textit{do}}\ a)\sqrt{A^l}p_s,
\end{aligned}$$

therefore $\|P_{\mathcal{T},oa,\mathcal{H}|s}\| \le \mathbb{P}^\pi(o|h||\textbf{\textit{do}}\ a)\sqrt{A^l}$ for the last inequality. For initial and normalization parameters,

$$\begin{aligned}
\|\tilde{b}_{\infty,s} - \hat{b}_{\infty,s}\| &\le \|(P_{\mathcal{H}_s} - \hat{P}_{\mathcal{H}_s})^\top\hat{V}_s(\hat{U}_s^\top P_{\mathcal{T},\mathcal{H}_s}\hat{V}_s)^{-1}\| + \|P_{\mathcal{H}_s}^\top\hat{V}_s((\hat{U}_s^\top P_{\mathcal{T},\mathcal{H}_s}\hat{V}_s)^{-1} - (\hat{U}_s^\top \hat{P}_{\mathcal{T},\mathcal{H}_s}\hat{V}_s)^{-1})\| \\
&\le \frac{\epsilon_{0,s}}{\sigma_M(P_{\mathcal{T},\mathcal{H}_s})} + \|P_{\mathcal{H}_s}/\hat{p}_s\|_2\frac{2\epsilon_{2,s}}{\sigma_M(P_{\mathcal{T},\mathcal{H}|s})^2} \le \frac{\epsilon_{0,s}}{p_s\sigma_M(P_{\mathcal{T},\mathcal{H}|s})} + \frac{2\epsilon_{2,s}}{\sigma_M(P_{\mathcal{T},\mathcal{H}|s})^2}.
\end{aligned}$$

$$\|\tilde{b}_{s,1} - \hat{b}_{s,1}\| \le \epsilon_{1,s}.$$

$\qquad\square$

### F.4 Proof of Lemma F.2

*Proof.* We show this lemma by induction on $t$. For $t = 1$, we bound $\|(\hat{U}_{s_1}L_{s_1})^{-1}(\tilde{b}_{1,s_1} - \hat{b}_{1,s_1})\|_1 \leq \delta_{1,s_1}$ by definition. Now assume it holds for $t-1$ and check the induction hypothesis.

$$\sum_{(s,a,o)_{1:t-1}} \pi(a_{1:t-1}|h_{t-1})\|(\hat{U}_{s_t}^\top L_{s_t})^{-1}(\tilde{B}_{(o,a,s)_{t-1:1}}\tilde{b}_{1,s_1} - \hat{B}_{(o,a,s)_{t-1:1}}\hat{b}_{1,s_1})\|_1$$

$$= \sum_{(s,a,o)_{1:t-2}} \sum_{a_{t-1},o_{t-1}} \pi(a_{t-1}|h_{t-1})\pi(a_{1:t-2}|h_{t-2})\|(\hat{U}_{s_t}^\top L_{s_t})^{-1}(\tilde{B}_{(o,a,s)_{t-1:1}}\tilde{b}_{1,s_1} - \hat{B}_{(o,a,s)_{t-1:1}}\hat{b}_{1,s_1})\|_1$$

$$= \sum_{a_{t-1}} \pi(a_{t-1}|h_{t-1}) \sum_{o_{t-1}} \sum_{(s,a,o)_{1:t-2}} \pi(a_{1:t-2}|h_{t-2})\|(\hat{U}_{s_t}^\top L_{s_t})^{-1}(\tilde{B}_{(o,a,s)_{t-1:1}}\tilde{b}_{1,s_1} - \hat{B}_{(o,a,s)_{t-1:1}}\hat{b}_{1,s_1})\|_1$$

$$= \sum_{a_{t-1}} \pi(a_{t-1}|h_{t-1}) \sum_{o_{t-1}} \sum_{(s,a,o)_{1:t-2}} \pi(a_{1:t-2}|h_{t-2})\|(\hat{U}_{s_t}^\top L_{s_t})^{-1}(\tilde{b}_{t,s_1} - \hat{b}_{t,s_1})\|_1.$$

We investigate the inside sum by decomposing $\|(\hat{U}_{s_t}^\top L_{s_t})^{-1}(\tilde{b}_{t,s_1} - \hat{b}_{t,s_1})\|_1$ as

$$\|(\hat{U}_{s_t}^\top L_{s_t})^{-1}(\tilde{b}_{t,s_1}-\hat{b}_{t,s_1})\|_1 = \|(\hat{U}_{s_t}^\top L_{s_t})^{-1}(\tilde{B}_{(o,a,s)_{t-1}} - \hat{B}_{(o,a,s)_{t-1}})(\hat{U}_{s_{t-1}}^\top L_{s_{t-1}})\|_1 \|(\hat{U}_{s_{t-1}}^\top L_{s_{t-1}})^{-1}\tilde{b}_{t-1,s_1}\|_1$$

$$+ \|(\hat{U}_{s_t}^\top L_{s_t})^{-1}(\tilde{B}_{(o,a,s)_{t-1}} - \hat{B}_{(o,a,s)_{t-1}})(\hat{U}_{s_{t-1}}^\top L_{s_{t-1}})\|_1 \|(\hat{U}_{s_{t-1}}^\top L_{s_{t-1}})^{-1}(\tilde{b}_{t-1,s_1} - \hat{b}_{t-1,s_1})\|_1$$

$$+ \|(\hat{U}_{s_t}^\top L_{s_t})^{-1}\tilde{B}_{(o,a,s)_{t-1}}(\hat{U}_{s_{t-1}}^\top L_{s_{t-1}})(\hat{U}_{s_{t-1}}^\top L_{s_{t-1}})^{-1}(\tilde{b}_{t-1,s_1} - \hat{b}_{t-1,s_1})\|_1.$$

For the first term,

$$\sum_{o_{t-1}} \sum_{(s,a,o)_{1:t-2}} \pi(a_{1:t-2}|h_{t-2})\|(\hat{U}_{s_t}^\top L_{s_t})^{-1}(\tilde{B}_{(o,a,s)_{t-1}} - \hat{B}_{(o,a,s)_{t-1}})(\hat{U}_{s_{t-1}}^\top L_{s_{t-1}})\|_1 \|(\hat{U}_{s_{t-1}}^\top L_{s_{t-1}})^{-1}\tilde{b}_{t-1,s_1}\|_1$$

$$= \sum_{o_{t-1}} \sum_{(s,a,o)_{1:t-2}} \pi(a_{1:t-2}|h_{t-2})\Delta_{(o,a,s)_{t-1}}\|(\hat{U}_{s_{t-1}}^\top L_{s_{t-1}})^{-1}\tilde{b}_{t-1,s_1}\|_1$$

$$\leq \Delta \sum_{(s,a,o)_{1:t-2}} \pi(a_{1:t-2}|h_{t-2})\|(\hat{U}_{s_{t-1}}^\top L_{s_{t-1}})^{-1}\tilde{b}_{t-1,s_1}\|_1$$

$$= \Delta \sum_{(s,a,o)_{1:t-2}} \pi(a_{1:t-2}|h_{t-2})\mathbb{P}(h_{t-2}) = \Delta,$$

where we used the definition of $\tilde{b}_{t-1,s_1} = \hat{U}_{s_{t-1}}^\top L_{s_{t-1}}\mathbb{P}((s,a,o)_{1:t-2})$. For the second term, by the induction hypothesis,

$$\sum_{o_{t-1}} \sum_{(s,a,o)_{1:t-2}} \pi(a_{1:t-2}|h_{t-2})\Delta_{(o,a,s)_{t-1}}\|(\hat{U}_{s_{t-1}}^\top L_{s_{t-1}})^{-1}(\tilde{b}_{t-1,s_1} - \hat{b}_{t-1,s_1})\|_1$$

$$\leq \Delta \sum_{(s,a,o)_{1:t-2}} \pi(a_{1:t-2}|h_{t-2})\|(\hat{U}_{s_{t-1}}^\top L_{s_{t-1}})^{-1}(\tilde{b}_{t-1,s_1} - \hat{b}_{t-1,s_1})\|_1$$

$$= \Delta((1+\Delta)^{t-1}\delta_1 + (1+\Delta)^{t-1} - 1).$$

The last term can also be derived following the same argument in [20]. It gives

$$\sum_{o_{t-1}} \sum_{(s,a,o)_{1:t-2}} \pi(a_{1:t-2}|h_{t-2})\|(\hat{U}_{s_t}^\top L_{s_t})^{-1}\tilde{B}_{(o,a,s)_{t-1}}(\hat{U}_{s_{t-1}}^\top L_{s_{t-1}})(\hat{U}_{s_{t-1}}^\top L_{s_{t-1}})^{-1}(\tilde{b}_{t-1,s_1} - \hat{b}_{t-1,s_1})\|_1$$

$$\leq \sum_{o_{t-1}} \sum_{(s,a,o)_{1:t-2}} \pi(a_{1:t-2}|h_{t-2})\|D_{(o,a,s)_{t-1}}(\hat{U}_{s_{t-1}}^\top L_{s_{t-1}})^{-1}(\tilde{b}_{t-1,s_1} - \hat{b}_{t-1,s_1})\|_1$$

$$\leq \sum_{o_{t-1}} \sum_{(s,a,o)_{1:t-2}} \pi(a_{1:t-2}|h_{t-2})\|D_{(o,a,s)_{t-1}}\|_1\|(\hat{U}_{s_{t-1}}^\top L_{s_{t-1}})^{-1}(\tilde{b}_{t-1,s_1} - \hat{b}_{t-1,s_1})\|_1$$

$$\leq (1+\Delta)^{t-1}\delta_1 + (1+\Delta)^{t-1} - 1.$$

Now combining these three bounds, we get

$$\sum_{a_{t-1}} \pi(a_{t-1}|h_{1:t-1}) \sum_{o_{t-1}} \sum_{(a,r,s')_{1:t-2}} \pi(a_{1:t-2}|h_{1:t-2})\|(\hat{U}_{s_t}^\top L_{s_t})^{-1}(\tilde{b}_{t,s_1} - \hat{b}_{t,s_1})\|_1$$

$$\leq \sum_{a_{t-1}} \pi(a_{t-1}|h_{1:t-1})(\Delta + (1+\Delta)((1+\Delta)^{t-1}\delta_1 + (1+\Delta)^{t-1} - 1))$$

$$\leq \sum_{a_{t-1}} \pi(a_{t-1}|h_{1:t-1})((1+\Delta)^t \delta_1 + (1+\Delta)^t - 1) = (1+\Delta)^t \delta_1 + (1+\Delta)^t - 1.$$

$\square$

### F.5 Proof of Lemma F.3

*Proof.* For the first inequality, we note that

$$\|P_{\mathcal{H}_s} - \hat{P}_{\mathcal{H}_s}\|_2 \leq p_s \|P_{\mathcal{H}|s} - \hat{P}_{\mathcal{H}|s}\|_2 + |p_s - \hat{p}_s| \|P_{\mathcal{H}|s}\|_2.$$

Let $N_s = \hat{p}_s N$. For the first term, we can use McDiarmid's inequality since a change at one sample among $N_s$ samples (conditioned on starting test from $s$) causes only $\sqrt{2/N_s}$ difference:

$$\|P_{\mathcal{H}|s} - \hat{P}_{\mathcal{H}|s}\|_2 - \mathbb{E}[\|P_{\mathcal{H}|s} - \hat{P}_{\mathcal{H}|s}\|_2] \lesssim \sqrt{\frac{1}{N_s} \ln(1/\eta)},$$

with probability at least $1 - \eta/100$. Let $\#(h_{s,i})$ be a count of a history $h_{s,i}$ after seeing $N_s$ histories that end with $s$. Also,

$$\mathbb{E}[\|P_{\mathcal{H}|s} - \hat{P}_{\mathcal{H}|s}\|_2] \leq \sqrt{\mathbb{E}[\|P_{\mathcal{H}|s} - \hat{P}_{\mathcal{H}|s}\|_2^2]} \leq \sqrt{\sum_{i=1}^{|H_s|} Var\left(\frac{1}{N_s}\#(h_{s,i})\Big|s\right)}$$

$$\leq \sqrt{\frac{1}{N_s} \sum_{i=1}^{|H_s|} \mathbb{P}(h_{s,i}|s)} \leq \sqrt{\frac{1}{N_s}}.$$

Therefore, we can conclude that $\|P_{\mathcal{H}|s} - \hat{P}_{\mathcal{H}|s}\|_2 \lesssim \sqrt{\frac{1}{N_s} \ln(1/\eta)}$. On the other hand, we can show that $|p_s - \hat{p}_s| \lesssim \sqrt{\frac{p_s}{N} \log(1/\eta)} + \frac{\log(1/\eta)}{N}$ via a simple application of Bernstein's inequality. Note that our sample complexity guarantees $N \gg \log(1/\eta)/p_s$. Hence,

$$\|P_{\mathcal{H}_s} - \hat{P}_{\mathcal{H}_s}\|_2 \leq p_s \|P_{\mathcal{H}|s} - \hat{P}_{\mathcal{H}|s}\|_2 + |p_s - \hat{p}_s|\|P_{\mathcal{H}|s}\|_2$$

$$\lesssim p_s \sqrt{\frac{1}{N_s} \log(1/\eta)} + \sqrt{\frac{p_s}{N} \log(1/\eta)}\|P_{\mathcal{H}|s}\|_2 \lesssim \sqrt{\frac{p_s}{N} \log(1/\eta)}.$$

Similarly, we can show that

$$\mathbb{E}[\|P_{\mathcal{T},\mathcal{H}|s} - \hat{P}_{\mathcal{T},\mathcal{H}|s}\|_2] \leq \sqrt{\mathbb{E}[\|P_{\mathcal{T},\mathcal{H}|s} - \hat{P}_{\mathcal{T},\mathcal{H}|s}\|_F^2]} \leq \sqrt{\sum_{j\in[\mathcal{T}], i\in[|\mathcal{H}_s|]} Var\left(\frac{A^l}{N_s}\#(\tau_j, h_{s,i})\Big|s\right)}$$

$$\leq A^l \sqrt{\frac{1}{N_s} \sum_{j\in[\mathcal{T}], i\in[|\mathcal{H}_s|]} \mathbb{P}(\tau_j, h_{s,i}|s)} \leq A^l \sqrt{\frac{1}{N_s}}.$$

Following the same argument with McDiarmid's inequality, we get the second inequality. The remaining inequalities can be shown through similar arguments. Taking over union bounds over all $s, a, o$ gives the Lemma. $\square$

### F.6 Proof of Lemma F.4

*Proof.* As in the proof in Theorem E.1, let us denote $\tilde{b}_{t,s_1} = \tilde{B}_{(o,a,s)_{t-1:1}}\tilde{b}_{1,s_1}$. Then, we can decompose the terms as before:

$$\sum_{r_t, s_{t+1}, \ldots, s_{t+l}} |b_{\infty, s_{t+l}}^\top B_{(o,a,s)_{t+l-1:1}} b_{1,s_1} - \hat{b}_{\infty, s_{t+l}}^\top \hat{B}_{(o,a,s)_{t+l-1:1}} \hat{b}_{1,s_1}|$$

$$= \sum_{r_t,\ldots,s_{t+l}} |\tilde{b}_{\infty,s_{t+l}}^\top \tilde{B}_{(o,a,s)_{t+l-1:t}} \tilde{b}_{t,s_1} - \hat{b}_{\infty,s_{t+l}}^\top \hat{B}_{(o,a,s)_{t+l-1:t}} \hat{b}_{t,s_1}|$$

$$= \sum_{r_t,\ldots,s_{t+l}} |\tilde{b}_{\infty,s_{t+l}}^\top \tilde{B}_{(o,a,s)_{t+l-1:t}} (\hat{U}_{s_t}^\top L_{s_t})(\hat{U}_{s_t}^\top L_{s_t})^{-1} \tilde{b}_{t,s_1} - \hat{b}_{\infty,s_{t+l}}^\top \hat{B}_{(o,a,s)_{t+l-1:t}} (\hat{U}_{s_t}^\top L_{s_t})(\hat{U}_{s_t}^\top L_{s_t})^{-1} \hat{b}_{t,s_1}|$$

$$\leq \sum_{r_t,\ldots,s_{t+l}} \|(\tilde{b}_{\infty,s_{t+l}}^\top \tilde{B}_{(o,a,s)_{t+l-1:t}} - \hat{b}_{\infty,s_{t+l}}^\top \hat{B}_{(o,a,s)_{t+l-1:t}})(\hat{U}_{s_t}^\top L_{s_t})\|_1 \|(\hat{U}_{s_t}^\top L_{s_t})^{-1} \tilde{b}_{t,s_1}\|_1$$

$$+ \sum_{r_t,\ldots,s_{t+l}} \|(\tilde{b}_{\infty,s_{t+l}}^\top \tilde{B}_{(o,a,s)_{t+l-1:t}} - \hat{b}_{\infty,s_{t+l}}^\top \hat{B}_{(o,a,s)_{t+l-1:t}})(\hat{U}_{s_t}^\top L_{s_t})\|_1 \|(\hat{U}_{s_t}^\top L_{s_t})^{-1}(\tilde{b}_{t,s_1} - \hat{b}_{t,s_1})\|_1$$

$$+ \sum_{r_t,\ldots,s_{t+l}} \|\tilde{b}_{\infty,s_{t+l}}^\top \tilde{B}_{(o,a,s)_{t+l-1:t}} (\hat{U}_{s_t}^\top L_{s_t})\|_1 \|(\hat{U}_{s_t}^\top L_{s_t})^{-1}(\tilde{b}_{t,s_1} - \hat{b}_{t,s_1})\|_1.$$

We follow the same induction procedure starting with showing the following equation:

$$\sum_{s_t,r_t,\ldots,s_{t+l-1}} \|(\hat{U}_{s_{t+l}}^\top L_{s_{t+l}})^{-1}(\tilde{B}_{(o,a,s)_{t+l-1:t}} - \hat{B}_{(o,a,s)_{t+l-1:t}})(\hat{U}_{s_t}^\top L_{s_t})\|_1 \leq (1+\Delta)^l - 1.$$

We show this equation by induction as in the previous proof. If $l = 1$, then

$$\|(\hat{U}_{s_{t+1}} L_{s_{t+1}})^{-1}(\tilde{B}_{(o,a,s)_t} - \hat{B}_{(o,a,s)_t})(\hat{U}_{s_t}^\top L_{s_t})\|_1 \leq \sum_{o_t} \Delta_{o_t,a_t,s_t} \leq \Delta,$$

by the definition of $\Delta$. Now we assume it holds for sequences of length less than $l$, and prove the induction hypothesis for $l$. We again split the term into three terms:

$$\sum_{r_t,\ldots,s_{t+l}} \|(\hat{U}_{s_{t+l}}^\top L_{s_{t+l}})^{-1}(\tilde{B}_{(o,a,s)_{t+l-1:t}} - \hat{B}_{(o,a,s)_{t+l-1:t}})(\hat{U}_{s_{t+l-1}}^\top L_{s_{t+l-1}})\|_1$$

$$\leq \sum_{s_t,r_t,\ldots,s_{t+l}} \|(\hat{U}_{s_{t+l}}^\top L_{s_{t+l}})^{-1}(\tilde{B}_{(o,a,s)_{t+l-1}} - \hat{B}_{(o,a,s)_{t+l-1}})(\hat{U}_{s_{t+l-1}}^\top L_{s_{t+l-1}})\|_1 \times$$

$$\|(\hat{U}_{s_{t+l-1}}^\top L_{s_{t+l-1}})^{-1}\tilde{B}_{(o,a,s)_{t+l-2:t}}(\hat{U}_{s_t}^\top L_{s_t})\|_1$$

$$+ \sum_{r_t,\ldots,s_{t+l}} \|(\hat{U}_{s_{t+l}}^\top L_{s_{t+l}})^{-1}(\tilde{B}_{(o,a,s)_{t+l-1}} - \hat{B}_{(o,a,s)_{t+l-1}})(\hat{U}_{s_{t+l-1}}^\top L_{s_{t+l-1}})\|_1 \times$$

$$\|(\hat{U}_{s_{t+l-1}}^\top L_{s_{t+l-1}})^{-1}(\tilde{B}_{(o,a,s)_{t+l-2:t}} - \hat{B}_{(o,a,s)_{t+l-2:t}})(\hat{U}_{s_t}^\top L_{s_t})\|_1$$

$$+ \sum_{r_t,\ldots,s_{t+l}} \|(\hat{U}_{s_{t+l}}^\top L_{s_{t+l}})^{-1}\tilde{B}_{(o,a,s)_{t+l-1}}(\hat{U}_{s_{t+l-1}}^\top L_{s_{t+l-1}})\|_1 \times$$

$$\|(\hat{U}_{s_{t+l-1}}^\top L_{s_{t+l-1}})^{-1}(\tilde{B}_{(o,a,s_{t+l-2:t}} - \hat{B}_{(o,a,s_{t+l-2:t})(\hat{U}_{s_t}^\top L_{s_t})\|_1$$

$$\leq \Delta + \Delta \sum_{r_t,\ldots,s_{t+l-1}} \|(\hat{U}_{s_{t+l}}^\top L_{s_{t+l}})^{-1}(\tilde{B}_{(o,a,s)_{t+l-1:t}} - \hat{B}_{(o,a,s)_{t+l-1:t}})(\hat{U}_{s_t}^\top L_{s_t})\|_1$$

$$+ \sum_{r_t,\ldots,s_{t+l-1}} \|(\hat{U}_{s_{t+l}}^\top L_{s_{t+l}})^{-1}(\tilde{B}_{(o,a,s)_{t+l-1:t}} - \hat{B}_{(o,a,s)_{t+l-1:t}})(\hat{U}_{s_t}^\top L_{s_t})\|_1,$$

where in the last step, we used $\sum_{r_t,\ldots,s_{t+l}} \|(\hat{U}_{s_{t+l}}^\top L_{s_{t+l}})^{-1}\tilde{B}_{(o,a,s)_{t+l-1:t}}(\hat{U}_{s_t}^\top L_{s_t})\|_1 = 1$ as well as $\sum_{r_{t+l-1},s_{t+l}} \|(\hat{U}_{s_{t+l}}^\top L_{s_{t+l}})^{-1}\tilde{B}_{(o,a,s)_{t+l-1}}(\hat{U}_{s_t}^\top L_{s_t})\|_1 = 1$. By induction hypothesis, it is bounded by

$$\Delta + \Delta((1+\Delta)^{l-1} - 1) + (1+\Delta)^{l-1} - 1 = (1+\Delta)^l - 1.$$

With the Lemma, we can verify that

$$\sum_{r_t,\ldots,s_{t+l}} \|(\tilde{b}_{s_{t+l}}^\top \tilde{B}_{(o,a,s)_{t+l-1:t}} - \hat{b}_{s_{t+l}}^\top \hat{B}_{(o,a,s)_{t+l-1:t}})(\hat{U}_{s_t}^\top L_{s_t})\|_1$$

$$\leq \sum_{r_t,\ldots,s_{t+l}} \|(\tilde{b}_{s_{t+l}} - \hat{b}_{s_{t+l}})(\hat{U}_{s_{t+l}}^\top L_{s_{t+l}})\|_\infty \|(\hat{U}_{s_{t+l}}^\top L_{s_{t+l}})^{-1}(\tilde{B}_{(o,a,s)_{t+l-1:t}} - \hat{B}_{(o,a,s)_{t+l-1:t}})(\hat{U}_{s_t}^\top L_{s_t})\|_1$$

$$+ \sum_{r_t,\ldots,s_{t+l}} \|\tilde{b}_{s_{t+l}}(\hat{U}_{s_{t+l}}^\top L_{s_{t+l}})\|_\infty \|(\hat{U}_{s_{t+l}}^\top L_{s_{t+l}})^{-1}(\tilde{B}_{(o,a,s)_{t+l-1:t}} - \hat{B}_{(o,a,s)_{t+l-1:t}})(\hat{U}_{s_t}^\top L_{s_t})\|_1$$

$$+ \sum_{r_t,\ldots,s_{t+l}} \|(\tilde{b}_{s_{t+l}} - \hat{b}_{s_{t+l}})(\hat{U}_{s_{t+l}}^\top L_{s_{t+l}})\|_\infty \|(\hat{U}_{s_{t+l}}^\top L_{s_{t+l}})^{-1}\tilde{B}_{(o,a,s)_{t+l-1:t}}(\hat{U}_{s_t}^\top L_{s_t})\|_1$$

$$\leq \delta_\infty + (\delta_\infty + 1)((1+\Delta)^l - 1).$$

Let $\epsilon_l := \delta_\infty + (\delta_\infty + 1)((1+\Delta)^l - 1)$. From the above, we can conclude that

$$\sum_{a_t,r_t,\ldots,s_{t+l}} \pi(a_{1:t}|h_t)|b_{\infty,s_{t+l}}^\top B_{(o,a,s)_{t+l:1}} b_{1,s_1} - \hat{b}_{\infty,s_{t+l}}^\top \hat{B}_{(o,a,s)_{t+l:1}}\hat{b}_{1,s_1}|$$

$$= \sum_{a_t,r_t,\ldots,s_{t+l}} |\pi(a_{1:t-1}|h_{t-1})b_{\infty,s_{t+l}}^\top B_{(o,a,s)_{t+l-1:t}}\tilde{b}_{t,s_1} - \hat{b}_{\infty,s_{t+l}}^\top \hat{B}_{(o,a,s)_{t+l-1:t}}\hat{b}_{t,s_1}|$$

$$\leq \epsilon_l \pi(a_{1:t-1}|h_{t-1})\|(\hat{U}_{s_t}^\top L_{s_t})^{-1}\tilde{b}_{t,s_1}\|_1 + \epsilon_l \pi(a_{1:t-1}|h_{t-1})\|(\hat{U}_{s_t}^\top L_{s_t})^{-1}(\tilde{b}_{t,s_1} - \hat{b}_{t,s_1})\|_1$$

$$+ \sum_{a_t,r_t,\ldots,s_{t+l}} \pi(a_{1:t-1}|h_{t-1})\|\tilde{b}_{\infty,s_{t+l}}^\top \tilde{B}_{(o,a,s)_{t+l-1:t}}(\hat{U}_{s_t}^\top L_{s_t})\|_1 \|(\hat{U}_{s_t}^\top L_{s_t})^{-1}(\tilde{b}_{t,s_1} - \hat{b}_{t,s_1})\|_1$$

$$\leq \epsilon_l \mathbb{P}^\pi((s,a,o)_{1:t-1}) + (1+\epsilon_l)\pi(a_{1:t-1}|h_{t-1})\|(\hat{U}_{s_t}^\top L_{s_t})^{-1}(\tilde{b}_{t,s_1} - \hat{b}_{t,s_1})\|_1,$$

where we used

$$\pi(a_{1:t-1}|h_{t-1})\|(\hat{U}_{s_t}^\top L_{s_t})^{-1}\tilde{b}_{t,s_1}\|_1 = \pi(a_{1:t-1}|h_{t-1})1^\top (\hat{U}_{s_t}^\top L_{s_t})^{-1}\tilde{b}_{t,s_1} = \mathbb{P}((o,a,s_{1:t}|s_1),$$

and

$$\sum_{r_t,\ldots,s_{t+l}} \|\tilde{b}_{\infty,s_{t+l}}^\top \tilde{B}_{(o,a,s)_{t+l-1:t}}(\hat{U}_{s_t}^\top L_{s_t})\|_1 = \sum_{s_t,r_t,\ldots,s_{t+l}} \|1^\top \tilde{D}_{(o,a,s)_{t+l-1:t}}\|_1 = 1.$$

Since $\epsilon_l < 1$, we get the Lemma. $\qquad\square$

### F.7 Proof of Lemma F.5

*Proof.* Note that from equation (18) in Lemma F.2, we have

$$\sum_{s_1,a_1,r_1,\ldots,r_{t-1},s_t} \pi(a_{1:t-1}|h_{t-1})\|(\hat{U}_{s_t}^\top L_{s_t})^{-1}(\tilde{b}_{t,s_1} - \hat{b}_{t,s_1})\|_1 \leq \epsilon_t.$$

Let $\mathcal{E}_b$ be a bad event where for a sampled trajectory $s_1, a_1, r_1, \ldots, r_{t-1}, s_t$, the difference in estimated probability is larger than $\epsilon_c \mathbb{P}^\pi(s_1, a_1, r_1, \ldots, r_{t-1}, s_t)$, i.e.,

$$\pi(a_{1:t-1}|h_{t-1})\|(\hat{U}_{s_t}^\top L_{s_t})^{-1}(\tilde{b}_{t,s_1} - \hat{b}_{t,s_1})\|_1 \geq \epsilon_c \mathbb{P}^\pi(s_1, a_1, r_1, \ldots, r_{t-1}, s_t).$$

Note that $\mathbb{P}^\pi(s_1, a_1, r_1, \ldots, r_{t-1}, s_t) = \pi(a_{1:t-1}|h_{t-1})1^\top (\hat{U}_{s_t}^\top L_{s_t})^{-1}\tilde{b}_{t,s_1} = \pi(a_{1:t-1}|h_{t-1})\|(\hat{U}_{s_t}^\top L_{s_t})^{-1}\tilde{b}_{t,s_1}\|_1$. If $\mathbb{P}^\pi(\mathcal{E}_b) > \epsilon_t/\epsilon_c$, then

$$\sum_{(s,a,o)_{1:t-1}} \pi(a_{1:t-1}|h_{t-1})\|(\hat{U}_{s_t}^\top L_{s_t})^{-1}(\tilde{b}_{t,s_1} - \hat{b}_{t,s_1})\|_1 \geq \sum_{(s,a,o)_{1:t-1}\in\mathcal{E}_b} \pi(a_{1:t-1}|h_{t-1})\|(\hat{U}_{s_t}^\top L_{s_t})^{-1}(\tilde{b}_{t,s_1} - \hat{b}_{t,s_1})\|_1$$

$$\geq \sum_{(s,a,o)_{1:t-1}\in\mathcal{E}_b} \epsilon_c \pi(a_{1:t-1}|h_{t-1})\|(\hat{U}_{s_t}^\top L_{s_t})^{-1}\tilde{b}_{t,s_1}\|_1$$

$$\geq \epsilon_c \mathbb{P}^\pi(\mathcal{E}_b) > \epsilon_t,$$

which is a contradiction. Similarly, by Theorem E.1, we have

$$\sum_{(s,a,r)_{1:t-1},s_t} |\mathbb{P}^\pi((s,a,r)_{1:t-1}, s_t) - \hat{\mathbb{P}}^\pi((s,a,r)_{1:t-1}, s_t)| \leq \epsilon_t.$$

Following the same argument, we can show the contradiction if the Lemma F.5 does not hold. $\quad\square$

## F.8 Auxiliary Lemmas for Spectral Learning

For completeness of the paper, we include the following lemmas from Appendix B in [20] and [46].

**Lemma F.6 (Theorem 4.1 in [46])** *Let $A \in \mathbb{R}^{m \times n}$ with $m \geq n$ and $\hat{A} = A + E$ for some $E \in \mathbb{R}^{m \times n}$. If singular values of $A$ and $\hat{A}$ are $\sigma_1 \geq \sigma_2 \geq ... \geq \sigma_n$ and $\hat{\sigma}_1 \geq \hat{\sigma}_2 \geq ... \geq \hat{\sigma}_n$ respectively, then*

$$|\sigma_i - \hat{\sigma}_i| \leq \|E\|_2, \qquad \forall i \in [n].$$

The following lemma is also called the Davis-Kahn's $\operatorname{Sin}(\Theta)$ theorem.

**Lemma F.7 (Theorem 4.4 in [46])** *Let $A \in \mathbb{R}^{m \times n}$ with $m \geq n$ with singular value decomposition (SVD) $(U_1, U_2, U_3, \Sigma_1, \Sigma_2, V_1, V_2)$ such that*

$$A = [U_1 \quad U_2 \quad U_3] \begin{bmatrix} \Sigma_1 & 0 \\ 0 & \Sigma_2 \\ 0 & 0 \end{bmatrix} \begin{bmatrix} V_1^\top \\ V_2^\top \end{bmatrix}.$$

*Similarly, $\hat{A} = A + E$ for some $E \in \mathbb{R}^{m \times n}$ has a SVD $(\hat{U}_1, \hat{U}_2, \hat{U}_3, \hat{\Sigma}_1, \hat{\Sigma}_2, \hat{V}_1, \hat{V}_2)$. Let $\Phi$ be the matrix of canonical angles between $range(U_1)$ and $range(\hat{U}_1)$, and $\Theta$ be the matrix of canonical angles between $range(V_1)$ and $range(\hat{V}_1)$. If there exists $\alpha, \delta > 0$ such that $\sigma_{min}(\Sigma_1) \geq \alpha + \delta$ and $\sigma_{max}(\Sigma_2) < \alpha$, then*

$$\max\{\|\sin \Phi\|_2, \|\sin \Theta\|_2\} \leq \|E\|_2/\delta.$$

**Lemma F.8 (Corollary 22 in [20])** *Suppose $A \in \mathbb{R}^{m \times n}$ with rank $k \leq n$ and $m \geq n$, and $\hat{A} = A + E$ with $E \in \mathbb{R}^{m \times n}$. Let $\sigma_k(A)$ be the $k^{th}$ singular value of $A$ and assume $\|E\|_2 \leq \epsilon \cdot \sigma_k(A)$ for some small $\epsilon < 1$. Let $\hat{U}$ be top-$k$ left singular vectors of $\hat{A}$, and $\hat{U}_\perp$ be the remaining left singular vectors. Then,*

- $\sigma_k(\hat{A}) \geq (1 - \epsilon)\sigma_k(A)$.

- $\|\hat{U}_\perp^\top U\|_2 \leq \|E\|_2/\sigma_k(\hat{A})$.

*Proof.* The first inequality follows from Lemma F.6, and the second inequality follows from Lemma F.7 by plugging $\alpha = 0$ and $\delta = \hat{\sigma}_k$. □

**Lemma F.9 (Lemma 9 in [20])** *Suppose $A \in \mathbb{R}^{m \times n}$ with rank $k \leq n$, and $\hat{A} = A + E$ with $E \in \mathbb{R}^{m \times n}$ for $\|E\|_2 \leq \epsilon \cdot \sigma_k(A)$ with small $\epsilon < 1$. Let $\epsilon_0 = \epsilon^2/(1-\epsilon)^2$ and $\hat{U}$ be top-$k$ left singular vectors of $\hat{A}$. Then,*

- $\sigma_k(\hat{U}^\top \hat{A}) \geq (1 - \epsilon) \cdot \sigma_k(A)$.

- $\sigma_k(\hat{U}^\top A) \geq \sqrt{1 - \epsilon_0} \cdot \sigma_k(A)$.

*Proof.* The first item is immediate since $\sigma_k(\hat{U}^\top \hat{A}) = \sigma_k(\hat{A})$. Let $U$ be top-$k$ left singular vectors of $A$. If the top-$k$ SVD of $A$ is $A = U\Sigma V^\top$, then

$$\sigma_k(\hat{U}^\top U\Sigma V^\top) \geq \sigma_{min}(\hat{U}^\top U) \cdot \sigma_k(\Sigma) \geq \sqrt{1 - \|\hat{U}_\perp^\top U\|_2^2} \cdot \sigma_k(A) \geq \sqrt{1 - \epsilon_0} \cdot \sigma_k(A),$$

where the first inequality holds since $V$ is orthonormal and $\hat{U}^\top U$ is full-rank, and the final inequality follows from Lemma F.8. □

**Lemma F.10 (Theorem 3.8 in [46])** *Let $A \in \mathbb{R}^{m \times n}$ with $m \geq n$, and let $\tilde{A} = A + E$ with $E \in \mathbb{R}^{m \times n}$. Then,*

$$\|\tilde{A}^\dagger - A^\dagger\|_2 \leq \frac{1 + \sqrt{5}}{2} \cdot \max\{\|A^\dagger\|_2^2, \|\tilde{A}^\dagger\|_2^2\} \cdot \|E\|_2.$$

---
**Algorithm 6** Exploring Deterministic MDPs with Latent Contexts
---

**Initialization:** For $O(M \log M)$ episodes, observe the possible initial states (discard all other information). If there is only one initial state $s_1$ for $O(M \log M)$ episodes, then set $C_1 = \{(\phi, s_1)\}$. Otherwise, let $C_1$ be a set of all observed initial states with $\{(\{(init, s_1)\}, s_1), \forall s_1 \text{ observed}\}$.

1: **for** $t = 1, ..., H$ **do**
2:     Let $C_{t+1} = \{\}$.
3:     **for** each $(C, s) \in C_t$ **do**
4:         **for** each $a \in \mathcal{A}$ **do**
5:            Let $O = \{\}$.
6:            1. Find any action sequence $a_1, ..., a_{t-1}$ that can result in a state $s$ with distinguishing observations $C$.
7:            2. For $O(M \log M)$ episodes, run the action sequence $a_1, ..., a_{t-1}$ (execute any policy for the remaining time steps).
8:            2.1. If we reached the state $s$ with distinguishing observations $C$, then run action $a$, and get a new observation of next state and reward $(s', r)$.
9:            2.2. Update $O \leftarrow O \cup \{(s', r)\}$ and record the probability $p$ of observing $(s', r)$ conditioned on $C$ and $s$.
10:           **if** $|O| = 1$ **then**
11:              3.1. With the only element $(s', r) \in O$, update $C_{t+1} \leftarrow C_{t+1} \cup \{(C, s')\}$.
12:            Record that there is a path from $(C, s)$ to $(C, s')$ by taking action $a$ with a reward $r$.
13:           **else**
14:              3.2. For all $(s', r) \in O$, let $C' = C \cup \{(s, a, s', r)\}$. Update $C_{t+1} \leftarrow C_{t+1} \cup \{(C', s')\}$.

15:              Record that there is a path from $(C, s)$ to $(C', s')$ by taking action $a$ with a reward $r$ and a recorded probability $p$.
16:           **end if**
17:         **end for**
18:     **end for**
19: **end for**

---

## Appendix G    Guarantees for Latent Deterministic MDPs

In this section, we provideupper bound for LMDP instances with deterministic MDPs. The upper bound for latent deterministic MDPs supports our conjecture on the sample complexity of learning general LMDPs, and could be of independent interest.

Although the lower bound is exponential in $M$, it could be tolerable with sufficiently small number of contexts $M = O(1)$. In this appendix, we briefly discuss whether the exponential dependence in $M$ is sufficient for learning deterministic MDPs with latent contexts. If that is the case, then we can exclude the possibility of $\Omega(A^H)$ lower bound for deterministic LMDP instances. Intuitively, the exponential dependence in time-horizon is unlikely in LMDPs for the following reason: under certain regularity assumptions, if the time-horizon is extremely long $H \gg S^2 A$ such that every state-action pair can be visited sufficiently many times, then each trajectory can be easily clustered and the recovery of the model is straight-forward. The following theorem shows that we do *not* suffer from $\Omega(A^H)$ sample complexity for deterministic LMDPs:

**Theorem G.1 (Upper Bound for Deterministic LMDPs)** *For any LMDP with a set of deterministic MDPs, there exists an algorithm such that it finds the optimal policy after at most* $O\left(H(SA)^M \cdot poly(M) \log M\right)$ *episodes.*

The algorithm for the upper bound is implemented in Algorithm 6. While the upper bound for deterministic LMDPs does not imply the upper bound for general stochastic LMDPs, we have shown that the exponential lower bound higher than $O((SA)^M)$ cannot be obtained via deterministic examples. We leave it as future work to study the fundamental limits of general instances of LMDPs, and in particular, whether the problem is learnable with $\tilde{O}((SA)^M)$ sample complexity, which can be promising when the number of contexts is small enough (*e.g.,* $M = 2, 3$).

## G.1 Proof of Theorem G.1

Algorithm 6 is essentially a pure exploration algorithm which searches over all possible states. After the pure exploration phase, we model the entire system as one large MDP with $poly(M) \cdot (SA)^M$ states. The optimal policy can be found by solving this large MDP.

The core idea behind the algorithm is that since the system is deterministic, whenever there exist more than one possibility of observations (a pair of reward and next state) from the same state and action, it implies that at least one MDP shows a different behavior from other MDPs for the state-action pair. Therefore, each observation can be considered as a new distinguishing observation that can separate at least one MDPs from other MDPs. Afterwards, we can consider a sub-problem of exploration in the remaining time-steps given the distinguishing observation in history and the current state. The argument can be similarly applied in sub-problems, which leads to the concept of conditioning on a set of distinguishing observations and the current state.

On the other hand, if an action results in the same observation for all MDPs given a set of distinguishing observations and a state, then we would only see one possibility. In this case, this state-action pair does not reveal any information on the context, and can be ignored for future decision making processes.

Algorithm 6 implements the above principles: for each time step $t$, we construct a set of all reachable states with a set of distinguishing observations in histories. In order to find out all possibilities, for each observation set, state, and action we first find the action sequence by which we can reach the desired state (with target distinguishing observation set). Since all MDPs are deterministic, the existence of path means at least one MDP always results in the desired state with the action sequence. The sequence can be found by the induction hypothesis that we are given all possible transitions and observations in previous time steps $1, ..., t - 1$. By the coupon-collecting argument, if we try the same action sequence for $O(M \log M)$ episodes, we can see all different transitions that all different MDPs resulting in the target observation set and state can give. By doing this for all reachable states and observation sets, we can find out all possibilities that can happen at the time step $t$. The procedure repeats until $t = H$ and eventually we can find all possible outcomes from all action sequences.

An important question is how many different possibilities we would encounter in the procedure. Note that as we find out a new distinguishing observation, we cut out the possibility of at least one MDP conditioning on that new observation. Since there are only $M$ possible MDPs, the size of distinguishable observation sets cannot be larger than $M - 1$. Based on this observation, we can see that the number of all possible combinations of the observation set and state is less than $\binom{MSA}{M-1} \cdot S$. Note that the $MSA$ is the total number of possible state-action-observation $(s, a, s', r)$ pairs. Hence in each time step, the iteration complexity does not exceed $\binom{MSA}{K-1} \cdot SA$ times the number of episodes for each possible state and observation set. Since we loop this procedure for $H$ steps, the total number of episodes is bounded by $O\left(HSA\binom{MSA}{M-1} \cdot M \log M\right)$, which results in the sample complexity of $O\left(H(SA)^M \cdot poly(M) \log M\right)$.

## Appendix H  Checklist