# OpenReview forum: "RL for Latent MDPs: Regret Guarantees and a Lower Bound"
_NeurIPS.cc/2021/Conference — NeurIPS 2021 Spotlight_

### Official Review · Reviewer_pwrf · 2021-07-16

**Rating:** 8
**Confidence:** 4

**Summary:**

This paper studies episodic RL in the regret minimization setup in Latent MDPs (LMDPs), which belong to the family of POMDPs. The paper presents a sample complexity lower bound for general LMDPs (i.e., in the absence of any restrictive assumption on the LMDP) that grows exponential with the number $M$ of contexts (or base MDPs) --- that is, learning a near-optimal policy in the worst-case may require an exponential (in $M$) number of episodes. It then presents L-UCRL, a UCRL2-style algorithm for LMDPs, which is proven to achieve $O(\sqrt{T})$ regret with high probability under some assumptions, where the regret bound depends polynomially on the number of state-action pairs and on the number of base MDPs.

**Limitations And Societal Impact:**

No potential societal impact, to my knowledge. For other limitations or constructive suggestions, see above.

**Main Review:**

The paper is written very well and admits a clear presentation. It investigates an interesting but challenging problem, which in my opinion is relevant both in practice and for NeurIPS.

Strengths: Although LMDPs were studied prior to this work under various names, no regret or sample complexity bound was provided for them, to my knowledge. The LB is new and interesting, and would be a nice addition to the literature. The main contribution of the paper is an algorithm for the considered LMDP setup, which under some separability assumptions, is shown to be sample efficient, enjoying a rate-optimal regret with polynomial dependencies on the size of the state-action space and the episode length.

The paper starts with a warm-up case, when contexts are revealed at the end of the episode, which I believe is interesting in its own right. It was interesting to see that optimism is maintained by just relying on a hidden reward function suitably injected to empirical estimates.
Overall the studied problem is highly relevant in practical aspects of RL, and the paper presents a solid contribution, advancing state-of-the-art. However I have a couple of concerns:

- Theorem 3.4 presenting the main technical result of the paper requires some further scrutiny. It is claimed to improve over the result by Li and Brunskill as the latter requires $H=\Omega(S^2A/\delta^2)$, while that in Theorem 3.4 depends on $S$ and $A$ only logarithmically. The comparison is not thorough however: The paper remains silent about the fact that here: (i) $H$ depends on $\delta^{-4}$ (much worse than $\delta^{-2}$ in Li and Brunskill [8]), and (ii) one has $H=\Omega(\log(HK))$, that is the episode length has to grow with time (or the current implementation is not anytime). Could you please explain?
- Further discussion related to Theorem 3.4 might be helpful. Specifically, $H$ has to satisfy a bound dependent on $\alpha$, which itself is determined via an inequality depending on both $\delta$ and $S$. What is the final, explicit dependency of minimal $H$ on the relevant parameters (in particular, $\delta$ and $S$)?

Some minor comments:

- In line 67, the authors mention “initialization”, which is seemingly supposed to be discussed/motivated in the previous statements (though, some pointer to that is provided in the abstract). Discussions in later sections make it clear, however. Clarifying that could improve the readability.

- A more concrete comparison between the regret bound of Theorem 3.3 and of Theorem 3.4 could be very helpful. Do the bounds together imply that one achieves the same regret (order-wise), but only at the expense of large enough $H$ and separability? Or we would lose more.

- Throughout: UCRL --- > UCRL2. If you refer to the algorithm in (Jaksch et al., 2010), you should use UCRL2 instead of UCRL. UCRL is indeed the precursor of UCRL2, and is presented in (Ortner & Auer, 2007).


**Time Spent Reviewing:**

6

---

> ### Author Response · Authors · 2021-08-10
> **Response to Reviewer pwrf**
>
> We appreciate your constructive feedback.
>
> **Conditions on the length of episodes $H$:** First, we would like to emphasize that comparing to Li \& Brunskill, there is a big difference between assuming $H \gg \Omega(SA)$ and allowing $H \ll SA$, since in the former case it is possible to visit {\it all} state-action pairs multiple times within a single episode, which significantly simplifies the problem.
>
> As you point out, in the $H \ll SA$ regime, we currently have worse dependence on $\delta$ and $\log K$. There are two main technical challenges. First, we need $H = O(\delta^{-4})$ to infer the true contexts in hindsight, w.h.p. We conjecture that it might be tightened to $O(\delta^{-2})$: information-theoretically, given our $\delta$-separation assumption, it would be possible to distinguish different MDPs using a length $H = O(\delta^{-2})$ trajectory. However, the EM implementation requires the stronger argument that an estimated belief for a length-$H$ trajectory has probability $\approx$ 1 for the true context.
>
> Second, logarithmic dependency on $K$ serves to ensure the correctness of inferred contexts for all $K$ episodes. Removing this may be possible using a more delicate (and much more complicated) EM analysis that would allow for a small chance of errors in the inferred contexts. It is an interesting question whether this dependence can be eliminated under similar separation assumption as we make.
>
> We think these issues are quite non-trivial, and in fact might be of sufficient independent interest to be addressed in a separate work. We added a detailed discussion in the last paragraph of Appendix D.3 to emphasize this issue. We will add a pointer to this remark in the revision.
>
>
> **Order of $H$ in Theorem 3.4:** The eventual order of $H$ is $O(\delta^{-4} \log^2(S/\delta))$, which is poly-logarithmic in $S$, instead of polynomial is $S$ as in Li and Brunskill [8]. We wanted to elaborate on the discrepancy in the dependence in $S$, nevertheless, we will also comment on the dependence in $\delta$ in the future version. We will explicitly point this out.
>
>
>
>
> **Theorem 3.3 vs Theorem 3.4:** You are correct - essentially the two theorems guarantee the same order of regret, at the expense of separation and initialization as you said.
>
> We also thank you for your other comments. We will fix them accordingly.

---

> > ### Comment · Reviewer_pwrf · 2021-08-29
> > **Response to Rebuttal**
> >
> > I have read the other reviews and the rebuttal, and would like to thank the authors for their responses to my comments. The rebuttal addresses my questions and I believe incorporating the promised changes into the revised version makes this readily well-executed paper even more solid. Thus, I increase my score to 8.

---

> > > ### Author Response · Authors · 2021-09-02
> > > **Thank you for the response!**
> > >
> > > We appreciate your positive feedback and increasing the score!

---

### Official Review · Reviewer_Nz8R · 2021-07-17

**Rating:** 7
**Confidence:** 3

**Summary:**

The authors propose an online algorithm to deal with the latent MDP setting. Theoretical lower bounds and upper bounds for the regret are derived under various assumptions.

**Limitations And Societal Impact:**

Yes.


**Main Review:**

Overall, the presentation of the main ideas is clear and the paper is easy to read.

The extension of UCRL to the latent-MDP setting seems novel. I did not check the proofs but the results as well as the arguments are convincing. I believe the contributions are solid for publication in NeurIPS.

For the planning part, the authors keep pointing to the PBVI algorithm in the main text but in the experiments, Q-MDP is used. I wonder whether there is any violation of the "planning-oracle" assumption with respect to Proposition (or Lemma?) 3.2.

The authors mention that even the case where the true context is revealed at the end of each episode is motivated by real-world examples. Perhaps the authors should also provide these examples in the text (or even better in the experiments).

**Time Spent Reviewing:**

2

---

> ### Author Response · Authors · 2021-08-10
> **Response to Reviewer Nz8R**
>
> We appreciate your review and suggestions.
>
> **$Q$-MDP in experiment:** Q-MDP is a well-used heuristic due to its simple implementation and good empirical performance. Proposition 3.2 says that the optimism holds for {\it any} policy, thus the optimism argument is not violated regardless of the planning algorithm we use.
>
> **Real-world examples for context-revealing setting:** One possible example could be a medical-decision making problem, where an underlying disease is a latent context which may be identified after temporary prescription is made. We will strengthen this motivation in the final version.

---

### Official Review · Reviewer_8oDh · 2021-07-27

**Rating:** 6
**Confidence:** 3

**Summary:**

In this paper, the authors study efficient algorithms and hardness results for Latent MDPs. In a latent MDP, in each iteration, an MDP is randomly drawn from a set of M possible MDPs, and the agent interacts with that MDP without knowing its identity. The goal is to find a policy so that the expected cumulative reward is maximized, where the expectation is taken over the choice of the MDP and its stochasticity. The authors first show that without further assumptions, LMDPs are hard to learn. The authors then give a set of sufficient conditions so that LDMPs are learnable with polynomial sample complexity. One possible condition that makes LMDPs learnable with polynomial number of samples is the case when the identity of the randomly chosen MDP is revealed to the agent at the end of each trajectory. The authors further show that such identity can be learned if certain seperatedness condition holds. Preliminary experimental results are attached.

**Limitations And Societal Impact:**

See main review

**Main Review:**

Overall, this paper contains an interesting set of results and I recommend acceptance.

The hardness result looks interesting, and suggests that extra conditions are indeed necessary to make LMDPs efficiently learnable. The authors then show that if the identity of the MDP is revealed to the agent at the end of each trajectory, then LMDPs are learnable with sublinear regret / polynomial sample complexity. The idea is to use the identity of the MDP to build confidence intervals, with which standard algorithms based on the principle of optimism in the face of uncertainty can be applied. The authors then discuss conditions under which the identity of the MDP can be learned.

Although the writing is good in general, the statement of the hardness result seems a bit incomplete. In particular, in the statement of Theorem 3.1, there is no constraint on the value of H. It would be better if the authors state that H=M in the hardness result, since the hardness result will be trivial if H can be arbitrarily large.

Moreover, the proof in Appendix B is not satisfying. The argument seems to be pretty informal, and should better be regarded as a proof sketch instead of a formal proof. In particular, statements like "therefore, we cannot gain any information from executing wrong action sequences besides of eliminating this wrong action sequence. " should be made formal using information theory.


**Time Spent Reviewing:**

5

---

> ### Author Response · Authors · 2021-08-10
> **Response to Reviewer 8oDh**
>
> We appreciate your time and work for the review.
>
> **Formal Proof of the Lower Bound:** The reviewer made a thoughtful observation on the lower bound result. First, in our lower bound result we mainly focused on the exponential lower bound $(SA)^M$, and less focused on polynomial dependency on other parameters (and often, polynomial dependency on parameters like $H$ is considered acceptable). Studying a tight lower bound on all parameters with matching polynomial factors could also be of interest in future.
>
> Second, in order to make the argument completely formal, we can apply a recent simplified lower bound analysis for multi-armed bandits from [1] for instance. Their main Theorem 1 and its proof in [1] says that for multi-armed bandit problems with arm sets $\mathcal{A}$,
>
> $\sum_{a \in \mathcal{A}} \mathbb{E} [N_{a} (T)] \ge \ln (T) / KL(\nu_a, \nu_a')$
>
> where $N_a (T)$ is the number of times that a sub-optimal arm $a$ is played, $\nu_a$ is a reward-distribution when $a$ is played, $\nu_a'$ is a reward-distribution when $a$ is played in a slightly different bandit problem with $a$ being the optimal arm, and $KL$ is the KL-divergence.
>
> In our hard instances, since all $A^M$ action-sequences yield the same trajectory distributions (except the last reward), we can slightly adapt the proof strategy from [1] and essentially apply the same argument as if it is a multi-armed bandit problem with $A^M$-arms. We will strengthen our lower-bound proof in the appendix based on this argument.
>
>
> [1] Garivier, Aurélien, Pierre Ménard, and Gilles Stoltz. "Explore first, exploit next: The true shape of regret in bandit problems." Mathematics of Operations Research 44.2 (2019): 377-399.

---

### Official Review · Reviewer_RsHS · 2021-08-03

**Rating:** 8
**Confidence:** 4

**Summary:**

The authors in this work consider RL under the latent MDP model, where an unknown MDP is sampled from a known set of $M$ MDPs each episode. In this setting, the authors first show a general statistical lower bound which indicates that (approximately) learning optimal policy in the L-MDP setting requires exponential number of episodes (exponential in $M$). Given the hardness result, the authors first resolve the problem in the standard *online* framework where each episode is revealed to the learner after the episode itself: in this scenario, they provide an optimistic parameter estimation algorithm that eventually yields an efficient algorithm with sublinear regret. Finally, the authors show that the learner can actually infer these episodes after they are played assuming separation between the different MDPs and benign initialization (which is also investigated and relaxed into other assumptions such as test sufficiency and reachability of MDPs).

**Limitations And Societal Impact:**

Sufficiently addressed.

**Main Review:**

I believe this is a submission ready for publication. It's a good paper, tackling a problem of potential significance in the community. The paper itself provides both negative and positive results, and gives a full view of the problem in terms of what is possible, what is necessary, etc. Such papers are much appreciated. Clear accept from me.

**Time Spent Reviewing:**

8

---

> ### Author Response · Authors · 2021-08-10
> **Response to Reviewer RsHS**
>
> We highly appreciate your positive and encouraging feedback. We will do our best to finalize the paper.

---

### Decision · Program_Chairs · 2021-09-27

**Decision:**

Accept (Spotlight)

**Comment:**

In this paper, the authors study efficient algorithms and hardness results for Latent MDPs. In a latent MDP, in each iteration, an MDP is randomly drawn from a set of M possible MDPs, and the agent interacts with that MDP without knowing its identity. The goal is to find a policy so that the expected cumulative reward is maximized, where the expectation is taken over the choice of the MDP and its stochasticity. The authors first show that without further assumptions, LMDPs are hard to learn. The authors then give a set of sufficient conditions so that LDMPs are learnable with polynomial sample complexity. One possible condition that makes LMDPs learnable with polynomial number of samples is the case when the identity of the randomly chosen MDP is revealed to the agent at the end of each trajectory. The authors further show that such identity can be learned if certain seperatedness condition holds. Preliminary experimental results are attached. There is a clear consensus amongst reviewers that this is a strong accept.